# Aridity synthesis for 8 selected key regions of the global climate system during the last 60 000 years

Florian Fuhrmann[1], Benedikt Diensberg[1], Xun Gong[2], Gerrit Lohmann[2], Frank Sirocko[1]

[1]Department for Geoscience, Johannes-Gutenberg-Universität, Mainz, 55099, Germany

[2]Alfred Wegener Institute for Polar and Marine Research, Bremerhaven, Germany

*Correspondence to*: Florian Fuhrmann, (flfuhrma@uni-mainz.de)

**Abstract.** A compilation of published literature on dust content in terrestrial and marine sediment cores was synchronized with pollen data and speleothem growth phases on the GICC05 time axis. Aridity patterns for eight key areas of the global climate system have been reconstructed for the last 60,000 years. These records have different time resolutions and different dating

methods, i.e. different types of stratigraphy. Nevertheless, all regions analyzed in this study show humid conditions during early marine isotope stage 3 (MIS3) and the early Holocene or deglaciation, but not always at the same time. Such discrepancies have been interpreted as regional effects, although stratigraphic uncertainties may affect some of the proposed interpretations. In comparison, most of the MIS2 interval becomes arid in all of the northern hemisphere records, but the peak arid conditions of the Last Glacial Maximum (LGM) and Heinrich event 1 differ in duration and intensity among regions. In addition, we also

compare the aridity synthesis with modelling results using a Global Climate Model (GCM). Indeed, geological archives and GCMs show agreement of aridity pattern for the Holocene or deglaciation, LGM and for the late MIS3.

## 1 Introduction

Two of the main foci of Paleoclimate research today are: i) well dated, high resolution archives of past climate (e.g., marine and terrestrial sediments, speleothems, tree rings and ice cores), ii) modelling of global and regional characteristics with Global

Climate Models (GCM), which include main processes in atmosphere, ocean, land and cryosphere as well as their coupling. Geological archives have the potential to provide information on the past states of climate variables at the global and regional level, and their evolution in time. The strength of modelling approach, on the other hand, is to understand the processes of climate change and global teleconnections. A reliable model of past climate change should faithfully reproduce the observed climate patterns as reconstructed from geo-archives. We will test this prerequisite for a set of records that approximating past

global aridity.

This paper emerges from the PalMod project, which among other things is developing a GCM time series of past global temperatures for the last glacial cycle (www.palmod.de). One prerequisite of the project was to work only with publicly available datasets. Thus, we had to use only available datasets from publicly accessible databases (PANGAEA, NOAA-NCDC,

Neotoma (global pollen database), ice core database from Copenhagen university, SISAL (speleothem database) and EPD (European pollen database)). For the calculation of the aridity index, the most complete and best dated records, dating back to 60 000 years b2k with the highest possible sample resolution were used. We could not include the records that do not meet these requirements in the calculation of the aridity index. All data were plotted on the age scale b2k.

We use continuous time series covering as much as possible of Holocene, 0 - 11 700 years before 2000 CE (yr b2k), Marine Isotope Stage 2 (MIS2) (25 000 - 18 000 yr b2k) and also the flickering climate of MIS3 (60 000 – 25 000 yr b2k, with MIS boundaries revised by (Spratt and Lisiecki, 2016)).

In this paper, we concentrate on published paleoclimate reconstructions for aridity, which is one of the most important indicators of climate change. We have screened published paleoclimate literature of the last 30 years to detect and select key

areas. These key areas were defined by the proxy availability, i.e. pollen, dust and speleothem growth must provide three independent sources of information related to past precipitation. These areas were selected because they were the smallest possible regions meeting the following criteria: i) publicly available records from data repositories, ii) well dated, iii) sufficient sample resolution of at least 1 sample per 250 years and as far back as possible up to 60 000 yr b2k. Several high-resolution data sets reaching as far as MIS3 (see Fig. S11, dashed lines) are available in the literature, but have not been included in this

synthesis, either because these archives are too far away from the selected key regions to fit into a suitable synthesis or because they cover only a small part of the last 60 000 years. Some important records of paleoclimatic research are not included because only one or two of the aridity proxies for the region are available and complementary data are not available. Many other cores are excellent archives, but cannot be incorporated into the numerical calculation of the aridity index, because the data are not accessible from official data repositories. Sediment cores from e.g. Lake Tulane (Grimm et al., 2006), Bear Lake (Jiménez-

Moreno et al., 2007), Lake Suigetsu (Bronk Ramsey et al., 2012), Petén-Itzá (Correa-Metrio et al., 2012) and Potrok Aike (Kliem et al., 2013) were not included in the aridity index as there were no other long time series with publicly available datasets beside pollen within the region of the archive. The excellent loess archive of Nussloch (Antoine et al., 2001; Moine et al., 2017) could not be used for the synthesis because no accessible data were available in official data repositories. The record of Dunaszekcso loess (Újvári et al., 2015) or pollen records from Tenaghi Phillipon (Pross et al., 2015) are not

incorporated, because no data are available that covered at least a longer period of the last 60 000 years.

Dust is mainly deflated in regions with less than 200 mm/a precipitation, and thus indicate an arid climate (either subtropical or polar) (Pye, 1987). Speleothem growth needs dripping water in a cave, and thus rain or snow melt (Spötl and Mangini, 2002). Arboreal pollen implies more precipitation than in a landscape with abundant grass pollen. Accordingly, we do not evaluate the full width of information from these paleoclimate proxies, but just reduce the evidence to its basic structure, which

is aridity. The most faithful aridity indicator is dust, which indicates deserts, whereas grass indicates steppic landscapes. Throughout this synthesis, we use 'climate improvement' for warmer and wetter climate conditions, whereas 'climate deterioration' (or similar terms) accord for colder and drier climate.

Figure 1 present the key regions. The detailed evidence for each of the selected key regions and their well dated and high-resolution proxy records are presented in Fig. 2 and Supplement S1-S7. The discussion compares the synoptic aridity

reconstruction for the time of LGM (26 500 – 19 000 yr b2k), and late MIS3 (32 000 yr b2k) with GCM simulations. Mix et al. (2001) define the LGM from comparably stable conditions during the time interval of 23 000 to 19 000 yr b2k. Clark et al. (2009) define the LGM by the maximum extend of the ice sheets and sea level low stand to 26 500 to 19 000 yr b2k for most parts of northern and southern hemisphere. We follow the wider definition of Clark et al. (2009), which encompasses the

regional differences in the results of this work.

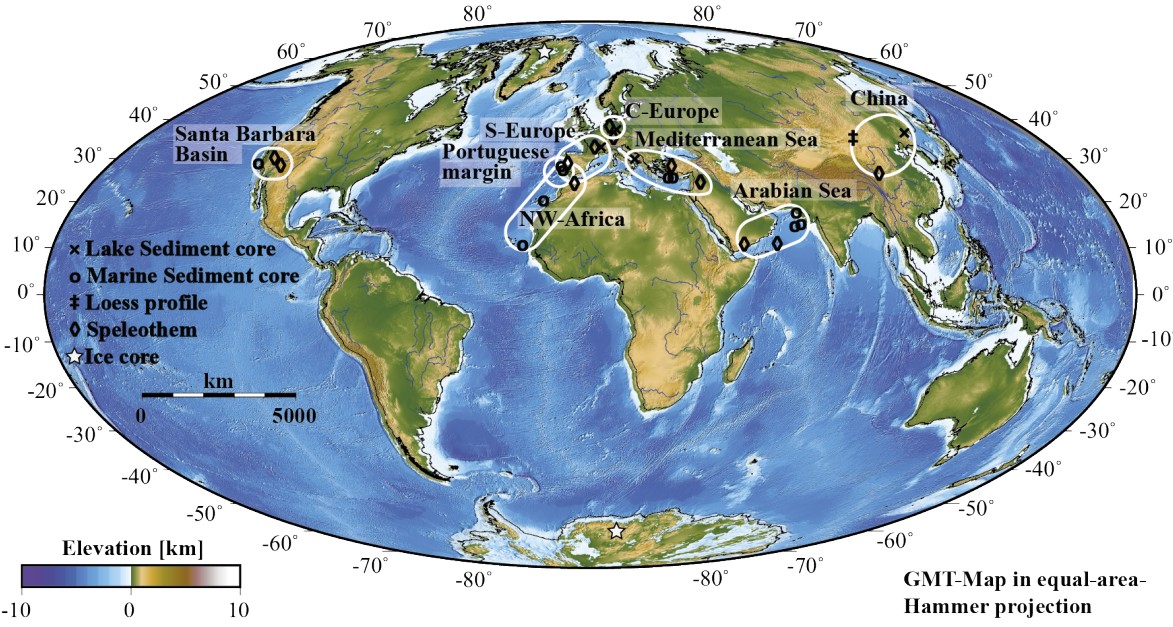

**Figure 1: Global map with the 8 selected key regions and archive type**, Generic Mapping Tool (GMT) (Wessel and Smith, 1991). Regions are indicated with white boundaries. Crosses sign lake sediment cores; open circles marine sediment cores; double sharp marks loess profile; Diamonds mark speleothems and white stars ice cores. The map is colour coded for the elevation.

We start the synthesis with Central Europe: Dust and pollen records from Eifel (Seelos et al., 2009; Sirocko et al., 2016), which we compare with speleothem data from nearby Bunker Cave (Fohlmeister et al., 2012, 2013; Weber et al., 2018) as well as the Spannagel Cave in Austria (Holzkämper et al., 2004, 2005; Spötl and Mangini, 2002). The speleothems of the Spannagel and Bunker Cave show very similar growth patterns and can be combined (Fohlmeister et al., 2012). The maar sediment cores of the Eifel Laminated Sediment Archive (ELSA)-project (Sirocko et al., 2016) show all Greenland Stadials (GS) and Greenland

Interstadials (GI) in the time series of eolian dust content (Dietrich and Sirocko, 2011; Seelos et al., 2009). Central Europe shows accordingly the same climatic structures, which are well known in North Atlantic marine sediments (e.g. Hodell et al., 2013; McManus et al., 1994; Naafs et al., 2013) and Greenland ice cores (North Greenland Ice Core Project Members et al., 2004; Rasmussen et al., 2014; Svensson et al., 2008). This Central European climate time series is then compared with the respective time series from all other key regions, which are documented in the supplement.

## 2 Methods

The synthesis is based on pollen profiles from sediment cores, growth phases of speleothems and several dust proxies like grain size of eolian fraction within sediment cores. We used the original stratigraphy of all records on the age scale of yr b2k but we are aware of a general error of up to ± 2 000 years for all MIS 3 dates.

Especially the growth phases of speleothems are a significant indicator for presence of mobile water. If a speleothem could grow, at least some precipitation occurred over the cave. If a speleothem does not grow (hiatus), either no precipitation or a change of the drip water system above the cave could be causes. Most speleothem datasets also provide $\delta^{18}O$ or other isotope measurements (Genty et al., 2003; Hoffmann et al., 2016; Wainer et al., 2009) which often have local characteristics, thus our index only uses growth phases as the most common and robust aridity indicator.

Pollen were divided into two classes for this work. Trees and shrubs (in the following only called tree pollen) have been grouped together because they require similar climatic conditions for their growth. Herbs and grasses (in the following only called grass pollen) were also combined. Trees require a much higher amount of rainfall than grasses for their growth. A simple statement about the relative precipitation in the catchment area of the core can therefore be made using the tree / grass pollen ratio. In general, a higher amount of tree pollen indicates a warmer and wetter climate than high amounts of grass pollen. The

temporal resolution of the pollen profiles is often low, but we have selected the available data with the highest resolution from each selected region for comparison.

The best chronology for the northern hemisphere is from the annual layer counted NGRIP ice core in Greenland (Rasmussen et al., 2014). These ice core data include also dust (Ruth et al., 2003, 2007). This record is a backbone of dust records, but outside of the chosen regions of the aridity index. Therefore, the NGRIP dust concentration was only used in comparison to

20 central European dust from the ELSA-Dust-Stack (2009).

Several dust proxies were used for this synthesis due to a large variety in dust over the key regions. Calcium carbonate ($CaCO_3$) in the ocean is deposited in higher rates with warmer sea surface temperatures if cores are situated above the lysocline. Therefore, lower $CaCO_3$ contents of ocean sediments in regions of high dust deposition (e.g. deMenocal et al., 2000; Leuschner and Sirocko, 2003) indicate increased mainly aridity, but is also effected by changing wind directions and ocean temperature

change. This proxy is thus used only for the construction of the aridity index in Portuguese margin, off west Africa and Arabian Sea. The grainsize record from the Jingyuan loess plateau in China shows phases of aridity. The larger the sediment grains, the lower the precipitation and temperature and the higher the wind speeds (Sun et al., 2010; Xiao et al., 1995, 2015). Dust content in sediments off NW-Africa is given in percentages of the whole sample composition. The ratio kaolinite/chlorite (K/C) is a dust proxy for the Mediterranean region (Ehrmann et al., 2017). Higher K/C ratios (more kaolinite than chlorite)

indicate an increased aeolian dust transport. During humid periods, kaolinite was stored within lakes or basins - due to increased erosion - and deflated during arid periods (Ehrmann et al., 2017).

## 2.1 Data treatment and aridity index calculation

We have generated all synthesis plots for the regions with ELSAinteractive++ (Diensberg, 2020). This software was written in C++ for the subproject PalMod at the University of Mainz. This software was developed for working with sediment cores on age or depth scale. Each data set of speleothem growth phases, tree pollen and dust proxies was resampled by linear interpolation to 50 years resolution. Subsequently, the resampled data sets were converted into index values. Therefore, the data for all time series were converted into percentages if the original data were not given as percentages. This was done by setting the maximum value of each data set to 100%, the minimum value to 0%, and normalizing values in between. Speleothem growth phases give information on humid phases, however "no growth" can either indicate changes in the dripwater system or arid conditions. Grass pollen act as a counterpart to tree pollen which indicate humidity. Grass pollen values and dust values are considered to be the more robust aridity proxies, compared to speleothem growth phases. Therefore dust proxies and tree pollen index values were separated into three parts (0, 1, 2) while speleothem growth was only separated into two parts (0, 1). Speleothem growth is given index values of 1, index values of 0 indicate no growth. Tree pollen contents below 33 % were assigned index values of 0. Tree pollen contents between 33 % and 66 % were assigned index value 1 and above 66 % index value 2 (see Table 1). Higher dust values are assigned to lower index values, since less precipitation and thus lower soil moisture is the prerequisite for dust deflation. Thus dust proxy values larger than 66% are assigned to the aridity index value 0, dust proxy values between 33% and 66% to the aridity index value 1 and dust proxy values below 33% to the aridity index value 2. The index values are then finally added up (a template is available in supplementary materials). The aridity index ranges from 0 (very dry conditions) to 5 (very humid conditions). Speleothem growth phases, higher tree pollen values and lower dust values combined indicate increased humidity.

The aridity index for all key regions uses the three proxy types (speleothem growth, tree pollen, dust proxy), except for St. Barbara basin region, where a dust record is not available. For Arabian Sea region, we used TOC as aridity proxy instead of tree pollen, as there were no available pollen data in databases but an excellent organic carbon record. High TOC in the Arabian Sea sediments is caused by high SW-monsoon intensity, intense upwelling and surface water nutrient content, high flux rated of organic matter causing low deep water oxygen content (Schulz, et al., 1998; Sirocko et al., 1993; Sirocko and Ittekkot, 1992). We use only one record per region for dust and vegetation, because there are few and these proxy records are very difficult to combine. For speleothems we use several records for the regions central Europe, Arabian Sea, Mediterranean Sea and St. Barbara basin. Since we only use age dating for Speleothem records, they can be easily combined. In addition, the caves are close to each other or have already been compared in previous works (Central Europe, Fohlmeister et al., 2012; Arabian Sea, Fleitmann et al., 2007; Shakun et al., 2007; Mediterranean Sea region, Ünal-Imer et al., 2015; St. Barbara basin: Cave of the Bells and Fort Stanton speleothems show the same climate signals and can both be compared with the St. Barbara basin).

| Speleothems | | Tree pollen | | Eolian dust | |
|---|---|---|---|---|---|
| 0 | no speleothem growth | 0 | tree pollen values < 33% | 0 | dust values > 66% |
| 1 | speleothem growth | 1 | tree pollen values > 33% & < 66% | 1 | dust values < 66% & > 33% |
| | | 2 | tree pollen values > 66% | 2 | dust values < 33% |

**Table 1. Components of the aridity index**: Speleothems can either account as value 0 (no speleothem growth) or 1 (speleothem growth); Tree pollen values below 33 % do not add to the aridity index, between 33 % and 66 % they account for index value 1 and above 66 % for 2; Dust values were internally normalized and act inverse to tree pollen. Dust values above 66 % do not increase the index, between 66 % and 33 % they count as value 1 and below 33 % as value 2. The aridity index ranges from 0 (highly arid conditions) to 5 (highly humid conditions).

## 2.2 Error estimates

In general, the main uncertainties of the proxies are the measurements of the original data, but for most original data no measurement uncertainties are specified, although each measurement has inaccuracies. Sources of error can be, for example, incorrectly counted pollen or device errors of the measuring instruments. As the uncertainty is difficult to assess quantitatively, we have applied a simple Monte Carlo simulation based on error estimates to get an approximation of the total error. To do this, we used the initial error values as displayed in Table 2. We estimated these based on the data density and the method originally used. Error estimates are different for each region, because the archives have different time-resolution as well as different methods were used. For example, 10% error in the pollen data of NW-Africa are assumed, as there is a low sample resolution within the dataset. The error for the Mediterranean Sea dataset is considered small at 2% because the data resolution is high and the stratigraphy with warves and age model is very good. These estimates are based on the experience of the ELSA pollen records (Sirocko et al., 2016). The estimated errors for pollen and dust include both timing errors and measurement inaccuracies of the proxies and represent the estimated maximum errors. Counting errors for pollen were considered very small, as the original investigators are very experienced. In addition, the sample rate of at least one sample every 250 years is high enough to smooth out minor errors. In our experience, the measurement inaccuracies of the devices are around 2%. We have therefore taken this value as a minimum measure for the dust error values. Furthermore, there are possible age uncertainties, which become more important for records with smaller sample intervals. Speleothem age errors were given in the original data sources. All speleothem age uncertainties in the speleothem growth data we used for this synthesis were less than 4%.

To calculate a total maximum error, we randomly disturbed the original data with a percentage error given by the error estimates (Tab 2). From the perturbed data we calculated a perturbed aridity index as described in Table 1 and Chapter 2.1. The variance over 100 000 runs indicates the approximate error of our aridity index (for the script, see S10). This error simulation is based on the method of Koehler et al (2009) and personal communication with M. Mudelsee.

The generated error estimations are displayed in Fig. 2 and Figs. S1-S9 by grey colour shades behind the mean data. The actual reconstructed aridity index values are integer ones (see Supplementary table STab 2) but are displayed with a 200-year running average to better illustrate the basic structures. Smoothed aridity index values below 1.5 account for arid conditions, values between 1.5 and 3.5 show intermediate aridity and values larger then 3.5 show more humid conditions (see Fig. 3 and Fig. 4).

| Regions | Speleothem age error [%] | Tree pollen error [%] | Eolian dust error [%] |
|---|---|---|---|
| Central Europe | 2.66 | 3 | 5 |
| Arabian Sea | 1.5 | -- | 3 |
| China | 2 | 2 | 2 |
| NW-Africa | 1 | 10 | 2 |
| Southern Europe | 1 | 4 | 2 |
| Portuguese Margin | 1 | 3 | 2 |
| Mediterranean Sea | 2.5 | 2 | 3 |
| St. Barbara Basin | 1 | 3 | -- |

**Table 2. Error estimations as input to simulation for all key regions for speleothems, tree pollen and eolian dust.**

## 2.3 Model description

We employ the General Circulation Model COSMOS (community of earth system models) which was developed at the Max-
Planck Institute for Meteorology in Hamburg (Jungclaus et al., 2006). COSMOS comprises the standardized IPCC4 model
configuration which incorporates the ocean-sea ice model MPIOM (Marsland et al., 2003), the ECHAM5 atmosphere model
at T31 spherical resolution (~3.75 × 3.75°) with 19 vertical levels (Roeckner et al., 2003) and the land surface model JSBACH
including vegetation dynamics (Brovkin et al., 2009). The ocean model is resolved at 40 unevenly spaced vertical layers and
takes advantage of a curve-linear grid at an average resolution of 3 × 1.8° on the horizontal dimension, which increases towards
the grid poles at Greenland and Antarctica (~30 km). High-resolution in the realm of the grid poles advances the representation
of detailed physical processes at locations of deep-water formation, as Weddell, Labrador and Greenland and Norwegian Seas.
The ocean model includes a dynamic-thermodynamic sea-ice model (Hibler, 1979). Net precipitated water over land, which is
not stored as snow, intercepted water or soil water, is either interpreted as surface runoff or groundwater and is redirected
towards the ocean via a high-resolution river routing scheme (Hagemann and Dümenil, 1997).

Our COSMOS version (COSMOS-landveg r2413, Year 2009) has no flux correction and has been successfully applied to test
a variety of paleoclimate hypotheses, ranging from the Cretaceous (Niezgodzki et al., 2017), Miocene climate (Knorr and
Lohmann, 2014; Stärz et al., 2017), the Pliocene (Stepanek and Lohmann, 2012), glacial (Gong et al., 2013, 2015; Zhang et
al., 2013, 2014) and interglacial climates (Lohmann et al., 2013; Pfeiffer and Lohmann, 2016; Wei and Lohmann, 2012) as
well as future climates (Gierz et al., 2015; Lohmann et al., 2008).

Here, we present results obtained from model setups encompassing the Pre-industrial (PI), LGM and late MIS3 (32 000 yr
b2k) climate conditions. Details of each experiment set-up have been documented in Wei and Lohmann, 2012 (for PI run),
Zhang et al., 2013 (for LGM run) and Gong et al., 2013 (for 32 000 yr b2k run), with modified sea level, ice sheets, greenhouse
gas concentrations and Astronomical parameters for their conditions in the past, respectively. For the late MIS3 run, the model
mimics a GS due to freshwater hosing and GI with overshoot in temperature (Gong et al., 2013).

## 3 Results

The Central Europe region is our starting point for the comparison with the other key regions, the detailed data description of the other areas is given in the supplement S1-S7.

### 3.1 Central European climate for the last 60 000 years

The Atlantic sea surface temperature pattern strongly influence the whole European continent today (e.g. Cassou et al., 2005). Nowadays, the annual mean temperature in Germany is about 9.6 °C and precipitation of about 800 mm/year (Deutscher Wetterdienst, 2018). An established geoarchive to reconstruct the climate of central Europe are the volcanic maar lakes of the Eifel, which cover the Holocene with varves (annually laminated sediments), reach far back into the Pleistocene and cover the entire last 60 000 years continuously (Sirocko et al., 2016). These maar lakes of the Eifel in western Germany are today up to
70 m deep, with a large water volume and anoxic bottom water, favouring the, preservation of annual layers (Negendank, 1989; Negendank and Zolitschka, 1993; Zolitschka et al., 2000). We use the long records of the ELSA Project at Mainz University (Sirocko, 2016; Sirocko et al., 2005, 2013) as a starting point for our study. Holocene cores are varved or at least laminated, which leads to a good understanding of sedimentation processes (Sirocko et al., 2016). The LGM and stadial core sections were dominated by sedimentation from annual dust storms (Schaber and Sirocko, 2005), see Figure 2. Dietrich and
Seelos (2010), Dietrich and Sirocko (2011) and Seelos et al. (2009) calculated a dust index which reveals the GIs in detail. The closest to the Eifel and well-dated speleothems come from the Bunker Cave in the Sauerland (Fohlmeister et al., 2012; Weber et al., 2018) which can be compared to the Spannagel Cave system from western Zillertal, Austria (Fohlmeister et al., 2012; Holzkämper et al., 2004, 2005).

The timespan from 60 000 to 48 000 yr b2k (early MIS3, GIs 17-13) is characterized by high precipitation visible in the fast
speleothem growth of Bunker and Spannagel caves. Nearly 100 % of tree pollen combined with lowest grass and herb pollen values also indicate a wet climate during that time as well as relatively high temperatures close to present day ones (Sirocko et al., 2016). An intermediate dust content in the ELSA-Dust-Stack suggest an intermediate to low aridity. A similar pattern of low dust concentration is visible in the NGRIP ice core. The pollen composition change in the Eifel began around 49 000 yr b2k towards more grass and herbs pollen. A drastic change in $\delta^{18}O$ occurred at the beginning of GI12 occurred at
46 860 yr b2k (Rasmussen et al., 2014). With the beginning of Heinrich event 5 (H5), the dust amount spikes in the ELSA-Dust-Stack as well as in the NGRIP core indicating a strong pulse of aridification around 48 000 yr b2k, ending the humid phase of early MIS3.

The period from 48 000 until 38 500 years b2k includes GIs 12-9. The speleothem growth in Spannagel ended at 45 700 yr b2k. The Bunker Cave speleothems show a hiatus between 50 000 and 46 000 yr b2k and a short growth recovery,
with growth ending around 45 000 yr b2k. The tree pollen proportion decreased to about 50% to 60%, still more tree pollen than grass pollen, but a much smaller amount than in early MIS3. While the dust amount in the ELSA-Dust-Stack increases to

higher intermediate values, the pattern within the NGRIP is characterized by the stadial pulses. ELSA-Dust-Stack, NGRIP dust and NGRIP $\delta^{18}$O show the same pattern and seem to react to the same mechanism.

From 38 500 to 22 000 yr b2k (GIs 8-2) a change towards lower precipitation and higher aridity occurred. No speleothem growth is documented from Bunker or Spannagel cave. The pollen concentration shows higher grass and herbs content and
5 lower tree pollen percentages, but still some birch and pine trees were present during this time (Sirocko et al., 2016). The ELSA-Dust-Stack comprises of multiple changes within this timespan and shows the general dust content as relatively high with larger variability. The NGRIP in contrast shows the highest dust concentrations in the time between 23 000 and 26 000 yr b2k, a phase where the dust content in the ELSA-Dust-Stack is high, but not at maximum values. The NGRIP $\delta^{18}$O$_{ice}$ whereas shows a phase of cold temperatures (-45° to -50° C) during this time (Kindler et al., 2014).

The timespan of 22 000 to 14 300 yr b2k also does show no speleothem growth as well as a complete absence of all pollen. The precipitation was at the lowest values of the aridity index for that region, while the ELSA-Dust-Stack shows the highest dust amounts from 23 000 up to 15 000 yr b2k. The biggest difference between ELSA-Dust-Stack and NGRIP-Dust can be seen in the period of 26 000 to 23 000 yr b2k, where the dust content in the NGRIP core has two distinct maxima, while the dust content in the Central European record increases only slightly. The return of vegetation followed shortly after
14 300 yr b2k (Litt and Stebich, 1999). Younger Dryas (YD) is apparent in the pollen data by a grass pollen increase and the Holocene is marked by a sharp increase in tree pollen. Throughout the Holocene, tree pollen values range around 90 %. The dust content of the ELSA-Dust-Stack varies between intermediate to low levels. Speleothem growth in Spannagel and Bunker Cave started again at 12 408 yr b2k and continues through the whole Holocene. Also, the $\delta^{18}$O of the NGRIP shows constant high values with exception of the YD event. The time from 14 300 yr b2k up to present day can be described as a humid phase
with intermediate to high precipitation and moderate temperatures.

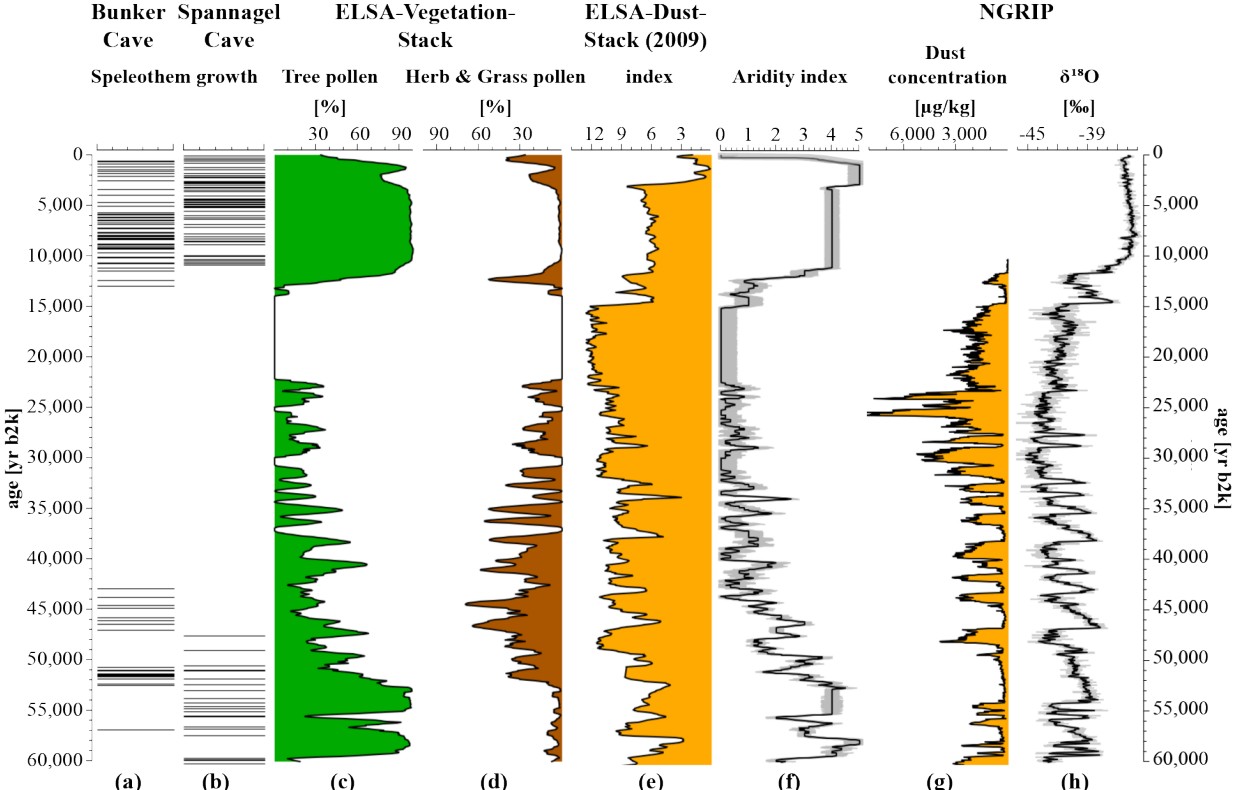

**Figure 2: Central European climate over the last 60 000 years**: **(a)** Bunker Cave (Fohlmeister et al., 2012, 2013; Weber et al., 2018) and **(b)** Spannagel Cave (Holzkämper et al., 2004, 2005; Spötl and Mangini, 2002) show speleothem growth phases, which require mobile water from frequent precipitation; **(c, d)** ELSA-Vegetation-Stack pollen data (Sirocko et al., 2016) are divided into tree- and herb & grass pollen. While trees require more precipitation, grasses are dominant for more arid conditions; **(e)** ELSA-Dust-Stack (Seelos et al., 2009) indicates more arid conditions with higher values, lower values account for more humid conditions. GIs are distinguishable by lower index values and are highly comparable to **(h)**; **(f)** Aridity index for Central Europe as result from **(a-e)**, for detailed information see method section; **(g)** Dust concentration from NGRIP ice core (Ruth et al., 2007); **(h)** $\delta^{18}O$ data from NGRIP ice core (North Greenland Ice Core Project Members et al., 2004) in comparison.

## 3.2 Aridity reconstruction of the last 60 000 years

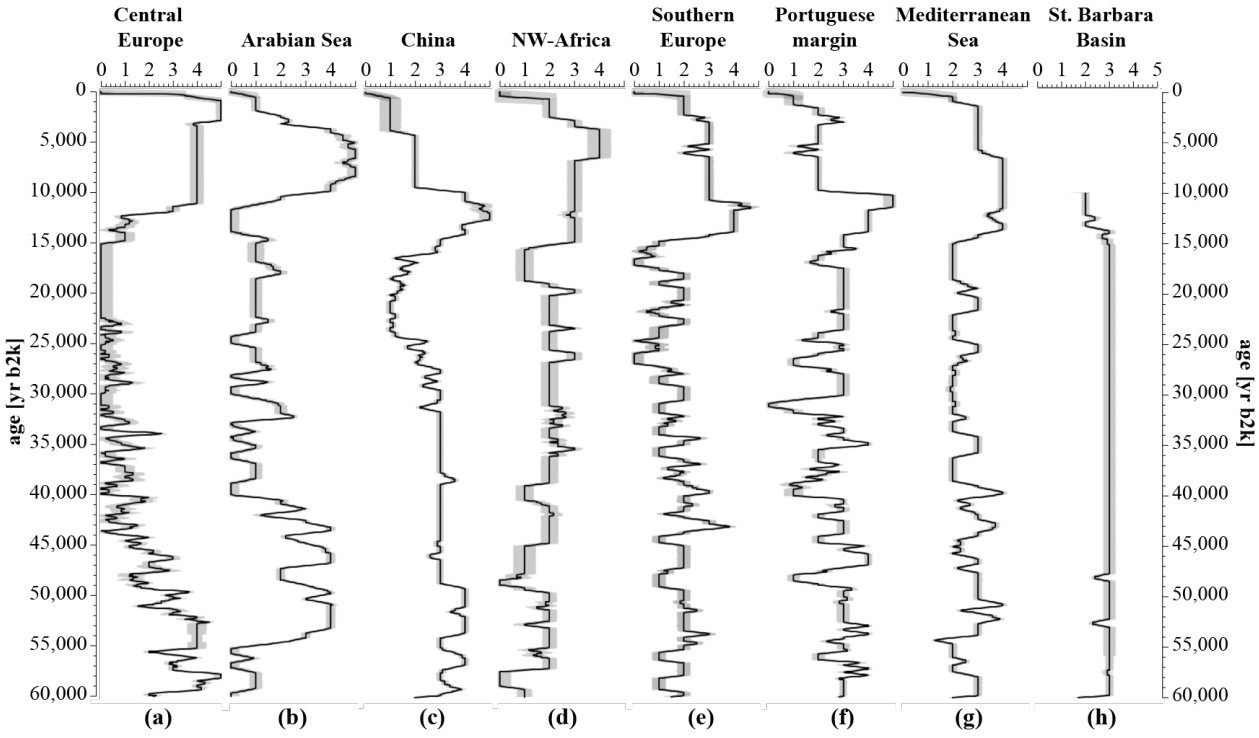

**Figure 3: Aridity indices for the key regions over the last 60 000 years**. Smaller values indicate more arid, higher values indicate more humid conditions. An early MIS3 wet phase and a Holocene or deglaciation wet phase on various timings for the regions are recognisable.

Figure 3 shows all aridity indices from the regional syntheses. A humid, early MIS3 can be identified for all regions adjacent to the St. Barbara Basin, but sometimes with a time lag of several millennia. We cannot be sure whether these shifts are caused by stratigraphy or whether they represent leads and lags in the climate system itself. Arid LGM conditions are identifiable for Central Europe, Arabian Sea, China NW-Africa, Southern Europe, Portuguese margin and Mediterranean Sea region. The last deglaciation is visible in all records by drastic changes around 14 700 yr b2k. A humid phase during deglaciation (14 700 –

11 700 yr b2k) or Holocene (11 700 – present) is also apparent for all regions, apart from St. Barbara Basin, where the proxies show an opposing signal to the other archives due to regional effects (Heusser, 1998; see S7 'St. Barbara Basin'). Fig. 3h is truncated, as no Holocene proxies are available for that region from these records (see S7).

Figure 4 is based on the aridity indices shown in Fig. 3 and additional information of the regional synthesis of China, (see Fig. 2 and S1-S7). Publicly available pollen data from Mingram et al. (2018) start at 10 150 yr b2k. Pollen reconstructions

from Stebich et al. (2015) give additional Holocene information on the Sihailongwan maar lake (see S3 'China'). Therefore, we have used the additional information following the construction of the aridity index to complete the interpretation of China shown in Figure 4. Blue bars indicate humid conditions, yellow bars indicate medium and red bars indicate arid conditions. Transitions or subdivisions between these conditions are indicated by overlapping bars. Figure 4 shows three large-scale

structures that connect all selected key areas. The Holocene is generally always relatively humid, but there are regional differences between the early and late Holocene. The period of the LGM is arid in all selected key regions, but the beginning of the arid phase varies between regions. These may also be related to stratigraphic inconsistencies or existing leads and lags in regional climate change. The early MIS3 is quite warm and humid in all regions. The signal is strongest in Central Europe and China from 60 000 to around 48 000 yr b2k, and in the Arabian Sea from 55 000 to 42 000 yr b2k (Fig. 4 and S1). This indicates teleconnections between the North Atlantic and the subtropical monsoons (Sirocko et al., 1993). North-west Africa and Southern Europe were slightly less humid in the early MIS3, but still shows enhanced precipitation compared to mid and late MIS3. The Portuguese margin region underwent larger changes compared to the other regions, as it was directly influenced by the North Atlantic. The Mediterranean Sea region also was more humid during mid and late MIS3 but not as humid as other regions, St. Barbara Basin shows humidity with same intensity. The early MIS3 was generally quite humid in the northern hemisphere. A pronounced aridification occurred with H5 (around 48 000 yr b2k), especially in NW-Africa, Portuguese margin region, Arabian Sea, Southern and Central Europe.

The period between 45 000 and 15 000 years b2k was less humid globally than the early MIS3 by 50 000 years b2k. During late MIS3, variations occur within regions, but between medium and more arid conditions. Heinrich events seem to have a strong influence, which is indicated by the increases in aridity during the Heinrich events. Towards the end of the LGM period all regions show arid or intermediate conditions, but Central Europe and the Arabian Sea were the most arid. The LGM is characterized by reduced precipitation, which is clearly visible by the red bars around 20 000 years b2k (Fig. 4).

The LGM period was followed by a global climate improvement, again with a different start time for each region. The earliest and most pronounced climate improvement took place in China, followed shortly after by the Mediterranean Sea, Central Europe, Southern Europe and the Portuguese margin regions. In the St. Barbara Basin, too, there were climate improvements even before the Holocene began. This general climate improvement is most likely associated with the warming of the North Atlantic by 14 700 yr b2k (Rasmussen et al., 2014). The early Holocene climate optimum occurred around 8 000 years b2k (Shakun and Carlson, 2010) with a temperature and precipitation maximum. This is particularly evident for Central Europe, the Arabian Sea, NW-Africa, the Mediterranean Sea and China.

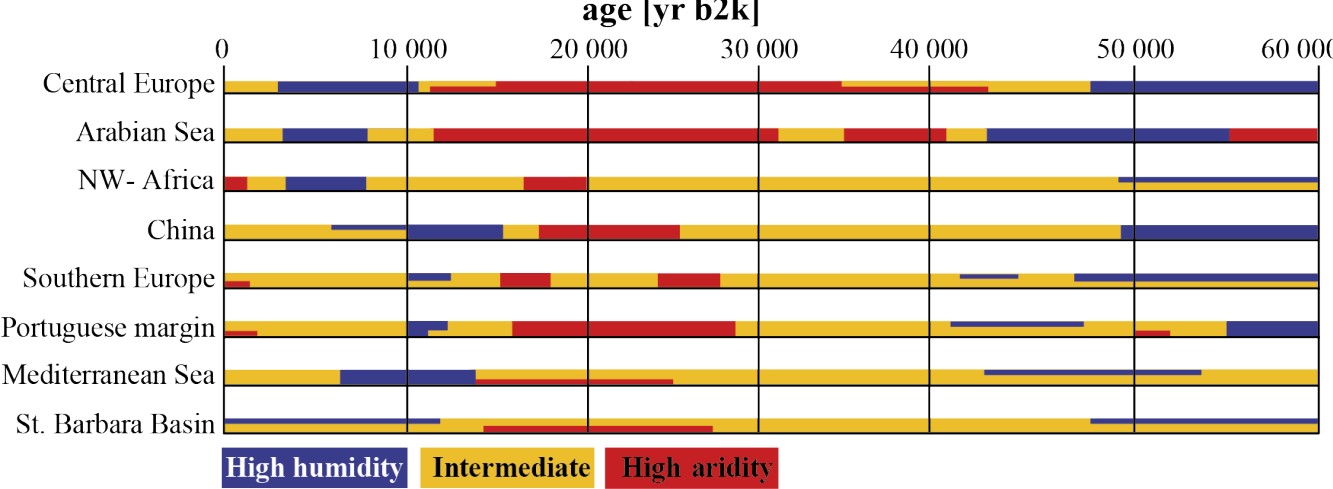

**Figure 4: Aridity synthesis for the selected key regions for the last 60 000 years**. Blue bars indicate high humidity, yellow bars intermediate humidity and red bars indicate high aridity. Overlapping half bars indicate transitions between both states. An early MIS3 wet phase and a Holocene or deglaciation wet phase with different starting times for the regions can be identified.

## 3.3 Global speleothem growth pattern

Figure 5 summarizes all speleothem growth phases mentioned in the regional syntheses. A consistent pattern shows the growth of all speleothems except NW-Africa during the early MIS3 phase. All other regions indicate fast growth rates and corresponding humid conditions at least during interstadials. A larger change occurs around H5 or, shortly afterwards, between 48 000 and 45 000 yr b2k. Speleothem growth stopped in Central Europe caves as well as in Arabian Sea region and drastically slowed down in Southern Europe, China and Mediterranean Sea region during late MIS3. The still fast growth for Portuguese margin region could be explained by a regional effect of enhanced moisture supply due to the position of the Buraca Gloriosa Cave. These speleothems show Heinrich events H4, H3, H1 as hiatus. Climatic conditions impair with progressing time. No growth is observed during LGM times for several regions including Central Europe, Arabian Sea and Southern Europe and very slow growth rates are observed for NW- Africa, China and Mediterranean region pointing out to more arid conditions during late MIS3 and LGM times compared to early MIS3 conditions. The effect of large-scale atmospheric teleconnections, also observed previously, can also be observed with the beginning of Bølling / Allerød (14 700 yr b2k) or after YD (11 703 yr b2k). Speleothem growth starts again for Central Europe, the Arabian Sea, Southern Europe. In China, the Mediterranean and the Santa Barbara Basin region, the growth rate is again increasing significantly. The available data show climatic amelioration after the LGM period globally but especially on the northern hemisphere (NH).

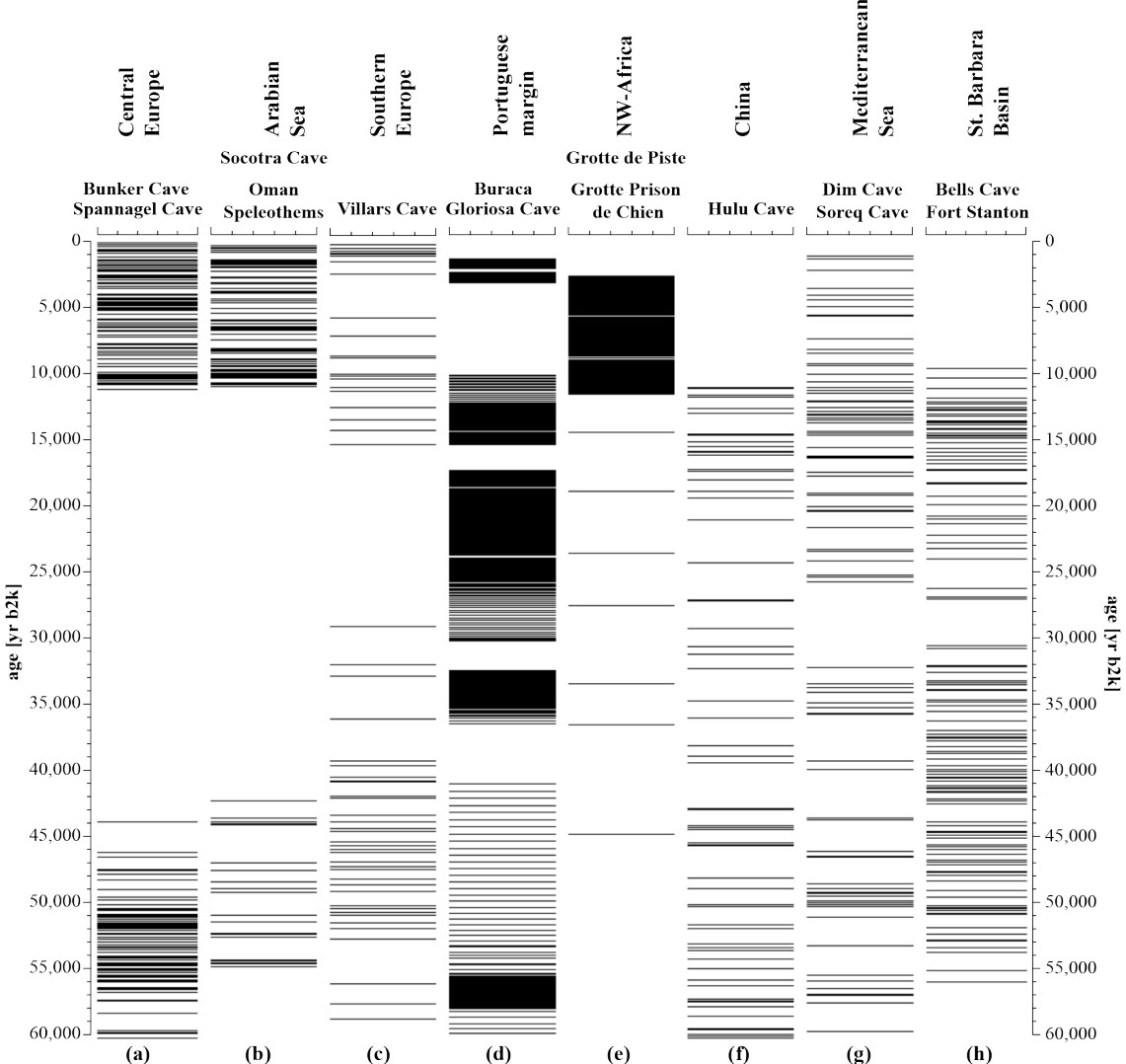

**Figure 5: Speleothem growth phases, which require mobile water from frequent precipitation for the selected key regions.**
Horizontal bars show age datings. **(a)** Central Europe with Bunker Cave and Spannagel Cave (Fohlmeister et al., 2012, 2013; Holzkämper et al., 2005; Spötl and Mangini, 2002; Weber et al., 2018); **(b)** Socotra Cave and Oman Caves from Arabian Sea region (Burns et al., 2003; Fleitmann et al., 2007); **(c)** Southern Europe represented by Villars Cave speleothems (Genty et al., 2003, 2006; Labuhn et al., 2015; Wainer et al., 2009, 2011); **(d)** Portuguese margin region with Buraca Gloriosa Cave (Denniston et al., 2018); **(e)** North-West Africa with Grotte Prison de Chien and Grotte de Piste (Wassenburg et al., 2012, 2016); **(f)** China with Hulu Cave (Liu et al., 2010; Wang et al., 2001); **(g)** Mediterranean Sea region with Dim Cave and Soreq Cave (Bar-Matthews et al., 2000; Ünal-İmer et al., 2015); **(h)** Bells Cave and Fort Stanton representing St. Barbara Basin (Asmerom et al., 2010; Wagner et al., 2010).

## 3.4 Global eolian dust pattern

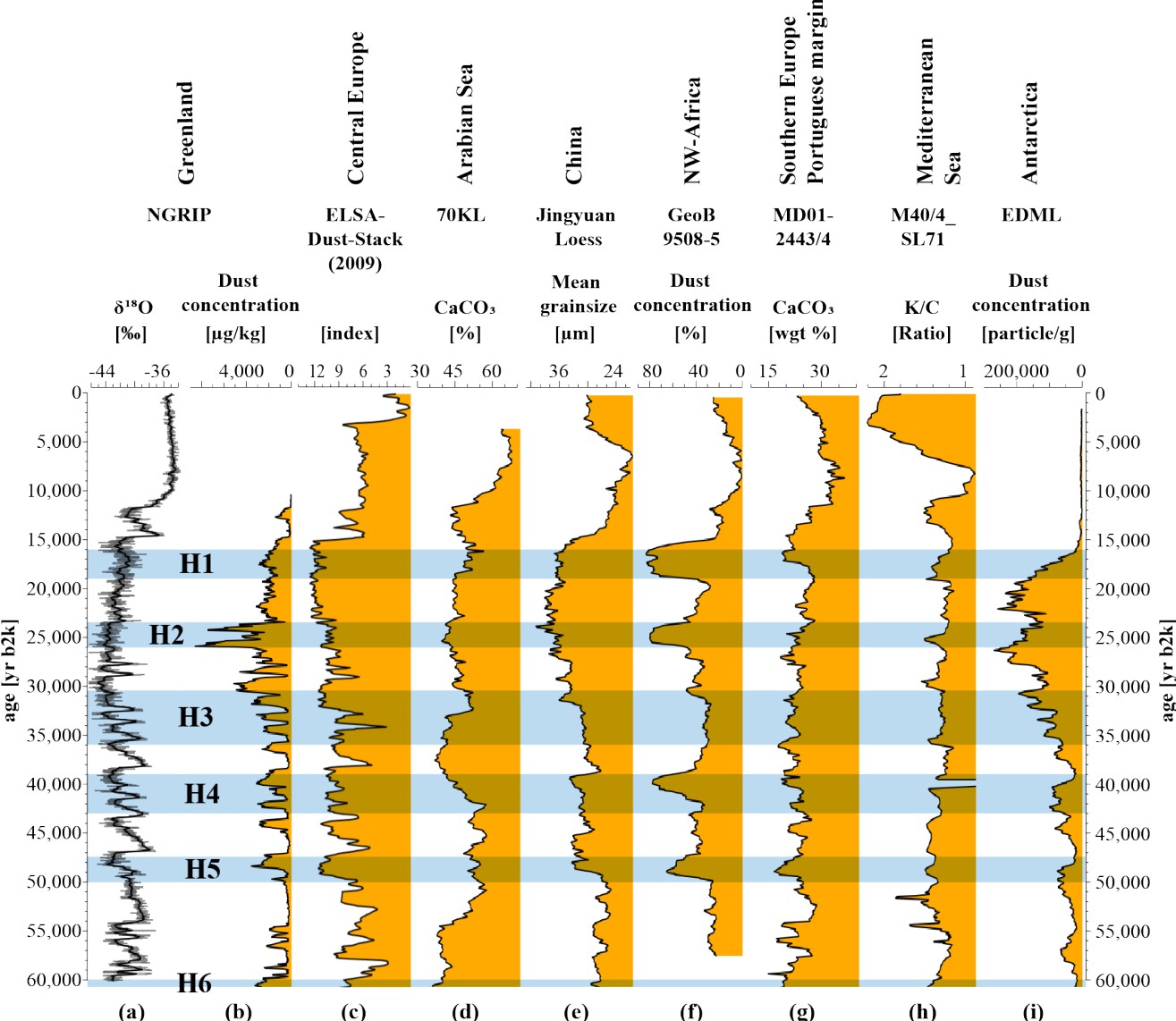

**Figure 6: Global eolian dust archives over the last 60 000 years**. More dust (left side of each column) indicates increased aridity, less dust indicates increased humidity. **(a)** NGRIP $\delta^{18}$O (North Greenland Ice Core Project Members et al., 2004) in comparison and **(b)** Dust concentration (Ruth et al., 2007); **(c)** ELSA-Dust-Stack (2009) for Central European dust (Seelos et al., 2009); **(d)** CaCO$_3$ accounting for dust from Arabian Sea marine core 70KL (Leuschner and Sirocko, 2003); **(e)** Mean Grainsize from Jingyuan Chinese Loess plateau (Sun et al., 2010); **(f)** Dust content within marine core GeoB9508-5 off North-West Africa (Collins et al., 2013); **(g)** CaCO$_3$ for Southern Europe and Portuguese margin from marine core MD01-2443/4 (Hodell et al., 2013) **(h)** K/C ratio from marine core M40/4_SL71 for Mediterranean Sea (Ehrmann et al., 2017); **(i)** Antarctica ice core EDML Dust concentration (Wegner et al., 2015) in comparison. Light blue bars highlight Heinrich events after (Naafs et al., 2013).

Similar patterns are visible in the dust archives used for this synthesis. For the early MIS3, low dust values for Central Europe, China, NW-Africa and medium dust values for all other regions except the Arabian Sea are visible. In general, the dust records also show the flickering of the MIS3 climate.

Apart from the regions' own patterns, some background structures can be seen. Heinrich events are most pronounced for NW-Africa and Southern Europe / Portuguese margin, as these regions belong directly to the North Atlantic. Also the Mediterranean, Central Europe and China show Heinrich events within the dust records. China, the Arabian Sea, NW-Africa and Central Europe show a pronounced dust maximum during the LGM period. Apart from the Holocene, the lowest dust values in the early MIS3 show for Central Europe, Arabian Sea, China, NW-Africa and Mediterranean - but at different times. The NGRIP dust concentration and the ELSA dust concentration (Seelos et al., 2009) are in good agreement for early and intermediate MIS3. Tipping points show stepwise changes in both archives at several points in time (~49 000, ~36 500, ~23 000, ~14 700 yr b2k), and are described in Sirocko et al. (2016) as Landscape Evolution Zones. The good agreement of both regions indicates a close connection with the North Atlantic climate variations. The sediments of the Arabian Sea are also closely linked to the North Atlantic variability, as known since Schultz et al. (1998) and visible in Fig. 6. Sun et al. (2010) also showed a correlation of Chinese loess with North Atlantic climate fluctuations. High dust contents during mean MIS3 and LGM are visible in most archives, often in connection with dustier Heinrich events or stadials in general. Increasing dust levels around 30 000 years b2k up to LGM show an almost global distribution, but regional differences can be observed: ELSA-Dust-Stack and NGRIP differ in the timing of the maximum dust values This could be explained by a process similar to the blocking of western winds by ice sheet growth described in Schenk et al. (2018) and Schiemann et al. (2017).

Between 30 000 and 17 000 years b2k, conditions were mostly arid, recognizable by the dust maxima in Greenland, Central Europe, China and the Portuguese margin during the LGM. With the YD and the onset of the Holocene, dust values decrease until the end of the Altithermal period. After that, dust increases in the Arabian Sea, China, NW Africa, the Portuguese margin and the Mediterranean Sea, indicating increasing aridification of the large desert areas in Africa, the Arabian Peninsula and China. During the last 60 000 years of climate evolution, major changes occurred simultaneously on the globe, indicating atmospheric teleconnections (Bjerknes, 1969; Markle et al., 2017; Sirocko, 2003; Sirocko et al., 1993, 1996; Zhou et al., 1999).

## 4 Discussion

The PalMod requirements limit the calculation of the aridity index partially. In particular, not all high-resolution archives could be taken into account in the selection of the regions, since too few further data from other archives in the proximity are publicly accessible. The requirements strongly influence the region selection, as regions with only one record type were excluded. Records that would also fall within the selected regions are e.g. Nussloch loess sequence as dust record for Central Europe / Southern Europe or the pollen record of Tenaghi Philippon for Mediterranean Sea region.

Since we use only one dust or pollen record per region to calculate the aridity index, these records would replace the existing ones. The ELSA-Dust-Stack (2009) and Nussloch are very similar with respect to the processes that formed these records.

Therefore their climatic patterns are very similar. The aridity index would therefore not be completely different from the version now generated with publicly available data. The situation would be similar for Tenaghi Philippon and Lago Grande di Monticchio. The archives show similar patterns and were subject to similar processes. These aridity indices would therefore also be very similar. To utilize more than one data set for pollen or dust from different archives would require a much more

complex method to compare the different data.

The ELSA dust stack by Seelos et al. (2009) see Fig. 2e, shows all Greenland Interstadials of the last 60 000 years. The resolution of the ELSA-Dust-Stack (2009) is very high and the data are publicly available. For Nussloch, among other things, grain size index data have been collected, which show the GIs. These data from Nussloch (e.g. Antoine et al., 2009; Rousseau et al., 2007) can basically be compared with the ELSA-Dust-Stack (2009), as both archives show paleo dust. Although the

data are not accessible, the figures of the original papers can be used for comparison. The disadvantage of a loess archives compared to terrestrial archives is an often discontinuous sedimentation although we are aware that some loess archives are often continuous, e.g. Marković et al. (2011, 2015), Obreht et al. (2019), Yang and Ding (2014). Nevertheless, it is clear that both archives are well suited to study paleoclimate developments. Both archives can be used for aridity reconstructions due to their paleo dust sedimentation.

The following chapter 4.1 compares our aridity index with a previous reconstruction from Herzschuh (2006). This reconstruction is based on various pollen records, but only with that one kind of proxy, which is pollen. Both methods reveal similar results, although Herzschuh has used a lot more records. Chapter 4.2 compares the aridity index with different climate simulations. Since the overall agreement is quite good as well, we consider our aridity reconstructions to be representative.

**4.1 Comparison of aridity index and previous aridity reconstruction**

Aridity reconstructions of Herzschuh (2006) for the last 50 000 years for Central Asia are in excellent agreement to the aridity reconstructions of this synthesis of the China region (S3). Herzschuh used only pollen data for the reconstruction, while our aridity index is based on three different proxies to refine the image. On the other hand, Herzschuh used several pollen records of one region to calculate her aridity index on a broader basis. Our aridity index consists of longer records and reaches until the beginning of MIS3 (60 000 yr b2k). It shows a humid early MIS3 and a decrease in humidity around 50 000 yr b2k to

intermediate conditions. Moderate dry conditions are reconstructed for 50 000 to 45 000 yr b2k from Herzschuh, similar to the aridity index, followed by an increase in humidity until 40 000 yr b2k. Minor differences between 45 000 and 30 000 yr b2k are visible, but the general trend shows broad consensus between both reconstructions for this period. The middle to late MIS3 are relatively humid in both reconstructions. In both reconstructions the aridity increases rapidly towards the LGM, with a strong LGM aridity maximum of 21 000 to 18 000 yr b2k. A gradual climate improvement after LGM is clearly evident in

both reconstructions, with a first increase in humidity up to 13 000 yr b2k with subsequent optimal climate conditions during the early Holocene (11 000 - 7 000 years b2k). The aridity index lacks pollen values in the Holocene data (see S3), but the agreement of Herzschuh compared to the published Holocene pollen data of Stebich et al. (2015) is obvious and shows an early Holocene climate optimum up to 4 000 years b2k in both reconstructions. A wet early Holocene can also be observed in

central Chinese speleothem growth rates from the 'Sanbao Cave', which are highest from 9 500 to 6 500 years b2k (Dong et al., 2010).

The selection of records is certainly limited, but representative within the regions. We are not aware of any records that completely contradict the reconstructed climatic conditions. Since the selected regions are relatively homogeneous in terms of climate, we expect similar results within the regions.

## 4.2 Comparison of proxy synthesis with model results

The above records describe the structure of past aridity globally, but do not show the mechanisms of past, present and future climate changes, which can be simulated by global climate models (GCM). Compared to our reconstructed aridity indices, we use the coupled climate model COSMOS. Figure 7 shows the precipitation changes and anomalies from the late MIS3 time slice with respect to pre-industrial (PI) and LGM conditions. Panel A and C show stadial conditions and panel B and D interstadial conditions. Panels A and B show the time slice 32 000 years b2k compared to pre-industrial time, while panels C and D show LGM conditions.

Model runs for late MIS3 interstadial times show increased precipitation for Europe, the North Atlantic, the Arabian Sea and large parts of the equatorial Pacific, while the remaining parts of the equator in Asia, the Central Atlantic and the northern parts of South America show lower precipitation. The stadial MIS3 state generally shows the same spatial trends, but with a generally higher aridity. Model and drought reconstructions are consistent under almost identical conditions for PI and MIS3. Barron and Pollard (2002) simulated a time slice of 42 000 years b2k for European precipitation. The results are comparable with our aridity index, but Central Europe, Southern Europe and the Portuguese margin were wetter in our reconstruction than in the Barron and Pollard simulation. In contrast, these results show that the 42 000 year b2k time slice was more humid than the 32 000 year b2k time slice for Europe. Our reconstruction assumes that precipitation during early and middle MIS3 was even higher than during late MIS3 and as high as during the early Holocene optimum. This statement can be made for all regions if the different starting times of the wet periods are included. Late MIS3 and especially comparisons between early and late MIS3 have not been investigated in current models so far.

Table 3 summarizes the results from the model simulations (transferred to symbols) from Fig. 7. This model structure is divided into GI and GS state and is compared for each of the key regions with the results for the drought reconstructions from Fig. 4. Relatively greater aridity in terms of PI or LGM for each region is represented by a minus (-), approximately equal conditions by a circle (o) and more humid conditions by a small plus (+). Additionally, the agreement of each simulation and the aridity reconstructions of this work are shown. A bad agreement is shown in red color, a medium one with yellow, a light green for good and dark green for very good agreement. The overall consistency of model and reconstruction is good. Most precipitation changes of the drought index are also captured in the simulation.

The aridity reconstructions show that Central Europe was humid during the early MIS3, followed by a medium to very arid period until the end of the LGM.

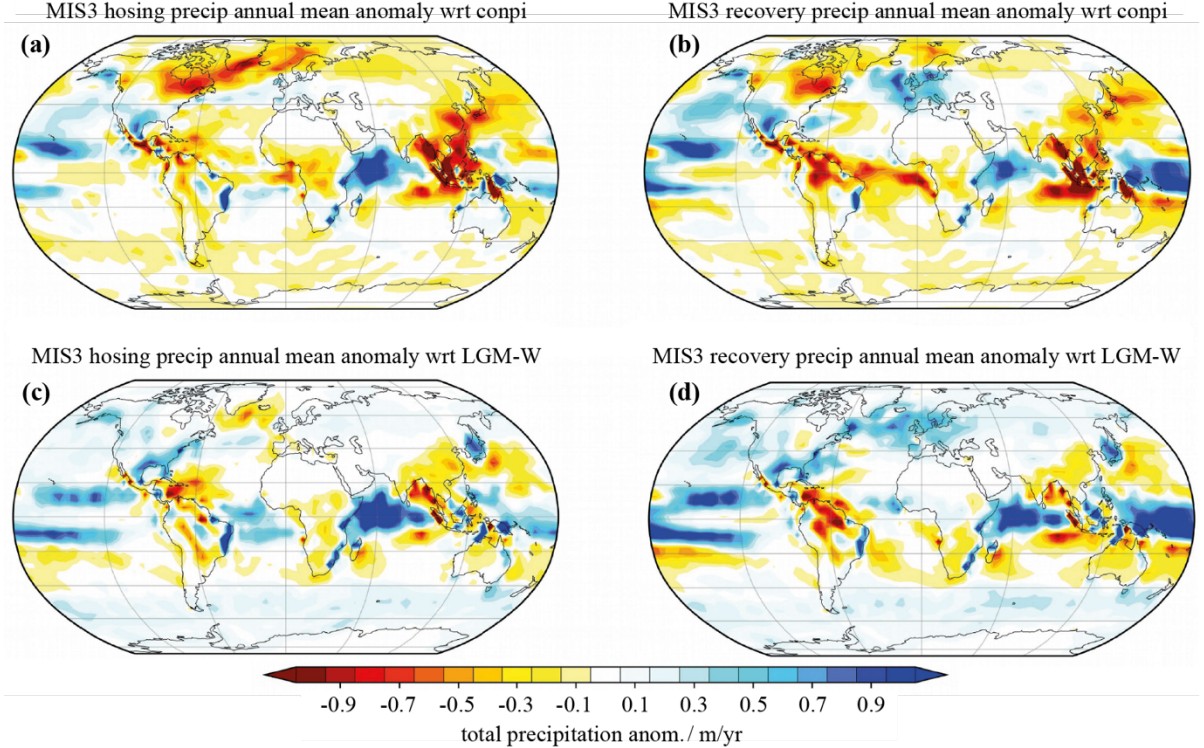

MIS3 hosing precip annual mean anomaly wrt conpi

MIS3 recovery precip annual mean anomaly wrt conpi

MIS3 hosing precip annual mean anomaly wrt LGM-W

MIS3 recovery precip annual mean anomaly wrt LGM-W

-0.9  -0.7  -0.5  -0.3  -0.1   0.1   0.3   0.5   0.7   0.9
total precipitation anom. / m/yr

**Figure 7: Simulated total precipitation anomalies for the late MIS3 climate relative to pre-industrial times (a, b) or relative to LGM (c, d).** Panel **(a, c)** show stadial conditions (hosing) and panels **(b, d)** show interstadial conditions (recovery). Simulation is compiled of various model simulations with COSMOS for pre-industrial (Wei and Lohmann, 2012), LGM (Zhang et al., 2013) and Late MIS3 (32 000 yr b2k) simulations (Gong et al., 2013).

|  | MIS 3 wrt. PI | | MIS 3 wrt. LGM | |
|---|---|---|---|---|
|  | 32 000 [a] | | 32 000 [a] | |
|  | GS | GI | GS | GI |
| Central Europe | o | + | o | + |
| Southern Europe | + | + | o | + |
| Portuguese margin | o | + | o | o |
| Mediterranean Sea | o | o | o | o |
| NW Africa | o | o | o | o |
| China | - | - | - | o |
| Arabian Sea | + | + | + | o |
| St. Barbara basin | o | o | o | o |

**Table 3. Comparison of model simulation with reconstructed aridity index**. Relatively larger aridity with respect to PI or LGM for each region is shown with a minus (-), approximately the same conditions with an open circle (o) and more humid conditions with a small plus (+). A bad agreement between aridity index and simulation is shown in red color, a medium one with yellow, a light green for good and dark green for very good agreement.

## 5 Conclusions

The aridity synthesis for the selected regions of the world climate allows five main conclusions:

(1) All regions beside St. Barbara Basin, which were analyzed here, underwent a wet phase during both the early MIS3 and the early Holocene and deglaciation (Fig. 3 and Fig. 4). The timing of these wet phases varied considerably from region to region.

(2) There were atmospheric teleconnections from the North Atlantic (Greenland) across Central Europe, the Arabian Sea to China. The changes in these regions occurred at similar times and with similar intensity. The other regions also underwent these changes, but with temporal differences (Fig. 3 and Fig. 4).

(3) Eolian dust, tree pollen and speleothem growth phases show congruent climatic patterns for the different regions, which we attribute to overregional aridity changes (Fig. 5, Fig. 6 and Fig. S8).

(4) Timing and regional extend of the aridity index are consistent with the model based precipitation simulations (Tab. 3).

(5) The quality of the aridity index is mainly limited by the original stratigraphy and sample resolution. Therefore, it is only a useful tool to observe general structures of the climate system.

## Author contributions

FF generated the aridity index, processed the data and prepared the manuscript. GL and XG developed and performed the climate simulation. BD developed the *ELSA*interactive++ program for data visualisation and analysis. FS stimulated the discussion on the global teleconnections within the climate system.

## Code / Data availability

Code and data are available in the supplement.

## Competing interests

We declare no competing interests.

## Acknowledgments

We would like to thank all four anonymous referees which have greatly improved the manuscript. This work was supported by German Federal Ministry of Education and Research (BMBF) as Research for Sustainability initiative (FONA); www.fona.de through PalMod project (FKZ: 01LP1510B). The authors thank Manfred Mudelsee for input on the error

estimations, Hannes Knapp, Philip Süßer and Jennifer Klose for data mining, Michael Weber, Johannes Albert, Sarah Britzius for discussion contributions to the manuscript.

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
