# Peer review of "Aridity synthesis for 8 selected key regions of the global climate system during the last 60 000 years"

_Climate of the Past, 2019_

## Referee Comment (RC1) · Anonymous Referee #1 · 24 Oct 2019

The manuscript describes an effort to generalize aridity patterns for the last 60 Kyears from a dataset comprising a selection of paleoclimate records, including pollen assemblages, speleothems, and a variety of dust proxies, organized in 10 regions. An aridity index is calculated for each region based on those paleorecords, and is used as a target for comparison with climate model simulations. The motivation of the work described here is relevant, the aim ambitious, and the devised general strategy very interesting. However, the presentation quality is not adequate; in particular the methodology is not described with sufficient detail, so that it is difficult to make an informed assessment on the robustness of the approach and the results. Therefore I recommend a substantial revision of the manuscript.

General comments

[Figure]

At this stage several passages in the text appear confusing, because of the use of the language, and sometimes contradictive. I provide some specific examples below, but I recommend carefully reviewing the entire manuscript in the spirit of addressing this comment.

The methods section is not satisfactory as it is now, since it resembles a short collection of sparse statements. It needs to be much more precise in detailing the different kinds of proxies used, and should be organized in a more organic way. It should also explain clearly what is the general strategy and what are the common rules used to (a) select and (b) treat the data and (c) the uncertainties. This is not discussed even in the supplement. Several datasets with potential relevance to this work are not even mentioned. The whole section should be substantially revised.

In addition, I think that the scope of the work should be clarified. When I read "synthesis" I would expect a complete data collection and selection by means of transparent filters, before aggregating the results. If on the other hand the strategy is to pick specific records, which are deemed representative of specific regions, then I think that (a) a discussion is needed on why these particular records were selected, and what is the inherent uncertainty in the choice, and (b) the main title and scope should reflect more faithfully this approach.

Concerning dust records, several proxies are used. While this may not pose a problem per se in the context of this study, a discussion is missing on other processes, in addition to "aridity", that could potentially affect the signal (changes in sedimentation rates controlled by productivity in the oceans and precipitation ice cores, etc.). What are the uncertainties related to the choice of specific proxies? In addition, any connection between sources of dust and specific paleodust records seems to have been disregarded, casting a doubt on the validity of certain regional interpretations (e.g. sources of dust to Greenland, EDML, Mediterranean Sea). These aspects should be thoroughly discussed.

Specific comments

1, 9 > "all regions show" it would be more appropriate to say "all the regions analyzed in this study show"

1, 10 > not always WITH the same timing

1, 11 > Perhaps what you mean is "Such discrepancies have been interpreted as regional effects, although stratigraphic uncertainties may affect some of the proposed interpretations"? Please clarify

1, 14 > "both lines of evidence show great agreement": which lines of evidence? Agreement of what with what?

1, 16 > FOCI

1, 20-21 > This sentence is awkward, please rephrase, e.g. Geological archives have the potential to provide information on the past states of climate variables at the global and regional level, and their evolution in time.

1, 26-27 > what do you mean by "ice sheets . . . are apparently also teleconnected with global sea level"? Please rephrase

2, 1-4 > How did you screen ∼2000 papers? Did you use some search algorithm and keywords?

2, 5-6 > Have you considered paleolake levels as a potential proxy as well?

2, 7 > "Arid" rather than "desert"

3, 12 > "The synthesis" rather than "The comparison"?

3, 12 > In this section you should explain in a very transparent way which are the rules for selecting specific records. And why specific one(s) are used to calculate the aridity index, rather than others (within a given region, e.g. Bunker vs Spannagel Cave in Figure 2). In addition, what are the rules to determine the time step of the aridity index,

given that the 3 records typically have different time axes?

3, 14 > What do you mean by "we use the original stratigraphy"? Aren't you saying that you port the chronologies to the GICC05 time scale? In addition, there is no mention as to how this operation was carried out: did you use some software?

3, 19-20 > What do you mean by "the errors . . . below 4% in total"?

3, 27-29 > Please rephrase this sentence

4, 1 > What is the global climate structure?

4, 4 > "For THE Northern Hemisphere . . ." and so forth, please review the use of the language throughout the manuscript

4, 8 > Larger than what?

4, 16 > Do you mean precipitation proxies?

4, 19 > "divided in three parts" is not clear at all. I guess what you are trying to say is that you assign each point in the pollen / dust time series to a category from 0 to 2, based on the current value of the rescaled record as a percentage with respect to the top value (which corresponds to 100%)? Is that correct? However, it is not clear what are those original values. One can only try to guess it is maybe the percentage of tree pollen is the whole pollen assemblage for a given point? What is it for dust? It could be many things since you indicated several different proxies for dust. In fact by looking at the supplement it seems it depends on each different proxy. Also, you do not spell out how you calculate the aridity index, one can grasp from the caption of table 1 that is the sum of the three "scores" for speleothem, tree pollen, and dust. Your procedure and the rationale behind it should be explained in detail and clearly in the methods section.

5, 2-9 > This section is also very confusing, it should be profoundly revised. First, you should probably mention that there are uncertainties on the age of the samples, and uncertainties on the specific values of the variables, in addition to their uncertainty as

proxies for a particular system. Second, you should clarify which are the cases where you have an uncertainty estimate from the original study and what it refers to. Then you can talk about the case where you have to assign the uncertainty arbitrarily to each sample in your time series, and you should specify on what grounds you assign a particular values (it could be the reference to a paper using the same kind of proxy, for instance). Fourth, as a key to read Table 2, you should describe explicitly if you have only one record for each kind of proxy for each region, or else how you dealt with multiple records. Finally, it may be more interesting to use other records than the "chosen" one, where available, to calculate the aridity index, as a metric for uncertainty / intra-regional variability.

5, 14 > I am not sure what you mean by "one of the large feedback regions": please rephrase

5, 19-20 > VARVE not warve

6, 4-5 > What does the dust concentration in the NGRIP ice core have to do with aridity in central Europe? I don't think it is appropriate to make such a statement without further discussion. As you know, there are several hypotheses concerning the interpretation of the Greenland dust records (e.g.(Mayewski et al., 2014; Steffensen et al., 2008)), and the major source of dust to the Greenland ice sheets are not uniquely attributed to Europe, to say the least (e.g.(Bory et al., 2003; Rousseau et al., 2014; Svensson et al., 2000; Újvári et al., 2015)). In view of these aspects, please state explicitly what is the link in your line of reasoning (e.g. generalized aridity in the northern hemisphere, in Eurasia, . . .), and what are the assumptions you make (e.g. Europe is major dust source to Greenland?), justifying them with adequate references to the literature.

6, 22 > Tree pollen?

6, 23 > "precipitation was at the lowest values of the whole record": which record are you referring to? To speleothem records? The aridity index?

6, 29 > Which speleothem?

S1-S9 > In these sections of the supplement I would expect to find more specific considerations on the selections of records (e.g. why data from (Pourmand et al., 2004) are not included in S1? Or (Skonieczny et al., 2019) in S2? Or the loess records in the discussion about central Europe? What's the link between EDML dust and Oceania? Etc….), before discussing those that are selected. Also, I did not find the details of how data are aggregated into the aridity index (e.g. why sometimes 4 records are considered, sometimes 2?). As mentioned already, the general rules for data selection should be spelled out clearly in the methods sections, and specific choices of notable datasets not included should be discussed in the supplement.

8, 6 > RELATED information

8, 11 > The proxies show an opposing signal ?

8, 13 > It is not clear how Figure 4 was produced. What is the role of the "additional information"? Are those the thin overlapping bars?

9, 11-12 > Please rephrase

9, 15 > "impair"?

9, 21 > How do you define a climate improvement? Please avoid terms like improvement and amelioration, impair?; expressions describing the changing state of the discussed variable should be preferred, such as drier, wetter, colder, etc.

10, 8 > representative of the Cariaco Basin

11, 4 > WHEN both hemispheres

14, 6-9 > Not clear, please rephrase

14, 24 > SPATIAL trends 16, 12 > All regions analyzed here

References

Bory, A. J.-M., Biscaye, P. E. and Grousset, F. E.: Two distinct seasonal Asian source regions for mineral dust deposited in Greenland (NorthGRIP), Geophys. Res. Lett., 30(4), doi:10.1029/2002GL016446, 2003.

Mayewski, P. A., Sneed, S. B., Birkel, S. D., Kurbatov, A. V. and Maasch, K. A.: Holocene warming marked by abrupt onset of longer summers and reduced storm frequency around Greenland, J. Quat. Sci., 29(1), 99–104, doi:10.1002/jqs.2684, 2014.

Pourmand, A., Marcantonio, F. and Schulz, H.: Variations in productivity and eolian fluxes in the northeastern Arabian Sea during the past 110 ka, Earth Planet. Sci. Lett., 221(1–4), 39–54, doi:10.1016/S0012-821X(04)00109-8, 2004.

Rousseau, D.-D., Chauvel, C., Sima, A., Hatté, C., Lagroix, F., Antoine, P., Balkanski, Y., Fuchs, M., Mellett, C., Kageyama, M., Ramstein, G. and Lang, A.: European glacial dust deposits: Geochemical constraints on atmospheric dust cycle modeling: European Glacial Dust Deposits, Geophys. Res. Lett., 41(21), 7666–7674, doi:10.1002/2014GL061382, 2014.

Skonieczny, C., McGee, D., Winckler, G., Bory, A., Bradtmiller, L. I., Kinsley, C. W., Polissar, P. J., De Pol-Holz, R., Rossignol, L. and Malaizé, B.: Monsoon-driven Saharan dust variability over the past 240,000 years, Sci. Adv., 5(1), eaav1887, doi:10.1126/sciadv.aav1887, 2019.

Steffensen, J. P., Andersen, K. K., Bigler, M., Clausen, H. B., Dahl-Jensen, D., Fischer, H., Goto-Azuma, K., Hansson, M., Johnsen, S. J., Jouzel, J., Masson-Delmotte, V., Popp, T., Rasmussen, S. O., Rothlisberger, R., Ruth, U., Stauffer, B., Siggaard-Andersen, M.-L., Sveinbjornsdottir, A. E., Svensson, A. and White, J. W. C.: High-Resolution Greenland Ice Core Data Show Abrupt Climate Change Happens in Few Years, Science, 321(5889), 680–684, doi:10.1126/science.1157707, 2008.

Svensson, A., Biscaye, P. E. and Grousset, F. E.: Characterization of late glacial continental dust in the Greenland Ice Core Project ice core, J. Geophys. Res. Atmospheres,

105(D4), 4637–4656, doi:10.1029/1999JD901093, 2000.

Újvári, G., Stevens, T., Svensson, A., Klötzli, U. S., Manning, C., Németh, T., Kovács, J., Sweeney, M. R., Gocke, M., Wiesenberg, G. L. B., Markovic, S. B. and Zech, M.: Two possible source regions for central Greenland last glacial dust, Geophys. Res. Lett., 42(23), 10,399-10,408, doi:10.1002/2015GL066153, 2015.

––––––––––––––––––––––––––––––

---

## Author Comment (AC1) · 7 Nov 2019

The manuscript describes an effort to generalize aridity patterns for the last 60 Kyears from a dataset comprising a selection of paleoclimate records, including pollen assemblages, speleothems, and a variety of dust proxies, organized in 10 regions. An aridity index is calculated for each region based on those paleorecords, and is used as a target for comparison with climate model simulations. The motivation of the work described here is relevant, the aim ambitious, and the devised general strategy very interesting. However, the presentation quality is not adequate; in particular the methodology is not described with sufficient detail, so that it is difficult to make an informed assessment on the robustness of the approach and the results. Therefore I recommend a substantial revision of the manuscript.

**General comments :**

We acknowledge the anonymous referee#1 for his review and the constructive and helpful comments. We revised it in detail as described. We still wait for the second review before we start fundamental rearrangements. Our point-by-point response to referee#1's comments is provided in red in the attachment.

At this stage several passages in the text appear confusing, because of the use of the language, and sometimes contradictive. I provide some specific examples below, but I recommend carefully reviewing the entire manuscript in the spirit of addressing this comment.

The methods section is not satisfactory as it is now, since it resembles a short collection of sparse statements. It needs to be much more precise in detailing the different kinds of proxies used, and should be organized in a more organic way. It should also explain clearly what is the general strategy and what are the common rules used to (a) select and (b) treat the data and (c) the uncertainties. This is not discussed even in the supplement. Several datasets with potential relevance to this work are not even mentioned. The whole section should be substantially revised.

→ We will upload the revised method section as soon as possible. The Methods and data treatment will be described in detail, but we would like to answer your specific comments right now, see below.

In addition, I think that the scope of the work should be clarified. When I read "synthesis" I would expect a complete data collection and selection by means of transparent filters, before aggregating the results. If on the other hand the strategy is to pick specific records, which are deemed representative of specific regions, then I think that (a) a discussion is needed on why these particular records were selected, and what is the inherent uncertainty in the choice, and (b) the main title and scope should reflect more faithfully this approach.

→ The special issue, which is the manuscript belonging to emerged out of the Palmod project. One prerequisite for PalMod is to work with publically available datasets only – from the most cited papers. See also our comment to p2,l 4-5. We went through "web of science" for every region and sorted the available papers by citation count. Afterwards we downloaded available data starting with the most

cited paper. Well cited papers - where data were not available - were not included into the compilation. This was a prerequisite from PalMod, that is why the paper is presented in the special issue.

Concerning dust records, several proxies are used. While this may not pose a problem per se in the context of this study, a discussion is missing on other processes, in addition to "aridity", that could potentially affect the signal (changes in sedimentation rates controlled by productivity in the oceans and precipitation ice cores, etc.). What are the uncertainties related to the choice of specific proxies? In addition, any connection between sources of dust and specific paleodust records seems to have been disregarded, casting a doubt on the validity of certain regional interpretations (e.g. sources of dust to Greenland, EDML, Mediterranean Sea). These aspects should be thoroughly discussed.
→ We used the same basic method through all regions in absence of real errors within the datasets. Therefore we had to construct a "proxy" for the uncertainty. We will address this in the method section.

**Specific comments :**

1, 9 > "all regions show" it would be more appropriate to say "all the regions analyzed in this study show" → I agree, changed in the manuscript

1, 10 > not always WITH the same timing → changed

1, 11 > Perhaps what you mean is "Such discrepancies have been interpreted as regional effects, although stratigraphic uncertainties may affect some of the proposed interpretations"? Please clarify → I agree, changed in the manuscript

1, 14 > "both lines of evidence show great agreement": which lines of evidence? Agreement of what with what? → rephrased to: Indeed, geological archives and GCMs show great agreement of aridity pattern for the Holocene, LGM and for the late MIS intervals.

1, 16 > FOCI → yes

1, 20-21 > This sentence is awkward, please rephrase, e.g. Geological archives have the potential to provide information on the past states of climate variables at the global and regional level, and their evolution in time. → rephrased according to your suggestion

1, 26-27 > what do you mean by "ice sheets . . . are apparently also teleconnected with global sea level"? Please rephrase → rephrased to: which control at least the climate of the high latitudes, but are apparently connected with global Sea level changes

2, 1-4 > How did you screen ~2000 papers? Did you use some search algorithm and keywords? → The special issue, which is the manuscript belonging to emerged out of the Palmod project. One prerequisite for PalMod is to work with publically available datasets only – from the most impactful or cited papers. In a first step, at least every abstract of the most cited papers was read. Afterwards, we searched for available datasets to the belonging paper. In a second step, datasets in Pangaea, noaa-ncdc, global pollen database (Neotoma), ice core database from Copenhagen university, SISAL speleothem database and European pollen database (EPD) were downloaded. The most complete records well dated back to 60 000 yr 2bk and high resolution samples were used for the compilation. Keywords used in the database research were e.g. speleothem, palaeo, ice core, dust, aeolian, eolian, pollen etc.

→ this will be part of the revised methods section in detail.

2, 5-6 > Have you considered paleolake levels as a potential proxy as well? → No, because not every region offers available data for that kind of proxy. I can imagine using paleolake levels in further works on that topic, probably in other regions.

2, 7 > "Arid" rather than "desert" → rephrased to "thus indicate an arid climate"

3, 12 > "The synthesis" rather than "The comparison"? → yes

3, 12 > In this section you should explain in a very transparent way which are the rules for selecting specific records. And why specific one(s) are used to calculate the aridity index, rather than others (within a given region, e.g. Bunker vs Spannagel Cave in Figure 2). In addition, what are the rules to determine the time step of the aridity index, given that the 3 records typically have different time axes?

→ as you suggest, the "Methods" section will be completely revised. We will upload the revised version of the manuscript as soon as possible, but not before 20.11.19.

3, 14 > What do you mean by "we use the original stratigraphy"? Aren't you saying that you port the chronologies to the GICC05 time scale? In addition, there is no mention as to how this operation was carried out: did you use some software?

→ We did not port the chronologies. We just homogenized the age labels BP (referred to 1950, or sometimes to year of paper release), ka BP etc. to years b2k (years before the year 2000 CE), which belongs to the GICC05 notation. The sentence will be rephrased for better understanding to: "We used the original stratigraphy of all records, but homogenized the notation of the age scales to yr b2k if possible." We compare data on a multi-millennial scale, thus uncertainties between BP, ka and b2k age scales are not that important. We will incorporate the last sentence into the method section.

3, 19-20 > What do you mean by "the errors . . . below 4% in total"? → rephrased to "growth data we used for this synthesis, all are below 4 % uncertainty"

3, 27-29 > Please rephrase this sentence → rephrased to: "The time resolution of the pollen profiles is often low, but we have chosen the accessible highest resolution data of each selected region for the comparison. The record in addition must have been reliably dated to be selected."

4, 1 > What is the global climate structure? →" The global climate evolution with processing time…"

4, 4 > "For THE Northern Hemisphere . . ." and so forth, please review the use of the language throughout the manuscript → we do agree with this comment. The use of language will be revised after other referee comments or comments in general were made to avoid duplications.

4, 8 > Larger than what? → "Grains of sand-size can be deflated…"

4, 16 > Do you mean precipitation proxies? → yes, rephrased for precision to "from all available precipitation proxies"

4, 19 > "divided in three parts" is not clear at all. I guess what you are trying to say is that you assign each point in the pollen / dust time series to a category from 0 to 2, based on the current value of the rescaled record as a percentage with respect to the top value (which corresponds to 100%)? Is that correct? However, it is not clear what are those original values. One can only try to guess it is maybe the percentage of tree pollen is the whole pollen assemblage for a given point? What is it for dust? It could be many things since you indicated several different proxies for dust. In fact by looking at the

supplement it seems it depends on each different proxy. Also, you do not spell out how you calculate the aridity index, one can grasp from the caption of table 1 that is the sum of the three "scores" for speleothem, tree pollen, and dust. Your procedure and the rationale behind it should be explained in detail and clearly in the methods section. → See reply to 3, 12, section "Methods" will be revised in detail.

5, 2-9 > This section is also very confusing, it should be profoundly revised. First, you should probably mention that there are uncertainties on the age of the samples, and uncertainties on the specific values of the variables, in addition to their uncertainty as proxies for a particular system. Second, you should clarify which are the cases where you have an uncertainty estimate from the original study and what it refers to. Then you can talk about the case where you have to assign the uncertainty arbitrarily to each sample in your time series, and you should specify on what grounds you assign a particular values (it could be the reference to a paper using the same kind of proxy, for instance). Fourth, as a key to read Table 2, you should describe explicitly if you have only one record for each kind of proxy for each region, or else how you dealt with multiple records. Finally, it may be more interesting to use other records than the "chosen" one, where available, to calculate the aridity index, as a metric for uncertainty / intra-regional variability. → See reply to 3, 12

5, 14 > I am not sure what you mean by "one of the large feedback regions": please rephrase → rephrased to "Central Europe is related strongly to North Atlantic climate changes."

5, 19-20 > VARVE not warve → changed within the manuscript

6, 4-5 > What does the dust concentration in the NGRIP ice core have to do with aridity in central Europe? I don't think it is appropriate to make such a statement without further discussion. As you know, there are several hypotheses concerning the interpretation of the Greenland dust records (e.g. (Mayewski et al., 2014; Steffensen et al., 2008)), and the major source of dust to the Greenland ice sheets are not uniquely attributed to Europe, to say the least (e.g. (Bory et al., 2003; Rousseau et al., 2014; Svensson et al., 2000; Újvári et al., 2015)). In view of these aspects, please state explicitly what is the link in your line of reasoning (e.g. generalized aridity in the northern hemisphere, in Eurasia,…), and what are the assumptions you make (e.g. Europe is major dust source to Greenland?), justifying them with adequate references to the literature.

→ We agree. We would have liked to show a well dated loess record like "Nussloch" but numerical timeseries were not available at present.

→ p6,l4-5 rephrased to ". An intermediate dust content in the ELSA-Dust-Stack suggest an intermediate to low aridity, which is supported by a similar pattern of low dust concentration in the NGRIP ice core. This corresponds to an overlying process, affecting both regions during this time." The authors would not make a statement on the source region beyond the papers, which are mentioned above.

6, 22 > Tree pollen? → "complete absence of all pollen." The record is counted for the whole time span and there were no preserved pollen at all.

6, 23 > "precipitation was at the lowest values of the whole record": which record are you referring to? To speleothem records? The aridity index? → yes

6, 29 > Which speleothem? → specified to: "Speleothem growth in Spannagel and Bunker Cave"

S1-S9 > In these sections of the supplement I would expect to find more specific considerations on the selections of records (e.g. why data from (Pourmand et al., 2004) are not included in S1? Or (Skonieczny

et al., 2019) in S2? Or the loess records in the discussion about central Europe? What's the link between EDML dust and Oceania? Etc. . ..), before discussing those that are selected. Also, I did not find the details of how data are aggregated into the aridity index (e.g. why sometimes 4 records are considered, sometimes 2?). As mentioned already, the general rules for data selection should be spelled out clearly in the methods sections, and specific choices of notable datasets not included should be discussed in the supplement.

→ Like mentioned within the manuscript (introduction) and Supplement 11, as well as within the reply to 2, 1-4: Data needed to be publically available to be considered within the synthesis. I did not find related data in the publically available databases for Pourmand et al. 2004 for example. Therefore, the paper was not mentioned. Many popular records were not chosen, as they do not reach until 60 ka, do not belong to the selected regions or do have lower sample resolution, as the scope of the study was different (for example longer chronologies).

The aggregation of data could be written in more detail, I agree. But the aridity index was built up every time out of the three proxy types (speleothem, tree pollen, dust) beside St. Barbara basin, where no dust record was available. This will be spelled out in more clearness.

The rules will be explained in the revised method section in more detail than before.

8, 6 > RELATED information → yes, specified

8, 11 > The proxies show an opposing signal ? → rephrased to "where the proxies show an opposing signal to the other archives due to regional effects"

8, 13 > It is not clear how Figure 4 was produced. What is the role of the "additional information"? Are those the thin overlapping bars? → Fig. 4 is a graphical interpretation of Fig. 3 for better visualisation of the changes of the analyzed regions through time.

The additional information (Stebich et al., 2015 Pollen Sihailongwan Maar Lake) are Holocene pollen, because Mingram et al. (2018) does not show them. Therefore, we used the additional information after the construction of the aridity index to complete the aridity interpretation. Other additional information only were used within the interpretation of Fig. 3 and 4 to support statements on aridity.

9, 11-12 > Please rephrase → irrelevant to the topic of the paper, therefore deleted

9, 15 > "impair"? → see next reply to 9,21

9, 21 > How do you define a climate improvement? Please avoid terms like improvement and amelioration, impair?; expressions describing the changing state of the discussed variable should be preferred, such as drier, wetter, colder, etc.

→ climate improvement is used for warmer and wetter climate, according to better living conditions. Climate deterioration (impair etc.) is used for colder and drier climate, according to poorer living conditions. This will be added within the introduction.

10, 8 > representative of the Cariaco Basin → yes
11, 4 > WHEN both hemispheres → yes
14, 6-9 > Not clear, please rephrase → yes
14, 24 > SPATIAL trends → yes
16, 12 > All regions analyzed here → yes

Bory, A. J.-M., Biscaye, P. E. and Grousset, F. E.: Two distinct seasonal Asian source regions for mineral dust deposited in Greenland (NorthGRIP), Geophys. Res. Lett., 30(4), doi:10.1029/2002GL016446, 2003.

Mayewski, P. A., Sneed, S. B., Birkel, S. D., Kurbatov, A. V. and Maasch, K. A.: Holocene warming marked by abrupt onset of longer summers and reduced storm frequency around Greenland, J. Quat. Sci., 29(1), 99–104, doi:10.1002/jqs.2684, 2014.

Pourmand, A., Marcantonio, F. and Schulz, H.: Variations in productivity and eolian fluxes in the northeastern Arabian Sea during the past 110 ka, Earth Planet. Sci. Lett., 221(1–4), 39–54, doi:10.1016/S0012-821X(04)00109-8, 2004.

Rousseau, D.-D., Chauvel, C., Sima, A., Hatté, C., Lagroix, F., Antoine, P., Balka- nski, Y., Fuchs, M., Mellett, C., Kageyama, M., Ramstein, G. and Lang, A.: European glacial dust deposits: Geochemical constraints on atmospheric dust cycle mod- eling: European Glacial Dust Deposits, Geophys. Res. Lett., 41(21), 7666–7674, doi:10.1002/2014GL061382, 2014.

Skonieczny, C., McGee, D., Winckler, G., Bory, A., Bradtmiller, L. I., Kinsley, C. W., Polissar, P. J., De Pol-Holz, R., Rossignol, L. and Malaizé, B.: Monsoon-driven Saharan dust variability over the past 240,000 years, Sci. Adv., 5(1), eaav1887, doi:10.1126/sciadv.aav1887, 2019.

Steffensen, J. P., Andersen, K. K., Bigler, M., Clausen, H. B., Dahl-Jensen, D., Fischer, H., Goto-Azuma, K., Hansson, M., Johnsen, S. J., Jouzel, J., Masson-Delmotte, V., Popp, T., Rasmussen, S. O., Rothlisberger, R., Ruth, U., Stauffer, B., Siggaard- Andersen, M.-L., Sveinbjornsdottir, A. E., Svensson, A. and White, J. W. C.: High- Resolution Greenland Ice Core Data Show Abrupt Climate Change Happens in Few Years, Science, 321(5889), 680–684, doi:10.1126/science.1157707, 2008.

Svensson, A., Biscaye, P. E. and Grousset, F. E.: Characterization of late glacial continental dust in the Greenland Ice Core Project ice core, J. Geophys. Res. Atmospheres, 105(D4), 4637–4656, doi:10.1029/1999JD901093, 2000.

Újvári, G., Stevens, T., Svensson, A., Klötzli, U. S., Manning, C., Németh, T., Kovács, J., Sweeney, M. R., Gocke, M., Wiesenberg, G. L. B., Markovic, S. B. and Zech, M.: Two possible source regions for central Greenland last glacial dust, Geophys. Res. Lett., 42(23), 10,399-10,408, doi:10.1002/2015GL066153, 2015.

---

## Referee Comment (RC2) · Anonymous Referee #2 · 15 Nov 2019

https://doi.org/10.5194/cp-2019-108

This study is an attempt to provide a global synthesis of aridity over the last 60 ka using a number of selected terrestrial (speleothem, lake, loess) and marine records of 10 regions on the globe. While the major outcomes (including the aridity index) of the manuscript results from an immense effort of synthesising various records having different chronologies, resolutions, proxies and associated uncertainties, it is particularly hard to judge what has been really done in terms methodology and if this is sound or not. In agreement with the opinion of referee #1, the methods section (+Supplementum) should be much more transparent to the reader, and the sometimes sloppy text and superficial statements, inappropriate usage of specific terms must be carefully re-

vised. This also applies to some of the argumentations (e.g. Europe-Greenland aridity relations).

In general, it is suggested that the authors should 1) clearly present the core concept of proxy record selection for this synthesis, and 2) exclusively include records having independent absolute chronologies (i.e. NGRIP/MIS tuned chronologies should be avoided). In my view, the concept of excluding proxy records, which are otherwise well-dated, but do not extend back to 60 ka, should be revised or at least some justifications for this decision are required. Just to mention one excellent example: the Nussloch loess record in Germany (Central Europe), which has a quite well-defined, robust and precise 14C-chronology (extending back to 55 ka), has been omitted. Moreover, further details on the aridity index and age uncertainty calculations of proxy records must be provided. In my view, any proper assessment of the scientific content of this work can only be provided after a thorough revision of the methodological part.

Specific comments

**Manuscript**

Page 1, lines 25-26, "MIS2 (Last Glacial Maximum (LGM) 24 000-14 700 yr b2k)": This is misleading, as the LGM was a globally recognizable, peak glacial period between 26-19 ka (broadly speaking), while the ages given are the widely accepted boundaries of MIS 2.

Page 2, lines 7-12: I would say dust is dominantly from deserts, but other dryland ecosystems (shrublands, grasslands and even forests with 300-500 mm annual rainfall; Breshears et al. 2003) can also produce fair amount of dust.

Page 3, lines 3-8: This is corroborated by other studies of loess records, 14C-dated in high (Nussloch, Germany; Moine et al., 2017) and extremely high (Dunaszekcso, Hungary; Ujvari et al., 2017) resolution. Why not using at least the Nussloch record for Central Europe, beyond the ELSA stack?

Page 3, line 15: Provide more details on GICC05 (b2k) timescale conversion. Does this simply mean a 50 yr addition to the calibrated radiocarbon chronologies? How this approach was applied to the luminescence chronologies?

Page 4, lines 8-9: What does this sentence mean?

Page 4, lines 9-10: Does "eolian content" mean eolian fraction of sediments?

Page 4, lines 12-14: Fuzzy text (K/C ratio and related interpretations) must be revised.

Page 4, line 25, Table 1: It is still not entirely clear how these aridity values are calculated from dust. Dust MARs or grain size or what has been used and in which way? What does the internal normalization mean?

Page 5, section 2.2: I suppose this section describes age uncertainties. State this clearly. Have you considered proxy uncertainties?

Page 5, line 7: Does the "error of our aridity index" mean uncertainties related to dating/chronological uncertainties?

Page 5, line 10, Table 2 (header): Clarify that the "tree pollen/eolian dust uncertainties" are dating/chronology uncertainties. Provide more details on the method used for uncertainty estimations. Has this been done by Monte Carlo simulations?

Page 5, line 14: In what sense is Central Europe a "feedback region"? Clarify.

Page 6, lines 4-5: Provide numbers for "low dust concentration". Do you refer to Greenland or dust source regions (or Central Europe) when talking about "intermediate to low aridity" in this sentence?

Page 6, line 6, "49.000 yr b2k": Provide uncertainty for this date.

Page 6, line 25: Provide numbers of "extreme cold temperature" for the NGRIP site based on reconstructions of Kindler et al. (2014) and state clearly that these temperature estimates are not only from $\delta$18Oice, but a combination of $\delta$18Oice and $\delta$15N

measurements (+∆age).

Page 6, lines 24-25: In this sentence you suppose a direct link between Central Europe and Greenland in terms of dust transport. On what basis? To my knowledge, the possibility of European dust sources for central Greenland (over the LGM) has been proposed in Ujvari et al. (2015), specifically based on Sr-Nd isotopic compositions. I would rather emphasize that the ELSA record reflect regional conditions and these could have differed from those in Greenland.

Page 7, lines 10-11: This is a bit strange suggestion or at least not explained properly. Central Europe cannot be taken as a reference, as no other regions. All regions have their own climatic history. The Greenland ice core records are usually taken as stratigraphic correlation targets as they have an unprecedented resolution, layer-counting chronology and reliable proxies.

Page 9, lines 11-12: I'm wondering why so many recent papers include one completely off-topic sentence about the migration of anatomically modern humans into Europe? Just one sentence pops up without any further discussion in most of these papers, including this one. This is pure hypothesis without any further evidence, therefore I strongly suggest deleting this sentence.

Page 12, lines 17-19: Talking about Heinrich-events, these should be indicated in Figure 6. Also, from where do you know if these are H-events or not in the studied dust records? Timing?

Page 13, line 4, "turning point": Do you refer to tipping points here?

Page 13, line 19: I suggest deleting the Gobi after "China" (in parenthesis), as there many other deserts in China, including the Taklimakan, Tengger, Hobq, Mu Us etc. deserts.

Page 14, lines 13-17 and Page 16, lines 2-6: These text parts should go somewhere in the Methodology section, in my opinion.

Page 14, line 28: Which simulation do you refer to? Barron and Pollard's?

**Supplementum**

Page 21, line 21: Records with tuned chronologies should be excluded, in my view.

Page 22, lines 2-3: This is exactly the reason, which precludes unambiguous GI identifications in OSL-dated records, including Jingyuan in China. Such an "exercise" is difficult even using 14C-chronologies, having an order of magnitude lower uncertainties.

Technical corrections

Page 3, line 23: write "pollens"

Page 3, line 30: dropstones? I would use "lithic clasts" or "detritus" or something like that

Page 5, line 19: write "varved" (same later)

Page 5, line 22: specify this abbreviation: Greenland Insterstadial (GI)

Page 6, line 2: write "caves"

Page 6, line 3: replace "strong precipitation amount" by "wet climate" or a similar expression

Page 6, line 5: write "beginning" (same below)

Page 6, line 10: write "hiatus"

Page 6, line 14: replace "on" by "to" after "apparently" and use "underlying" instead of "overlaying"

Page 9, line 18: I can't find these red bars. Or do you refer to figure 4?

Page 9, line 23: delete "bevor" and write "before"

Page 10, line 3, Figure 4 caption (second row), "red bars indicate high humidity": this should be aridity, I guess

Page 14, line 3: delete "at" and use "in" before "Central Chinese"

Page 14, lines 7-9: first half of sentence makes no sense, rewrite please

Page 16, line 12: "large humidity" is bad phrasing, write "increased humidity" or simply "wet phase"

Page 16, line 13: write "considerably"

References

Breshears, D.D. et al., 2003.Wind and water erosion and transport in semi-arid shrubland, grassland, and forest ecosystems: quantifying dominance of horizontal wind-driven transport. Earth Surf. Process. Landf. 28, 1189–1209.

Moine, O. et al., 2017. The impact of Last Glacial climate variability in west-European loess revealed by radiocarbon dating of fossil earthworm granules. Proc. Natl. Acad. Sci. USA 114, 6209–6214.

Ujvari et al., 2015. Two possible source regions for central Greenland ice core dust. Geophys. Res. Lett. 42, 10399–10408.

Ujvari, G. et al., 2017. Coupled European and Greenland last glacial dust activity driven by North Atlantic climate. Proc. Natl. Acad. Sci. USA 114, 10632–10638.

---

## Referee Comment (RC3) · Anonymous Referee #3 · 22 Nov 2019

Fuhrmann et al. collected published proxy data to assess changes in regional aridity for various regions. To make the data comparable and reduce the complexity, the authors developed an aridity index that is compared with modelled precipitation anomalies between MIS3 and the LGM and MIS3 and the preindustrial. Generally, the compilation and homogenization of aridity records and their comparison with the results of model experiments is an interesting approach. However, as outlined below, I feel that (i) the methods are not sufficiently described to allow a proper assessment of the approach and significance of the results, (ii) that the authors use unreasonable generalizations for the definitions of time slices and regions, and (iii) that there is no significant new information added by the paper. I recommend to reconsider the paper only after a fundamental revision.

[Figure]

1) Parts of the paper are written in a very confusing style. For example, on p3/l14 the authors describe that they "...use the original stratigraphy of all records". On p3/l16 they say "Speleothems are used for synchronisation between different archives of one region" which implies changes of the original stratigraphies.

2) Aridity index. The calculation of the aridity index is not sufficiently described, but as I understand from Table 1, the authors assign an integer value between 0 and 2 (or 0 and 1 for speleothem growth) to the different proxy records and then add the values(?). What do the authors mean with "...the original values have been recalculated into percentages, proportional to the maximum value of each specific dataset..."? Is the aridity index only calculated from speleothem growth, pollen and dust, or are other parameters included? In the methods section it is stated that "...isotope data like d18O, Sea Surface Temperature (SST) reconstructions or Ice Raft Debris (IRD) data are added to complete the picture." Are those records part of the aridity index? If not, to what have you included those data?

3) Uncertainty estimation. The uncertainty estimation needs better explanation. If the aridity index is binned into integer values between 0 and 5 (as I speculate), does it make sense that the error is smaller than 1 in some cases as for example shown in figure 2f?

4) Title: The title is misleading and not a good representation of the content. The data collection is far from being "global" since some of the most important regions (i.e. the Amazon) and much of the tropics (where aridity matters most) are not represented. I would suggest to find a title like "Regional aridity synthesis for the last 60ầL'000 years"

5) I find some of the generalizations and associations of records with specific regions strange and do not understand why this is done at all: For example in Figure 5 the Susah Cave (located at 33N/22E close to the Mediterranean) is labeled with NW Africa, and a Bahamas cave with the Cariaco Basin. The Cariaco Basin is under the influence of the ITCZ, the Bahamas are not. These are different systems and thousands of km

apart and do not necessarily anything to do with each other. The power of a compilation of high-resolution aridity records is that we may understand the regional response of the climate system to specific perturbations or forcings. Here, this useful information is compromised through an unreasonable combination of records from different systems and a very broad definition of time slices (see below).

6) The used LGM definition (24 to 14.7 ky) is very unfortunate and should be revised. The LGM has been previously defined to extend from 23 to 19 ka (Mix et al. 2001, Quat. Sci. Rev., 20, 627-657). This time interval has been chosen, because the climate is comparably stable. The LGM definition of the authors, however, merges the actual LGM with Heinrich Stadial 1, during which the climate system was exposed to significant changes in external forcings and internal perturbations. The global deglacial warming starts at about 18.5 with the onset of HS1 shortly before the deglacial increase in atmospheric $CO_2$ (Shakun et al. 2012, Nature, 484, 49-54). The distribution of orbital insolation changes significantly and we see a change from a relatively strong AMOC to a weak AMOC with the onset of HS1 (McManus et al. 2004, Nature, 428, 834-837). Very likely, even the deglaciation of the Southern parts of the Ice sheets starts already during HS1 as evidenced by records related to river discharge at some of the more southerly locations (i.e. Menot et al. 2006, Science, 313, 1623-1625).

7) Comparison to model experiments. In my view, a comparison to model experiments only makes sense, if there is a coherency between the changes in boundary conditions applied to the model and those expected for the reconstructed time slices. This is not the case here: The model experiments have been performed with fixed boundary conditions. By contrast the definitions of the time slices (LGM: 24 000-14 700 yr b2k, MIS3: 60 000 – 24 000 yr b2k) are so broad that huge changes in boundary conditions and perturbations are present within each time slice. Hence it is impossible to pin down potential reason or mechanisms for the changes. The authors have done an effort to specifically compile high-resolution records and yet they lose all the information through unreasonable broad time slice definitions.

More specific points:

-p1/l11: "In comparison, the MIS2 interval becomes arid in all northern hemisphere records, but the peak arid conditions of the Last Glacial Maximum (LGM) differ in duration and intensity among regions." This is not true. MIS2 includes the B/A interval which is clearly very humid. Peak arid conditions in much of the northern Hemisphere tropics occur during HS1, which should not be confused with the LGM

-p1/l17: "two focus" must be "two foci"

-p2/l13: "We present the 10 key regions..." Key for what? Many important "key" regions of global importance (i.e. the Amazon) are missing

-p4/l1: "The global climate structure is well documented within Greenland and Antarctica ice cores". I disagree with this statement. Ice cores represent the high latitudes. There is very little info about the tropics and subtropics, i.e the strength of the monsoons, neoglaciation etc.

-P5/l14: "Central Europe is one of the large feedback regions to North Atlantic climate changes" Do the authors mean that Central Europe is amplifying North Atlantic climate changes?

---

## Referee Comment (RC4) · Anonymous Referee #4 · 30 Nov 2019

Dear Authors,

You provide a manuscript attempting to synthesize global aridity. You select several key regions with a decent data coverage from different geoarchives. Having read your manuscript and the discussion up to date, I have a clear opinion about your manuscript.

Your conceptual idea of using suites of geoarchives to address aridity is in my opinion great and clearly worth investigating and publishing. At the same time I hold the opinion that several aspects need some work before publication. I agree with most points of other reviewers, see also comments below.

My main comments are: You mention that you focus on openly available data in Supplements to papers. The ELSA vegetation stack data is available in the Pangaea database

– using also the NOAA and PANGAEA databases as source would have been appropriate. Please add a Table in Supplements where data are from (websites/databases). You screened 'about 2000 papers' – that is not a reproducible statement. Please ensure that your data processing is 100% transparent and reproducible. Yet I have only a decent idea how this was done. If necessary, please provide sheets and computer code in Supplements.

Uncertainty of the aridity index seems constant with time and data resolution – that clearly does not make sense. Please adjust your method of uncertainty estimation to be at least more realistic. An idea may be to use a relative reliability index, where both lowest data resolution and highest data uncertainty play a role. You do mention that different age models will have an impact on your results. It would be nice to get an idea how this impacts results in one example, but I do see that this is difficult.

For Asia and Europe, more than single dust records are available in databases – please synthesize these. The presented data selection seems biased towards the authors' work, and I suggest to compile data for several regions in a more extensive way, and maybe focus on less regions. Obvious questions are, why are data from Tenaghi Phillipon and more Mediterranean cores not used? Why is there only 1 dust record from Asia and Europe? More are available.

The data selection for several regions is problematic in my opinion: Southern Europe: Data from the Lac du Bouchet is in my opinion hardly comparable to the Portuguese Margin – two datasets from the Portuguese Margin are probably leading to a location bias here, too. This should in my opinion at least be discussed. Why are SST data from the Mediterranean not included? Why are loess data from Spain neglected? Cariaco Basin: the dust record here may actually not reflecting local dust, but African aridity (also discussed in the reference you cite) – please be more self-critical in the discussion.

More drastically, data from New Zealand and Australia probably do not indicate the

same climate system at all – combining these at least requires a more sensitive discussion. In my opinion these should not be combined for an aridity analysis.

More detailed comments are:

Please avoid abbreviations in the abstract

The first sentence of the introduction is in my opinion not generally true.

You begin with your own data – OK, a scientific reasoning is more appropriate.

Page 8, line 9: You mention geographic regions and China as country – please avoid such political statements

Page 9, Line 11f: this is not a result, but more speculation

---

## Author Comment (AC2) · 18 Dec 2019

→ We thank reviewer#2 for his constructive and helpful comments to improve the manuscript. We have revised it in detail. Our point-py-point response to your suggestions are provided in red in the attached PDF file.

This study is an attempt to provide a global synthesis of aridity over the last 60 ka using a number of selected terrestrial (speleothem, lake, loess) and marine records of 10 regions on the globe. While the major outcomes (including the aridity index) of the manuscript results from an immense effort of synthesising various records having different chronologies, resolutions, proxies and associated uncertainties, it is particularly hard to judge what has been really done in terms methodology and if this is sound or not. In agreement with the opinion of referee #1, the methods section (+Supplementum) should be much more transparent to the reader, and the sometimes sloppy text and superficial statements, inappropriate usage of specific terms must be carefully revised. This also applies to some of the argumentations (e.g. Europe-Greenland aridity relations).

In general, it is suggested that the authors should 1) clearly present the core concept of proxy record selection for this synthesis, and 2) exclusively include records having independent absolute chronologies (i.e. NGRIP/MIS tuned chronologies should be avoided). In my view, the concept of excluding proxy records, which are otherwise well- dated, but do not extend back to 60 ka, should be revised or at least some justifications for this decision are required. Just to mention one excellent example: the Nussloch loess record in Germany (Central Europe), which has a quite well-defined, robust and precise 14C-chronology (extending back to 55 ka), has been omitted. Moreover, further details on the aridity index and age uncertainty calculations of proxy records must be provided. In my view, any proper assessment of the scientific content of this work can only be provided after a thorough revision of the methodological part.

→ Well cited papers - where data were not available - were not included into the compilation. This was a prerequisite from PalMod, that is why the paper is presented in the special issue. We fully agree, that these data would be useful, but they are not publically available.

1) Is done now in the revised method section
2) This would be the perfect attempt. Unfortunately, those records are rare and often not publically available. Especially the Nussloch loess record is not available in databases to our knowledge. Also many other striking paleoclimate records either do not cover a larger part of the last 60 000 years or are not available from official data repositories. We move a section with not used, but important records from the supplement to the paper and extend it for more clearness.

Specific comments

**Manuscript**

Page 1, lines 25-26, "MIS2 (Last Glacial Maximum (LGM) 24 000-14 700 yr b2k)": This is misleading, as the LGM was a globally recognizable, peak glacial period between 26-19 ka (broadly speaking), while the ages given are the widely accepted boundaries of MIS 2.

→ Yes, changed to "the MIS2 (24 000-14 700 yr b2k)" and explanation of abbreviation "LGM" at page 1, line 13.

Page 2, lines 7-12: I would say dust is dominantly from deserts, but other dryland ecosystems (shrublands, grasslands and even forests with 300-500 mm annual rainfall; Breshears et al. 2003) can also produce fair amount of dust.

→ agree on that comment. We changed that according to referee#1 to "indicate an arid climate".

Page 3, lines 3-8: This is corroborated by other studies of loess records, 14C-dated in high (Nussloch, Germany; Moine et al., 2017) and extremely high (Dunaszekcso, Hungary; Ujvari et al., 2017) resolution. Why not using at least the Nussloch record for Central Europe, beyond the ELSA stack?

→ Well cited papers - where data were not available - were not included into the compilation. This was a prerequisite from PalMod, that is why the paper is presented in the special issue. We fully agree, that these data would be useful, but they are not publically available. We include a section with not used, but important records within the paper.

Page 3, line 15: Provide more details on GICC05 (b2k) timescale conversion. Does this simply mean a 50 yr addition to the calibrated radiocarbon chronologies? How this approach was applied to the luminescence chronologies?

→ We used the original stratigraphy of all records on the age scale of yr b2k. Sometimes this means an addition of 50 yr, yes. We compare data on a multi-millennial scale, thus uncertainties between BP, ka and b2k age scales are not that important. This is mentioned now in the revised method section.

Page 4, lines 8-9: What does this sentence mean?

→ rephrased to "The grainsize record from the loess plateau in China shows phases of aridity. The larger the sediment grains, the lower the precipitation and temperature and the higher the wind speeds."

Page 4, lines 9-10: Does "eolian content" mean eolian fraction of sediments?

→ yes, rephrased to "Dust or eolian content of the sediment is…"

Page 4, lines 12-14: Fuzzy text (K/C ratio and related interpretations) must be revised.

→ rephrased to "Kaolinite / chlorite ratio can be used as a dust proxy for the Mediterranean Sea region. Higher K/C ratios (more kaolinite than chlorite) indicates increased eolian dust transport. During humid periods, kaolinite was stored within lakes or basins - due to increased erosion - and deflated during arid periods (Ehrmann et al. 2017)."

Page 4, line 25, Table 1: It is still not entirely clear how these aridity values are calculated from dust. Dust MARs or grain size or what has been used and in which way? What does the internal normalization mean?

→ The whole methods section including data treatment and aridity index calculation is revised.

Page 5, section 2.2: I suppose this section describes age uncertainties. State this clearly. Have you considered proxy uncertainties?

→ We used the original stratigraphy of all records on the age scale of yr b2k but we are aware of a general error of up to ± 2 000 years for all MIS 3 dates. This is now incorporated into the introduction. Beside this, uncertainty estimation is revised as well.

Page 5, line 7: Does the "error of our aridity index" mean uncertainties related to dating/chronological uncertainties?

→ We estimated uncertainties of the proxies, as no original uncertainties are aware beside the age uncertainties of the speleothem growth phases.

Page 5, line 10, Table 2 (header): Clarify that the "tree pollen/eolian dust uncertain- ties" are dating/chronology uncertainties. Provide more details on the method used for uncertainty estimations. Has this been done by Monte Carlo simulations?

→ This was some kind of a simple Monte Carlo simulation, yes.

Page 5, line 14: In what sense is Central Europe a "feedback region"? Clarify.

→ rephrased according to Review#1: "Central Europe is related strongly to North Atlantic climate changes."

Page 6, lines 4-5: Provide numbers for "low dust concentration". Do you refer to Greenland or dust source regions (or Central Europe) when talking about "intermediate to low aridity" in this sentence?

→ rephrased according to Review#1: "An intermediate dust content in the ELSA-Dust-Stack suggest an intermediate to low aridity, which is supported by a similar pattern of low dust concentration in the NGRIP ice core. This corresponds to an underlying process, affecting both regions during this time."

Page 6, line 6, "49.000 yr b2k": Provide uncertainty for this date.

→ as previously mentioned, we are aware of up to ± 2000 years uncertainty for MIS3 dates (see reply to Page 5, section 2.2). Therefore, p.6 l.6 is rephrased to "The pollen composition change began around 49 000 yr b2k towards more grass and herbs pollen." The updated manuscript is changed according to this suggestion.

Page 6, line 25: Provide numbers of "extreme cold temperature" for the NGRIP site based on reconstructions of Kindler et al. (2014) and state clearly that these temperature estimates are not only from $\delta 18O_{ice}$, but a combination of $\delta 18O_{ice}$ and $\delta 15N$ measurements (+Δage).

→ We assume, you mean Page 6, line 21 instead of line 25: We agree about your addition of $\delta 18O_{ice}$ to this sentence, but we refer to Figure 3a and Figure 5 of the cited Paper, where the temperature reconstruction is performed using $\delta 18O$. Therefore, we specified the sentence according to your suggestion: "The NGRIP $\delta 18O_{ice}$ whereas shows a phase of extreme cold temperatures (-45° to -50° C) during this time". For age uncertainty, we refer to the previous comment, as no uncertainty is given with the original paper, even within the supplement. We are aware of age uncertainties within the GICC05 age scale of Rasmussen et al. (2014) but would not make any statement about age uncertainty beyond that.

Page 6, lines 24-25: In this sentence you suppose a direct link between Central Europe and Greenland in terms of dust transport. On what basis? To my knowledge, the possibility of European dust sources for central Greenland (over the LGM) has been proposed in Ujvari et al. (2015), specifically based on

Sr-Nd isotopic compositions. I would rather emphasize that the ELSA record reflect regional conditions and these could have differed from those in Greenland.

→ The dust records of Central Europe and Greenland correlate over most of the last 60 000 years. ELSA-Dust-Stack (2009) shows the GI / GS changes in detail. The biggest difference between ELSA-Dust-Stack and NGRIP-Dust can be seen in the period of 26 000 to 23 000 yr b2k, where the dust content in the NGRIP core has two distinct maxima, while the dust content in the Central European record increases only slightly. We do not want to suppose a direct link in dust transport as we know, that the majority of NGRIP dust is from Asian deserts (Steffensen et al., 2008, Mayewski et al., 2014 etc.). We interpret the observed correlation in such a way that similar climatic conditions must have prevailed in both regions.

Page 7, lines 10-11: This is a bit strange suggestion or at least not explained properly. Central Europe cannot be taken as a reference, as no other regions. All regions have their own climatic history. The Greenland ice core records are usually taken as stratigraphic correlation targets as they have an unprecedented resolution, layer-counting chronology and reliable proxies.

→ Agree. We rephrased this paragraph to: "The Central Europe region acts as an example for the nine other regions. For further detailed information on the other nine regions, see Supplement S1-S9."

Page 9, lines 11-12: I'm wondering why so many recent papers include one completely off-topic sentence about the migration of anatomically modern humans into Europe? Just one sentence pops up without any further discussion in most of these papers, including this one. This is pure hypothesis without any further evidence, therefore I strongly suggest deleting this sentence.

→ Agree, we deleted this sentence

Page 12, lines 17-19: Talking about Heinrich-events, these should be indicated in Figure 6. Also, from where do you know if these are H-events or not in the studied dust records? Timing?

→ As you suggest, we incorporate HE times within Fig. 6. Heinrich Events either have been interpreted by the original authors (Hodell et al. 2013; Collins et al., 2013 for Southern Europe, Portuguese margin, Collins et al., 2013 for NW-Africa region) or phases of increased dust fall within the timings of H-Events.

Page 13, line 4, "turning point": Do you refer to tipping points here?

→ To our knowledge, turning point and tipping point is used in similar ways. We accept your suggestion and change this to tipping point

Page 13, line 19: I suggest deleting the Gobi after "China" (in parenthesis), as there many other deserts in China, including the Taklimakan, Tengger, Hobq, Mu Us etc. deserts.

→ We fully agree on that comment, as especially Taklimakan desert is more important on dust transport than Gobi.

Page 14, lines 13-17 and Page 16, lines 2-6: These text parts should go somewhere in the Methodology section, in my opinion.

→ We moved the methodological part of the model simulation to a new subchapter of the methods and extended it for more clarity (p.14, l.12-17). We do not want to move P.16,l.2-6 as they describe Table 3. Therefore, these lines should remain close to it.

Page 14, line 28: Which simulation do you refer to? Barron and Pollard's?

→ Yes, added this to the sentence.

**Supplementum**

Page 21, line 21: Records with tuned chronologies should be excluded, in my view.

→ There are no records used within this paper, which are exclusively dated by tuning. All records, which are used, are 14C or Th/U dated (or OSL dated for Jingyuan). Most of the data sets are correlated crosswise or tuned afterwards with other nearby data sets to strengthen the stratigraphy.

Page 22, lines 2-3: This is exactly the reason, which precludes unambiguous GI identifications in OSL-dated records, including Jingyuan in China. Such an "exercise" is difficult even using 14C-chronologies, having an order of magnitude lower uncertainties.

→ As you can see on Page 6, line 15 of the Supplement, we are aware of this. Therefore, Sun et al., 2010 talk about loess interstadial / stadial, which we also did for this paper.

Technical corrections

Page 3, line 23: write "pollens"→ Most Palynological papers use 'pollen' as plural, thus we follow this notation.

Page 3, line 30: dropstones? I would use "lithic clasts" or "detritus" or something like that

→ As IRD layers are not part of the aridity index but comparable to the records of Southern Europe and Portuguese margin. Therefore, the whole sentence has been deleted.

Page 5, line 19: write "varved" (same later) → Yes

Page 5, line 22: specify this abbreviation: Greenland Interstadial (GI) → This is done previously on Page 3 line 4-5

Page 6, line 2: write "caves" → Yes

Page 6, line 3: replace "strong precipitation amount" by "wet climate" or a similar expression → Yes

Page 6, line 5: write "beginning" (same below) Page 6, line 10: write "hiatus" → Yes

Page 6, line 14: replace "on" by "to" after "apparently" and use "underlying" instead of "overlaying" → Yes

Page 9, line 18: I can't find these red bars. Or do you refer to figure 4? → Yes, we mentioned the wrong Figure!

Page 9, line 23: delete "bevor" and write "before" → Yes

Page 10, line 3, Figure 4 caption (second row), "red bars indicate high humidity": this should be aridity, I guess → Yes

Page 14, line 3: delete "at" and use "in" before "Central Chinese" → Yes

Page 14, lines 7-9: first half of sentence makes no sense, rewrite please → Rephrased to "In order to our reconstructed precipitation we employ the coupled climate model COSMOS which was developed at the Max-Planck Institute for Meteorology in Hamburg." → deleted "with the large-scale pattern of model simulation,"

Page 16, line 12: "large humidity" is bad phrasing, write "increased humidity" or simply "wet phase" → Yes

Page 16, line 13: write "considerably" → Yes

References

Breshears, D.D. et al., 2003.Wind and water erosion and transport in semi-arid shrubland, grassland, and forest ecosystems: quantifying dominance of horizontal winddriven transport. Earth Surf. Process. Landf. 28, 1189–1209.

Moine, O. et al., 2017. The impact of Last Glacial climate variability in west-European loess revealed by radiocarbon dating of fossil earthworm granules. Proc. Natl. Acad. Sci. USA 114, 6209–6214.

Ujvari et al., 2015. Two possible source regions for central Greenland ice core dust. Geophys. Res. Lett. 42, 10399–10408.

Ujvari, G. et al., 2017. Coupled European and Greenland last glacial dust activity driven by North Atlantic climate. Proc. Natl. Acad. Sci. USA 114, 10632–10638.

---

## Author Comment (AC3) · 18 Dec 2019

Fuhrmann et al. collected published proxy data to assess changes in regional aridity for various regions. To make the data comparable and reduce the complexity, the authors developed an aridity index that is compared with modelled precipitation anomalies between MIS3 and the LGM and MIS3 and the preindustrial. Generally, the compilation and homogenization of aridity records and their comparison with the results of model experiments is an interesting approach. However, as outlined below, I feel that (i) the methods are not sufficiently described to allow a proper assessment of the approach and significance of the results, (ii) that the authors use unreasonable generalizations for the definitions of time slices and regions, and (iii) that there is no significant new information added by the paper. I recommend to reconsider the paper only after a fundamental revision.

→ We acknowledge Referee#3 for his review and constructive and helpful comments. They have greatly improved the manuscript, especially the use of some definitions. We have incorporated the suggestions within the manuscript. See our point-by-point reply to your comments in red in the PDF.

1) Parts of the paper are written in a very confusing style. For example, on p3/l14 the authors describe that they "...use the original stratigraphy of all records". On p3/l16 they say "Speleothems are used for synchronisation between different archives of one region" which implies changes of the original stratigraphies.

→ We uploaded an updated version of the manuscript, which incorporates a fundamentally revised method section. Your suggested point is spelled out now in detail (p4, l2). The linguistic revision will take place as soon as the scientific content of the manuscript has been accepted.

2) Aridity index. The calculation of the aridity index is not sufficiently described, but as I understand from Table 1, the authors assign an integer value between 0 and 2 (or 0 and 1 for speleothem growth) to the different proxy records and then add the values(?). What do the authors mean with "...the original values have been recalculated into percentages, proportional to the maximum value of each specific dataset..."? Is the aridity index only calculated from speleothem growth, pollen and dust, or are other parameters included? In the methods section it is stated that "...isotope data like d18O, Sea Surface Temperature (SST) reconstructions or Ice Raft Debris (IRD) data are added to complete the picture." Are those records part of the aridity index? If not, to what have you included those data?

→ The revised method section includes an detailed explanation on the used proxies, generation of the aridity index and all your mentioned points. Additional information are explained now on Page 10, lines 13ff. They only were used for generating Fig. 4, even in the previous version of the paper.

3) Uncertainty estimation. The uncertainty estimation needs better explanation. If the aridity index is binned into integer values between 0 and 5 (as I speculate), does it make sense that the error is smaller than 1 in some cases as for example shown in figure 2f?

→ Yes, we could also quantize the errors. This would highlight the absolute classification error. Since no error is larger than 1.5 this results in rounded errors of 0 or 1. The chosen style emphasizes the reliability of the underlying data, since in the worst cases the aridity index is of by one bin.

→ For example, Fig. 2f that you mention has different errors. During the phase of 15 000-25 000 the error is about 1, while around 45 000 it is much smaller and around 50 000 it increases again.

4) Title: The title is misleading and not a good representation of the content. The data collection is far from being "global" since some of the most important regions (i.e. the Amazon) and much of the tropics (where aridity matters most) are not represented. I would suggest to find a title like "Regional aridity synthesis for the last 60 000 years"

→ We agree on your comment and rephrase the title: "Aridity synthesis for 10 selected key regions of the global climate system during the last 60 000 years"

5) I find some of the generalizations and associations of records with specific regions strange and do not understand why this is done at all: For example in Figure 5 the Susah Cave (located at 33N/22E close to the Mediterranean) is labeled with NW Africa, and a Bahamas cave with the Cariaco Basin. The Cariaco Basin is under the influence of the ITCZ, the Bahamas are not. These are different systems and thousands of km apart and do not necessarily anything to do with each other. The power of a compilation of high-resolution aridity records is that we may understand the regional response of the climate system to specific perturbations or forcings. Here, this useful information is compromised through an unreasonable combination of records from different systems and a very broad definition of time slices (see below).

→ This paper emerges from the PalMod project, that is why it is presented in this special issue of CP, which belongs to the project. One prerequisite of PalMod was to work with publically available datasets only – from the most cited papers. We see that these archives are climatically not fully homogeneous, but drastic changes should be visible within the archives of one region. We had to choose the most complete, highly cited and well dated (back to 60 000 yr b2k) records with highest sample resolution, which are available for the chosen regions. The reality is that this has been the best possible approach to summarize the regions in as small and detailed a way as possible with publically available data.

6) The used LGM definition (24 to 14.7 ky) is very unfortunate and should be revised. The LGM has been previously defined to extend from 23 to 19 ka (Mix et al. 2001, Quat. Sci. Rev., 20, 627-657). This time interval has been chosen, because the climate is comparably stable. The LGM definition of the authors, however, merges the actual LGM with Heinrich Stadial 1, during which the climate system was exposed to significant changes in external forcings and internal perturbations. The global deglacial warming starts at about 18.5 with the onset of HS1 shortly before the deglacial increase in atmospheric $CO_2$ (Shakun et al. 2012, Nature, 484, 49-54). The distribution of orbital insolation changes significantly and we see a change from a relatively strong AMOC to a weak AMOC with the onset of HS1 (McManus et al. 2004, Nature, 428, 834-837). Very likely, even the deglaciation of the Southern parts of the Ice sheets starts already during HS1 as evidenced by records related to river discharge at some of the more southerly locations (i.e. Menot et al. 2006, Science, 313, 1623-1625).

→ Reviewer#2 also mentioned the misleading LGM definition, we rephrased the sentence to (p1, l26): "This is achieved to a large extent for the Holocene, 0 - 11 700 years before 2000 CE (yr b2k), but mechanisms operating during the MIS2 (24 000 - 12 500 yr b2k) or the flickering climate of MIS3 (60 000 – 24 000 yr b2k) are not fully understood." The LGM definition (now: p2,l31) is as follows: "Mix et al. (2001) define the LGM from comparably stable conditions to last during the time interval of 23 000 to 19 000 yr b2k. Clark et al. (2009) define the LGM from maximum ice sheet extend and sea level low stand to 26 500 to 19 000 yr b2k for most parts of northern and southern hemisphere. We follow the wider definition of Clark, which encompasses the regional differences in the results of this work."

7) Comparison to model experiments. In my view, a comparison to model experiments only makes sense, if there is a coherency between the changes in boundary conditions applied to the model and those expected for the reconstructed time slices. This is not the case here: The model experiments have been performed with fixed boundary conditions. By contrast the definitions of the time slices (LGM: 24 000-14 700 yr b2k, MIS3: 60 000 – 24 000 yr b2k) are so broad that huge changes in boundary conditions and perturbations are present within each time slice. Hence it is impossible to pin down potential reason or mechanisms for the changes. The authors have done an effort to specifically compile high-resolution records and yet they lose all the information through unreasonable broad time slice definitions.

→ Unfortunately, no time transient model experiments were finished for MIS3 right now. Palmod for example is trying to fulfil this right now. As we mentioned, the models represent timeslices (see Tab.3, 42 ka and 32 ka) in comparison to LGM or PI and were accounted to be representative for this period.

More specific points:

-p1/l11: "In comparison, the MIS2 interval becomes arid in all northern hemisphere records, but the peak arid conditions of the Last Glacial Maximum (LGM) differ in duration and intensity among regions." This is not true. MIS2 includes the B/A interval which is clearly very humid. Peak arid conditions in much of the northern Hemisphere tropics occur during HS1, which should not be confused with the LGM.

→ We did not use the MIS and LGM definitions in a completely consistent way. We now follow strictly the following definitions, which are now incorporated into the introduction: The boundaries of the MIS have been developed by Imbrie (Imbrie et al., 1984) and Martinson (Martinson et al., 1987) with refinements by Thompson and Goldstein (2006), which we use for this paper. It is the begin of MIS3 at about 60 000 yr b2k and the end at about 24 000 yr b2k. MIS 2 was defined from 24 000 yr b2k to 12 000 yr b2k.

→ We have rephrased the sentence according to your note: "In comparison, most of the MIS2 interval becomes arid in all of the northern hemisphere records, but the peak arid conditions of the Last Glacial Maximum (LGM) and Heinrich event 1 differ in duration and intensity among regions."

-p1/l17: "two focus" must be "two foci" → We agree, it is changed.

-p2/l13: "We present the 10 key regions…" Key for what? Many important "key" regions of global importance (i.e. the Amazon) are missing

→ Unfortunately, no global data coverage with the prerequisites of the PalMod Project were available. We mentioned this in the revised Method section in detail as well as in the introduction.

P2/l5 of the revised manuscript: "We have screened published paleoclimate literature of the last 30 years to detect and select 10 key areas, for which enough information from various lines of evidence is available to bring the information about past aridity to a synthesis. We define these key areas by the proxy availability, i.e. pollen, dust and speleothem growth must provide three independent sources of information related to past precipitation. These areas were selected because they were the smallest possible regions meeting the criteria set out in the methods chapter."

See as well the reply to your comment 5)

-p4/l1: "The global climate structure is well documented within Greenland and Antarctica ice cores". I disagree with this statement. Ice cores represent the high latitudes. There is very little info about the tropics and subtropics, i.e. the strength of the monsoons, neoglaciation etc.

→ Indeed, NGRIP represents Northern Hemisphere while Antarctica represents Southern Hemisphere. But many authors see evidence for atmospheric teleconnections for example between Arabian Sea and Greenland (for example Bjerknes, 1969; Pourmand et al., 2004; Markle et al., 2017; Sirocko et al., 1996; Zhou et al., 1999). Lots of records show GI / GS signals, even in tropics or subtropics and hence an underlaying influence on several areas. Also, the majority of NGRIP-dust is expected to come from east Asian deserts (Mayewski et al., 2014; Steffensen et al., 2008 or comments of referee#1), located around the 40th latitude.

-P5/l14: "Central Europe is one of the large feedback regions to North Atlantic climate changes" Do the authors mean that Central Europe is amplifying North Atlantic climate changes?

→ We rephrased this sentence to: "Central Europe is related strongly to North Atlantic climate changes".

---

## Author Comment (AC4) · 18 Dec 2019

→ We thank anonymous Referee#4 for his constructive and helpful review of our manuscript. All of your suggestions were answered in red within the text below.

Dear Authors,

You provide a manuscript attempting to synthesize global aridity. You select several key regions with a decent data coverage from different geoarchives. Having read your manuscript and the discussion up to date, I have a clear opinion about your manuscript.

Your conceptual idea of using suites of geoarchives to address aridity is in my opinion great and clearly worth investigating and publishing. At the same time I hold the opinion that several aspects need some work before publication. I agree with most points of other reviewers, see also comments below.

My main comments are: You mention that you focus on openly available data in Supplements to papers. The ELSA vegetation stack data is available in the Pangaea database – using also the NOAA and PANGAEA databases as source would have been appropriate. Please add a Table in Supplements where data are from (websites/databases). You screened 'about 2000 papers' – that is not a reproducible statement. Please ensure that your data processing is 100% transparent and reproducible. Yet I have only a decent idea how this was done. If necessary, please provide sheets and computer code in Supplements.

→ No, we did focus on data from Pangaea and NOAA-NCDC databases. In addition, we used global pollen database (Neotoma), ice core database from Copenhagen university, SISAL speleothem database and European pollen database (EPD). Most speleothem data were taken from tables of the original papers, the other data were downloaded from the mentioned databases. This is explained more clearly in the revised method section. Nevertheless, we agree on your comment and have rephrased the sentence to: "We have screened published paleoclimate literature of the last 30 years to detect and select 10 key areas…"

Uncertainty of the aridity index seems constant with time and data resolution – that clearly does not make sense. Please adjust your method of uncertainty estimation to be at least more realistic. An idea may be to use a relative reliability index, where both lowest data resolution and highest data uncertainty play a role. You do mention that different age models will have an impact on your results. It would be nice to get an idea how this impacts results in one example, but I do see that this is difficult.

→ Differences in data resolution was accounted for in our initial error estimates. For time variance we guessed the largest error for a dataset and assumed it as error for the full set, knowingly overestimating the error of younger data. To give a more detailed error development over time, we would need more information on used methods and how the probes were sampled. Those information are not available for most of the used datasets.

For Asia and Europe, more than single dust records are available in databases – please synthesize these. The presented data selection seems biased towards the authors' work, and I suggest to compile data for several regions in a more extensive way, and maybe focus on less regions. Obvious questions

are, why are data from Tenaghi Phillipon and more Mediterranean cores not used? Why is there only 1 dust record from Asia and Europe? More are available.

→ This paper emerges from the PalMod project, that is why it is presented in this special issue of CP, which belongs to the project. One prerequisite of PalMod was to work with publically available datasets only – from the most cited papers. We had to choose the most complete, highly cited and well dated (back to 60 000 yr b2k) records with highest sample resolution, which are available for the chosen regions. Other records do not necessarily cover the same time phases and not that much records were publically available. Therefore, we have chosen the most relying records we are aware of. For example, Tenaghi Philippon data in Pangaea and EPD only cover the late holocene, although several paper present longer time series (e.g. Pross et al., 2007, 2015; Glais et al., 2016; etc.).

The data selection for several regions is problematic in my opinion: Southern Europe: Data from the Lac du Bouchet is in my opinion hardly comparable to the Portuguese Margin – two datasets from the Portuguese Margin are probably leading to a location bias here, too. This should in my opinion at least be discussed. Why are SST data from the Mediterranean not included? Why are loess data from Spain neglected? Cariaco Basin: the dust record here may actually not reflecting local dust, but African aridity (also discussed in the reference you cite) – please be more self-critical in the discussion.

→ For southern Europe, close dust records are missing. Nussloch loess profile data are not publicly accessible, the same is the case for the Spanish loess data addressed by you. Hence we had to choose the closest, reliably dated record.

→ Several other proxy data, like SST, tree rings, varve thickness, lake or sea levels etc. are not available for each region, therefore we have only chosen speleothem growth, pollen and dust.

→ Cariaco Basin Al/Ti ratio of core 1002C is controlled by fluvial input (which is obviously a local component You are right that the dust source is Africa, but the dust sources are usually further away. For example, the Asian desert areas Gobi, Taklamakan etc. are the main dust sources in the NGRIP ice core. The dust of the NGRIP core is also interpreted as a regional signal of the North Atlantic, as well as of the East Asian monsoon region (e.g. Ruth et al., 2007).

More drastically, data from New Zealand and Australia probably do not indicate the same climate system at all – combining these at least requires a more sensitive discussion. In my opinion these should not be combined for an aridity analysis.

→ We refer to our previous reply to Referee#3: We see that these archives are climatically not fully homogeneous, but drastic changes should be visible within the archives of one region. We had to choose the most complete, highly cited and well dated (back to 60 000 yr b2k) records with highest sample resolution, which are available for the chosen regions. The reality is that this has been the best possible approach to summarize the regions in as small and detailed a way as possible with publically available data.

More detailed comments are:

Please avoid abbreviations in the abstract

→ Many other paper also use abbreviations in the abstracts, for example Clark et al., 2009 or Mix et al. 2001, to name just two. By using three abbreviations (LGM, MIS, GCM) we can save about 70 characters in our abstract.

The first sentence of the introduction is in my opinion not generally true.

→ We have included a "main" before the foci: "Paleoclimate research today has two main foci:"

Unfortunately, in the area of paleoclimate research, not so many other topics are funded today.

You begin with your own data – OK, a scientific reasoning is more appropriate.

→ Rephrased to: "We start the synthesis with Central Europe:…" The scientific reason is given afterwards: p.3/l10ff:

"The maar sediment cores of the Eifel Laminated Sediment Archive (ELSA)-project (Sirocko, 2016; Sirocko et al., 2016) show all Greenland Stadials (GS) and Greenland Interstadials (GI) in the time series of eolian dust content (Dietrich and Sirocko, 2011; Seelos et al., 2009). Central Europe shows accordingly the same climatic structures, which is well known in North Atlantic marine sediments (e.g. Hodell et al., 2013; McManus et al., 1994; Naafs et al., 2013) and Greenland ice cores (North Greenland Ice Core Project Members et al., 2004; Rasmussen et al., 2014; Svensson et al., 2008)."

Page 8, line 9: You mention geographic regions and China as country – please avoid such political statements → The text passage you cited lists the key regions covered in this paper with arid LGM conditions. Since one of these regions is China, which only includes records from China, it is also named so.

Page 9, Line 11f: this is not a result, but more speculation → Therefore, we have deleted this sentence according to review#2 as well.

---

## Author Response (AR2)

Dear Authors,
many thanks for your revisions.

In the replies to all four reviewers you mention the requirements of the PalMod Project. Apparently all reviewers consider these requirements difficult, and suggest other and scientifically more useful data selection. This should make you think how useful these constraints are, and if you can really derive robust aridity for all regions with these constraints. These constraints are clearly impacting on the robustness of your results (negatively, according to four reviewers), and I do think that this needs to be openly stated in the manuscript.

Dear reviewer,
Thanks again for your comments. We have revised the manuscript according to the suggestions. Our replies are marked in red as follows.

I do see the synthesis as an important contribution, and I do see that you carefully thought about how to combine information from different geoarchives.

I do see that you are trying to improve your manuscript, but when you are unable to make your data basis wider and follow reviewers' suggestions, please discuss how the PalMod requirements impact on the robustness of your result – this is yet missing in the discussing part. Generally, I have the impression that you neglect reviewers' well-meant suggestions in some cases without need.

→ We do not neglect your suggestions or those of other reviewers. We have tried to implement as much of it as possible - but we are also bound by the requirements of PalMod.
→ A section on the PalMod requirements impacting the synthesis is added to the discussion.

Before publication I expect that the discussion needs to be more self-critical and needs to include statements for all regions how the PalMod requirements impact on/bias your overall result and the aridity synthesis. Without this, the paper should not be published in Climate of the past in my opinion.

→ We have added this to the discussion. But since we do not use more than one pollen and dust dataset for a region, we do not estimate the bias to be that large. We fully agree, that the requirements strongly influence the region selection, as regions with only one record type are excluded.
Chapter 4.1 compares our aridity index with a previous reconstruction from Herzschuh (2006). This reconstruction is based on various pollen records, but only with that one kind of proxy, which is pollen. Both methods reveal similar results, although Herzschuh has used a lot more records. We therefore assume that our results are representative.

Further, in the last reviews you were asked you to explain your methodology in a way that is 100% reproducible – from data to result. This still is not at all the case. As science should be reproducible, this is not acceptable as it is at the moment. I really demand from you to provide methods in detail and name where required software is available – possibly in Supplements. For speleothems you consider AGE uncertainty, for the other proxies you 'estimate' PROXY uncertainty – but you state that you use data on their original age models. This seems unlogical to mix. Please elaborate how you derive proxy uncertainty in a reproducible way.

→ We have rephrased the methodology section and additionally, provide the Matlab code and our excel template as supplementary materials.

We were asked to provide uncertainties for the aridity index. However, the aridity index is based on data sets from primary literature, which do not contain any errors in the original. Possible errors could be, however, that e.g. the individual pollen counters have miscounted. Measurement inaccuracies in dust data are also possible, for example due to the devices used. In order to be able to give at least rough error values, we have followed a method of Mudelsee (pers. communication) and Kohler (2009, as already cited in the paper). The input parameters from Table 2 for pollen and dust are all estimated - based on the data density and the method used by the author of the data set. This is now described in more detail.

Two reviewers mention that some of your your data selections are difficult or not useful for a local compilation. Please take these concerns serious and do NOT use the Palod requirements as reason to compare not-comparable datasets. These don't force you to compare different things. Please don't use the data from New Zealand and Australia together, I seriously think that mixing these is scientifically misleading. Possibly this reduces the number of regions – but that is in my opinion better than producing misleading results. The Lynch Crater is influenced by monsoons, NZ by westerlies. Although these may have similar temperature and precipitation, that is not comparable.

→ According to your and the other reviewers suggestions, we have deleted the two regions Cariaco Basin and Oceania.

'China' is a political statement, not a geographic one – what about 'east Asia'?

→ Many paper also define this region as "China". For example, see citations in our manuscript from Mingram et al., 2018; Wang et al., 2001; Stebich et al., 2015, Herzschuh, 2006; or Uno, I., Eguchi, K., Yumimoto, K. *et al*. Asian dust transported one full circuit around the globe. *Nature Geosci* **2,** 557–560 (2009); K.E. Kohfeld, S.P. Harrison, Glacial-interglacial changes in dust deposition on the Chinese Loess Plateau, Quaternary Science Reviews (2003); and many more.

In our opinion, the region should not be called East Asia, as East Asia is a much more inhomogeneous region. It includes many more countries than just China. Nor are we aware of any territorial conflict in the region of Jingyuan Loess, Hulu Cave or Sihailongwan Maar, which would require an "apolitical" statement.

Nevertheless we leave this decision to the editor.

Anonymous Referee #1

Dear Reviewer,
Thanks again for your helpful suggestions. We have changed the manuscript accordingly. Our answers to your comments are marked in red color again within this reply.

The authors indeed made some efforts to improve the manuscript from the first version. However, there are several aspects from my comments that in fact still remain unanswered, in particular concerning the methodology. Therefore I still recommend major revisions to the manuscript. This includes passages with typos and poor use of the English language.
→ We went through the whole manuscript in detail and revised the language. If this does not seem sufficient and the editor considers the manuscript appropriate, we will assign a professional proofreader to make the very last minor revisions.

SITE SELECTION
You now explain more clearly the PalMod requirements you stuck to for site selection, in particular the requirement for the datasets to be available in publicly accessible databases, and that they cover the last 60000 years. I recommend explaining also what the criteria were in terms of "high sampling resolution" and "highly cited".
→ high cited papers: more than 5 citation per year
→ high sample resolution: more than 1 sample per 250 years
→ In the beginning we looked for literature with more than 5 citations per year. But since we could not find any data for many papers, we searched the databases afterwards and in a first step we made sure that the records cover as much as possible of the last 60 000 years. From this selection, we chose those records which had a better data resolution than 1 sample per 250 years. Only the pollen record for NW-Africa does not fall under this criterion, but it is the best resolved long record publicly available for the region. Apart from that, we are not aware of any other pollen record, that covers the last 60 000 years in that region.
However, this was not possible for Speleothem data, as they do not exclusively grow continuously. Therefore, we also considered speleothem data with more than 250 years between two samples.
For clarity we have deleted the subordinate clauses " from the most cited papers" and "highly cited" as they were not necessary to anyone outside the PalMod project.

ARIDITY INDEX
You now try to better explain how the aridity index is calculated. You say it's the sum of the three scores from the individual (speleothem, pollen, dust) proxy records, each one first scaled to the 0-100% interval, and then discretized as follows: 0/1 for speleothems, and 0/1/2 for pollen and dust (0-33%, 33-66%, 66-100%), resulting in an overall aridity index score between 0 and 5 (e.g. Table 1). Yet, in Figures 2, 3, etc. you have aridity index time series with non-integer values. This is clearly inconsistent with the definition you gave of the aridity index. Aren't you by any chance linearly rescaling the 0-100% dust and pollen time series between 0 and 2?
→ The reconstructed aridity indices are smoothed with a 200-year running average (see P6/ L17) to better display the underlying structures. The actual reconstructed values are integer values, see new Supplementary material. We made this more clear and rephrased it:
"The generated error estimations are displayed in Fig. 2 and Figs. S1-S9 by grey colour shades behind the mean data. The actual reconstructed aridity index values are integer ones (see

Supplementary table) but are displayed with a 200-year running average to better illustrate the basic structures."

I also do not understand how you merge the 3 individual time series, each one representing the "aridity score". The three time series have different temporal resolution. Please explain how this is dealt with.
→ See P5/L11, but we have rephrased it for more clearness to: "Each data set of speleothem growth phases, tree pollen and dust proxies was resampled by linear interpolation to 50 years resolution."

In connection to the point just above, how do you treat "no data" versus "0 or hiatus"? For instance, in the ELSA dust stack (Fig. 2e) you see no values for the last ~3000 years. Is this because there is no dust or no dated material?
→ This particular example is a display error due to the scaling of the x-axis of the ELSA Dust Stack (2009). The values of the last 3000 years are below 3. Therefore, we have replaced Fig. 2e accordingly.
→ No data vs. hiatus is not easy to distinguish, especially in speleothems. For this reason, speleothems were divided into two parts instead of three for the aridity index. The aridity index is more robust when all proxies have similar patterns and especially when all proxies are present. This is the case, for example, for central Europe or China, so the estimated errors are smaller there.

The Central-European aridity index plotted in Fig. 2f is different (truncated) than in Fig. 3a.
→ Thank you for this suggestion. We have replaced Figure 3, where the last 3000 years were mistakenly cut off.

You claim to be using one and only one record, per type of proxy, per region (with the exception of Santa Barbara basin), but in fact I still see two speleothem records in Figure 2. Please try to be consistent.
In connection to the point just above, you have also not explained how you deal with multiple records of the same type within each region. For instance, for Central Europe I still see both Bunker Cave and Spannagel Cave (Fig. 2a,b). Which one is used for calculating the Central Europe speleothem aridity score? If you used both, please justify why and explain how you merged the two records, i.e. based on AND / OR logical operator.
→ We try to be more clear now:
We use only one kind of proxy for dust and vegetation, because there are few and these proxies are very difficult to combine. For speleothems we use several records for the regions central Europe, Arabian Sea, Mediterranean Sea and St. Barbara basin. Since we only use age dating for Speleothem records, they can be easily combined. In addition, the caves are close to each other or have already been compared in previous work:
Central Europe: The speleothems of the Spannagel and Bunker Cave show very similar growth patterns and can be combined (Fohlmeister et al., 2012).
Arabian Sea speleothems from Oman caves and Socotra Island can be compared (Fleitmann et al., 2007; Shakun et al., 2007).
Mediterranean Sea region, Dim Cave and Soreq cave can be compared (Ünal-Imer et al., 2015).
St.Barbara basin: Cave of the Bells and Fort Stanton speleothems show the same climate signals and can both be compared with the St. Barbara basin.
This section is added within the methodology part 2.1.

UNCERTAINTIES
→ The whole section is revised to be more precise:
We used the initial error values as displayed in Table 2. We estimated these based on the data density and the method originally used. These estimates are based on the experience of the ELSA pollen records (Sirocko et al., 2016). Errors for pollen and dust values result in estimated errors of the aridity index on the X- and Y-axis, since time errors must also be estimated. The age errors of the speleothems, however, result in an error on the Y-axis. Counting errors for pollen were considered very small, as the original investigators are very experienced. In addition, the sample rate of at least one sample every 250 years is high enough to smooth out minor errors. In our experience, the measurement inaccuracies of the devices are around 2%. We have therefore taken this value as a minimum measure for the dust error values. Furthermore, there are possible age uncertainties, which become more important for records with smaller sample intervals. Speleothem age errors were given in the original data sources. All speleothem age errors in the speleothem growth data we used for this synthesis were below 4% uncertainty.
To calculate a total error, we have randomly disturbed the original data with a probability given by the error estimates. From the perturbed data we calculated a perturbed aridity index as described in Table 1 and Chapter 2.1. The variance over 100 000 runs indicates the approximate error of our aridity index. This error simulation is based on the method of Koehler et al (2009) and personal communication with M. Mudelsee.
The generated error estimations are displayed in Fig. 2 and Figs. S1-S9 by grey colour shades behind the mean data. The actual reconstructed aridity index values are integer ones (see Table S2 in the Supplement) but are smoothed with a 200-year running average to better display the underlying structures. Smoothed aridity index values below 1.5 account for arid conditions, values between 1.5 and 3.5 show intermediate aridity and values larger then 3.5 show more humid conditions (see Fig. 4).

This section is still too generic. What is an error for pollen and dust? (You use all kinds of dust proxies by the way, and I am not sure how you deal with the different cases). Mentioning that you learn from the experience of the ELSA stack is just not enough. Okay the description of the Montecarlo procedure, but where do you get the perturbations from in the first place? This needs to be spelled out clearly, and the actual values/ranges of these perturbations properly justified. → See reply to Uncertainties.

The uncertainties in Table 2 are expressed as percentages. With respect to what? Age or value of the proxy variable (i.e. x or y axis)? To the actual value of the aridity index in the 0-1 or 0-2 range for the individual proxies?
→ The percentages refer to the original values of the data. This means that all values are considered for example as ± 2% → Dust concentration of 42% ±2% estimated uncertainty.

How do you combine these individual uncertainties into the overall aridity index uncertainty (the grey shaded area)? → See reply to Uncertainties in general

RESULTS
In this section (and the corresponding supplementary sections for the different regions) you could indeed introduce in the discussion the relevant records that are not used for the calculation of the aridity index, for instance the records of Nussloch and Dunaszecsko you

cited, and compare qualitatively with the records you actually used for the calculation of the aridity index.

→ We have added a paragraph about this to the discussion, according to your suggestion.

Anonymous Referee#4

Review of revised paper by Fuhrmann et al. "Aridity synthesis for 10 selected key regions of the global climate system during the last 60 000 years"
Thanks again for your helpful suggestions. We have revised the manuscript according to your suggestions. Our answers to your comments are marked in red.

The manuscript by Fuhrmann et al. has been substantially improved. However, there are a few remaining issues that I strongly recommend to revise:

-P1/L26: "This paper emerges from the PalMod project which develops a long GCM time series of past global temperatures...". PalMod is doing much more than just a long GCM time series of past temperatures!
→ changed to „This paper emerges from the PalMod project, which among other things is developing a long GCM time series of past global temperatures (www.palmod.de)."

-P1/L27: "One prerequisite of the project was to work only with publically available datasets from the most cited papers". I doubt that "most cited" was a PalMod prerequisite. Personally, I also think that this is a very questionable criterion. Generally, the number of citations is not a faithful quality indicator.
Comment of F.Sirocko:
This was indeed not a prerequisite of the Palmod programm, but was discussed with the group leader of WG3 to be a good aproach, when we realised how little published data are accessible indeed. It was decided in the first WG3 meeting at Mainz by F.Sirocko and M.Kucera.
The data mining prerequiste was insisted on by M.Kucera and S.Mulitza, even if we expressed our sincere problems with a 100% data mining aproach several times - from the very beginning of Palmod planning to the proposal for the second funding phase.

→ We agree that the citation number is not an indicator of quality. However, the synthesis is based on publicly available datasets, since data mining with the most cited papers produced very few publicly available datasets. These few datasets have then been sorted by citation for the project. For clarity we have deleted the subordinate clause " from the most cited papers" and we have rephrased the section on methodology to provide more clarity.

-P2/L8: Note that the "...the boundaries of the MIS have been developed by Imbrie et al. (1984) and Martinson et al. (1987)" are outdated. The stacks provided by Lisiecki and Raymo (2005, Paleoceanography, 20, doi:10.1029/2004PA001071) or Lisiecki and Stern (2016, Paleoceanography, 10.1002/2016pa003002) are up to date.
→ We have taken the updated boundaries from Spratt & Lisiecki (2016, CP), as they have refined the MIS 2 borders furthermore.

-P2/L13: What is "highly cited"?
→ In the beginning we looked for literature with more than 5 citations per year. But since we could not find any data for many papers, we searched the databases afterwards and in a first step we made sure that the records cover as much as possible of the last 60 000 years. From this selection, we chose those records which had a better data resolution than 1 sample per 250 years. Only the pollen record for NW-Africa does not fall under this criterion, but it is the

best resolved long record publicly available for the region. Apart from that, we are not aware of any other pollen record, that covers the last 60 000 years in that region.

However, this was not possible for Speleothem data, as they do not exclusively grow continuously. Therefore, we also considered speleothem data with more than 250 years between two samples. For clarity we have deleted the subordinate clauses " from the most cited papers" and "highly cited" as they were not necessary to anyone outside the PalMod project.

-P3/Fig 1: It seems that the location of the Susah Cave is wrong in Fig. 1 (c.f. Rogerson et al., Clim. Past, 15, 1757–1769, https://doi.org/10.5194/cp-15-1757-2019.). Also, it seems that Susah Cave is actually in the Mediterranean group? In Fig. 5, Susah Cave is listed under NW-Africa.

→ You are right, there we made a mistake. We wrote down the place wrong and did not notice this. Therefore we removed Susah Cave from the regional synthesis NW-Africa and adjusted the chapter accordingly.

-P4/L19: "The global climate evolution is well documented within Greenland and Antarctica ice cores". I still disagree. For example, tropical precipitation is not represented in ice cores.
→ according to your suggestion, we have deleted the sentence.

-P5/L3: "In sediments from the Cariaco Basin, the Al/Ti ratio gives the proportion between terrigenous river sediments with higher Al/Ti ratios and Saharan dust with respective lower Al/Ti ratios." Please back this statement up by a citation or robust evidence. Yes, there is some Saharan dust arriving in the Caribbean, but this is negligible compared to river input, and I doubt that the dust signal is visible in the Cariaco Basin.
→ The reference was directly after the second sentence: "In sediments from the Cariaco Basin, the Al/Ti ratio gives the proportion between terrigenous river sediments with higher Al/Ti ratios and Saharan dust with respective lower Al/Ti ratios. Ratio of 14 represent pure Saharan dust (Yarincik et al., 2000)." But according to your and another referees suggestions, we have deleted Cariaco Basin and Oceania as regions, so you can't find this section anymore.

-P7/L25: "The Atlantic sea surface temperature pattern (caused by the Atlantic meridional overturning circulation - AMOC) strongly influence the whole European continent today". Please revise this sentence. The Atlantic sea surface temperature pattern is not only caused by the AMOC.
→ We agree, that AMOC is not the only cause. We have accordingly deleted this.

-P7/L28/29; "... of central ??..", "...with varved ??"
→ An established geoarchive to reconstruct the climate of central Europe are the volcanic maar lakes of the Eifel, which cover the Holocene with varves (…).

-P10/Fig. 3, P13/Fig. 5: As already argued in my first review I still find it very unreasonable to merge the Bahamas cave with the records from the Cariaco Basin and potentially (see comment on Fig 5 above) the Susah Cave (located at 33N/22E close to the Mediterranean) with NW African records. The prerequisites from PalMod cannot be relevant for my assessment of this paper.
→ According to your and the other referees suggestions, we have deleted Cariaco Basin and Oceania as regions. Susah Cave has been removed from NW-Africa.

[revised manuscript text omitted]

**Supplements:**

**S1 Arabian Sea:**

10  The Arabian Sea comprises the region from the Persian Gulf to the Indian Sea and is characterized by warm and high saline waters and fluvial input from the Indus River. High dust fluxes mainly from Arabian desert are preserved in the sediments. The high surface-water productivity from monsoonal inputs into the ocean and upwelling offshore west Pakistan lead to a stable oxygen minimum zone (OMZ) in water depths between 200 m and 1200 m. This OMZ results in excellent preservation conditions for dark, organic-carbon rich, laminated sediments during mild interstadials and in contrast

15  to light colored, bioturbated sediments during stadials and especially Heinrich events (Schulz, et al., 1998). Arabian Sea and North Atlantic regions are closely coupled by atmospheric teleconnections (Burns et al., 2003; Deplazes et al., 2014; Leuschner and Sirocko, 2000, 2003; Schulz, et al., 1998; Sirocko et al., 1996a and others). There is evidence for a general relationship between these two regions on timescales of the last 110 000 years within low-latitude monsoonal variability and high northern latitude records of Greenland ice cores (Schulz, et al., 1998). The sediment cores SO130-289KL and SO90-136KL are from

20  very close positions and show nearly the same pattern within Reflectance and Total Organic Carbon (TOC) content. Furthermore, they can be correlated one-to-one to the NGRIP ice core (North Greenland Ice Core Project Members et al., 2004) on every GI from 17 to 1 and the YD cold event. In Addition, the Heinrich events 6 to 1 (H6 - H1) appear in superposition (see Fig. S1). The speleothem growth in this region can be correlated to the Greenland ice cores as well (Burns et al., 2003). Speleothems from Oman and Socotra Cave in Yemen are very close to the Arabian Sea and hence used for this synthesis. The

25  sediment core 70KL shows $CaCO_3$ content in percent as a dust indicator. High $CaCO_3$ values show low dust contents from Arabian Peninsula desert and vice versa – more dust accounts for lower $CaCO_3$ values due to higher dilution of the sediment because of increased sedimentation rates.

From 60 000 to 55 000 yr b2k no speleothem growth is apparent for the Arabian sea region while the dust values show a maximum. The Reflectance data simultaneously show very high values (according to bioturbation) indicating the extend of

30  H6 to this region. The TOC content is on small values resulting from only slight upwelling and low bioproduction but still all

GIs are visible with less expression compared to the NGRIP ice core. The dust reconstruction from $CaCO_3$ content shows high amounts in dust values. A high aridity is reconstructed for this period.

From 55 000 to 41 800 yr b2k, the GIs 14 to 11 are visible in the TOC. High variations between stadial and interstadial times occur with the highest values of about 5% TOC during GI14 and 12. The lowest values go along with H5, but in general, stadials account for lower TOC values. In the time of 55 000 to 41 800 yr b2k, the mean TOC values were on the highest values for the last 60 000 years. Also, the Reflectance data show this pattern with clearly visible H5 and GI expression. Within the dust content, a minimum during this period is apparent (lowest values beside the Holocene) and speleothem growth occurred in the Socotra Cave during this early MIS3 time indicating high precipitation and strong monsoonal variability resulting in low aridity as visible in the aridity index (see Fig. S1e).

From 41 800 to 27 700 yr b2k the GIs 10 to 3 are present within Reflectance and TOC data and comparable to the NGRIP ice core. Intermediate TOC values and the large differences between interstadial and stadial, especially at H4 and H3  times, are remarkable. The dust content varies between high and medium values through this period with higher values around 35 000 yr b2k and lower values during GI5. This interstadial seems to be nearly as strong as GI12, which is known as one of the 'warmest interstadials' for several regions within TOC and Reflectance data. No speleothem growth is observed during the high glacial period. The aridity was high at the beginning of this phase. GI5 and the double GI3 and 4 appear to have had strong impact on the precipitation, so the aridity was lower during this time with an increase to stronger aridity afterwards. Between 27 700 and 11 700 yr b2k are GI2 and 1 as well as H2 and H1 and the YD apparent. The TOC decreases from high values at the end of GI3 to very low ones during LGM and especially H1 and YD. The dust remains on intermediate to high values with dust pulses within H1 and H2. These events increased aridity to very high values but the aridity within the rest of this time phase was high nevertheless.

With the onset of the Holocene at 11 700 yr b2k, speleothem growth in Oman caves started. The dust values decrease to minimum during the Holocene climate optimum around 8 000 to 6 000 yr b2k. The Reflectance and TOC data increase drastically and show higher temperature as well during early Holocene, with a slight decrease towards present day. During early Holocene times the precipitation seems very strong with low aridity. Towards present day, the aridity increases strongly.

[Figure]

[Figure]

**Figure S1:** Arabian Sea climate over the last 60 000 years: **(a)** Socotra Cave and **(b)** Oman Caves (Burns et al., 2003; Fleitmann et al., 2007) show speleothem growth phases, which require mobile water from frequent precipitation; **(c)** 70KL CaCO₃ (Leuschner and Sirocko, 2003) indicates more arid conditions with lower values, higher values account for more humid conditions; **(d)** SO90-136KL (Schulz, et al., 1998) TOC content exhibits the GIs comparable to **(g)** in total by higher carbon values; **(e)** Aridity index for Central Europe as result from **(a-d)**, for detailed information see method section; **(f)** SO130-289KL Reflectance data (Deplazes et al., 2014) resembling **(d)** and **(g)**; **(g)** δ¹⁸O data from NGRIP ice core (North Greenland Ice Core Project Members et al., 2004) in comparison.

**S2 North-West Africa:**

To understand the climate history of North-West Africa and the continental margin off West Africa is important for understanding changes in Sahara – Sahel aridification as induced by changes in Atlantic water sea surface temperatures (SST) or large scale mechanisms like for example AMOC, North Atlantic Oscillation (NAO), Inter Tropical Convergence Zone (ITCZ). Life in this region strongly depends on the availability of water. Nowadays, the mean annual temperature ranges between 12 and 15 °C and mean annual precipitation is about 468 mm/yr for Atlas Mountain ranges related to the Azores high position (Wassenburg et al., 2012). Sediment core GIK15627-3 from offshore Morocco reveals the time from 250 000 to 5 000 years b2k of paleovegetation for NW-Africa (Hooghiemstra et al., 1992). No long pollen time series are available from terrestrial archives right now, hence this record was chosen despite a relatively low sample resolution. Nearby speleothems are

known from Atlas mountain range like Grotte Prison de Chien (Wassenburg et al., 2012) or Grotte de Piste (Wassenburg et al., 2016). GeoB9508-5, a sediment core from offshore Senegal, West

5   Africa,  (Collins et al., 2013; Mulitza et al., 2008) reveals strong Heinrich Stadials in dust content during times of reduced AMOC (Mulitza et al., 2008; see Fig. S2).

The timespan from 60 000 to 50 000 yr b2k comprises GI17— - 14 . No Speleothem growth

10  is apparent during the whole timespan . The amount of tree pollen decreases from values around 60 % to 0 % (53 000 yr b2k) and rises again during a wet period from 52 500 to 50 500 yr b2k. The dust content shows no major fluctuations during this phase and remains at its lowest levels until the beginning of the Holocene. Low dust values and

15  varying tree pollen amounts, the aridity is on intermediate to lower values. Between 50 000 and 37 000 yr b2k with GI13 – 8, aridity is increased. A first speleothem age is known to be around 45 000 yr b2k, shortly after the end of H5. Tree pollen amount is constantly on lower values (~ 30 %) with a little decrease during H5. Dust values peak strongly during Heinrich Stadials 5 and 4 indicating intermediate to high

20  aridity. From 37 000 to 27 000 yr b2k (GIs 7-3) NW-Africa underwent increasing aridity. Speleothem growth was only sporadic but comes along with higher tree pollen values for the same times. Tree pollen remain on lower values between 20 and 30 % with a small increase during the

25  growth phases known from the speleothem. Dust content is on intermediate values with a small increase during H3 at the end of this period. The climate of this time phase appears cold and moderate arid. The onset to H2 marks the begin of the next period (27 000 to 14 800 yr b2k, GI2 within).  Grotte Prison de Chien shows at least  some dating indicating sporadic precipitation for this period. In general, tree pollen values decrease to very low values indicating even more arid conditions between 20 000 and 15 000 yr b2k (Hooghiemstra et

30  al., 1992). The dust values are on a maximum during this phase in general but H2 and H1 are clearly identifiably within the record. With all that information combined, the period expresses the impact of the LGM period and shows arid and cold conditions. With the end of H1 and the onset of warming towards the Holocene, the amount of tree pollen increases again (14 800 yr b2k until present, GI1 and YD), while speleothems from North Morocco also show some growth phases. Gradually, less arid conditions show a climate amelioration until 8 500 yr b2k, the 'African Humid Period' (AHP) or

EHTO (Early Holocene Temperature Optimum)), where lots of age datings can be found indicating fast speleothem growth. Dust shows a small peak during YD cold event. However, the general strong decrease from the end of LGM until EHTO is evident. Little dust was mobile during this phase. In the last 4 000 years, dust values rise again, indicating an aridity increase for youngest times.

[Figure]

[Figure]

Figure S2: : NW-Africa climate over the last 60 000 years: **(a)** Grotte Prison de Chien (Wassenburg et al., 2012) and Grotte de Prison (Wassenburg et al., 2016) show speleothem growth phases, which require mobile water from frequent precipitation; **(b, c)** GIK15627-3 marine core pollen data (Hooghiemstra et al., 1992) are divided into tree- and herb & grass pollen. While trees require more precipitation, grasses are dominant for more arid conditions; **(eHE are distinguishable by higher dust concentrations; (f) Aridity index for NW-Africa as result from (a)-(e), for detailed information see method section; (g) SST reconstructions from marine core SU81-18 (Bard, 2002) with all HE, and less distinct GIs, apparent; (h~~HE are distinguishable by higher dust concentrations; **(e)** Aridity index for NW-Africa as result from **(a)-(d)**, for detailed information see method section; **(f)** $\delta^{18}O$ data from NGRIP ice core (North Greenland Ice Core Project Members et al., 2004) in comparison.

**S3 China:**

Most of the Asian continent is influenced by the East Asian Monsoon, which is the most important moisture source. Most of annual precipitation (~ 80 %) falls in summer season (e.g. Mingram et al., 2018; Wang et al., 2001) with mean annual

precipitation about 1015 mm/yr at Hulu Cave and 715 mm/yr at Sihailongwan maar lake (Stebich et al., 2015). The mean annual temperature is about 2.9 °C (Schettler et al., 2006), varying from -18.1 °C for January and + 20.7 °C for July at Sihailongwan site and 15.4 °C at Hulu Cave (Wang et al., 2001).

Hulu Cave is one of the most popular east Asian monsoon records through the last glacial cycle. For this record, five speleothems from the cave were stacked together to compile a continuous record for the timespan of 11 000 – 75 000 yr b2k. The long-term trend follows summer insolation pattern, suggesting an increased summer continent-ocean temperature difference and so, enhanced summer monsoon (Wang et al., 2001). The Hulu record shows a link between East Asia Monsoon and North Atlantic climate by apparent GI-variations within the $\delta^{18}O$ record. The Sihailongwan maar lake (SHL) lies within the Long Gang Volcanic Field in NE-China. The SHL-core is continuously warved until 65 000 yr b2k, providing an excellent stratigraphy for climate reconstructions by not only paleovegetation. The tree pollen amount replicates the stadial / interstadial variations of the North Atlantic (see Fig. S3) as well as the total organic carbon. Stadials and especially Heinrich events are characterized by steppic plants like Artemisia. Interstadials in general show higher amounts of tree pollen (Mingram et al., 2018). However, the Holocene pollen data are not publically available for this paper but kept in mind for the discussion of this synthesis. The China Loess Plateau is well known for its loess paleosol sequences. Jingyuan and Weinan sections from Sun et al. (2010) and Lu et al. (2007) are established and show reliable indications for North Atlantic - China climate teleconnections with "loess interstadial / loess stadial" within the mean grainsize of the Jingyuan record.

The timespan from 60 000 yr b2k until 50 000 yr b2k (GI17 - GI13, early MIS3) is regarded as the most humid period of the record (Liu et al., 2010; Lu et al., 2007; Mingram et al., 2018). Speleothem growth is apparent and $\delta^{18}O$ values are on minimum (more negative equals higher temperatures, see (Liu et al., 2010; Wang et al., 2001)) during this period. The tree pollen show high values in order of 60 % with identifiable GI variability. Also, dust values from the Jingyuan loess section show small grainsizes according to a lower dust content. In result, the archives account for humid climate conditions through the early MIS3 phase. Within the period of 50 000 yr b2k to 34 000 yr b2k, GIs 12 to 5 are comprised. This phase also shows continuous speleothem growth with intermediate $\delta^{18}O$ values, GIs are easy to identify. Also, the tree pollen show medium contents with still relative high values during interstadials but lower compared to early MIS3 times. The dust content rises to intermediate values until 38 000 yr b2k indicating stronger winds and lower temperatures combined with a higher aridity. Towards the end of this phase, lower $\delta^{18}O$ values combined with the sharp GI-type increases in tree pollen and lower dust contents are visible, suggesting a phase of stagnation within the climate conditions.

From 34 000 yr b2k to 14 800 yr b2k (GIs 4-2) the climate is characterized by glacial conditions, especially during last glacial maximum and the Heinrich events 3 to 1. The growth rates of the Hulu Cave speleothems went down while the $\delta^{18}O$ values rise to their highest values of the record. Synchronously, the tree pollen decrease to about 20 % during LGM, with high contents of Artemisia especially during Heinrich events, the GIs during this phase are less expressive with general lower temperatures and shorter durations. Between 29 550 yr b2k and 18 250 yr b2k, the minimum of thermophilous plants and tree pollen is apparent (Mingram et al., 2018). The dust values are on a maximum during this time span. The large grainsize comes along

with low temperatures and precipitation values. All that combined is clearly visible in the aridity index with a strong expressed LGM. The deglaciation and the Holocene itself (14 800 yr b2k to present, GI1) can be characterized by the amelioration of the climate conditions, in China as well as on a global scale. The $\delta^{18}$O values decrease while the growth rate increases. The tree pollen rise up to 80 % while synchronously the dust content lessens to the minimum of the record during the Holocene temperature optimum (6 000 yr b2k – 4 000 yr b2k). According to Stebich et al. (2015) the Sihailongwan Maar pollen indicate a maximum precipitation at 4 550 yr b2k (not included into Fig. S3). The aridity decreased during the Holocene transition to intermediate values, but a lack in data for the Holocene period makes it complicated to draw further estimates.

[Figure]

**Figure S3:** China climate over the last 60 000 years: **(a)** Hulu Cave (Liu et al., 2010; Wang et al., 2001) show speleothem growth phases, which require mobile water from frequent precipitation; **(b)** $\delta^{18}$O data with apparent GIs comparable to **(g)**, more negative $\delta^{18}$O values account for more humid conditions (Wang et al., 2001); **(c, d)** Sihailongwan Maar Lake pollen data (Mingram et al., 2018) are divided into tree- and herb & grass pollen. While trees require more precipitation, grasses are dominant for more arid conditions; **(e)** Jingyuan Loess mean grainsize (Sun et al., 2010) indicates more arid conditions with larger grains, smaller grains account for more humid conditions; **(f)** Aridity index for China as result from **(a-e)**, for detailed information see method section; **(g)** $\delta^{18}$O data from NGRIP ice core (North Greenland Ice Core Project Members et al., 2004) in comparison.

**S4 Southern Europe**

Southern Europe is affected by the Atlantic Ocean water masses as well as from the Mediterranean Sea and so, influenced by changes in AMOC, NAO, ITCZ and other large-scale mechanisms. Nowadays, the mean annual temperature close to the Villars Cave (in southern France) speleothem site is 12.1 °C and mean annual precipitation 1020 mm/yr, evenly spread through

5 the whole year (Wainer et al., 2009). A sediment core of volcanic maar lake Lac du Bouchet (Lake Bouchet) shows pollen spectra until the end of the last interglacial (Reille and de Beaulieu, 1988, 1990).  Although 'Nussloch paleosol loess sequence' (Antoine et al., 2001) is well established but publicly available data are not accessible. MD01-2443 marine sediment core (Hodell et al., 2013) is the closest dust archive with sufficient stratigraphy for Southern Europe. SST, L* (color reflectance) and $\delta^{18}$O of the

10 core show GIs (Hodell et al., 2013; Martrat et al., 2007). Well dated speleothem data are available for the Villars Cave, 200 km away from the Atlantic coast, with speleothem growth between 52 000 and 29 000 yr b2k and for the Holocene (see Fig. S4). The growth speed significantly slowed down between 42 000 and 29 000 yr b2k and finally stopped with the onset to LGM conditions. Also, a hiatus from 55 700 to 52 000 yr b2k is present within the record. GIs 14, 13, 12 and on minor extend GI11 are preserved as well as H5

15 ~~(uk'37) record of Martrat et al. (2007) with all Heinrich events as well as from an Ice Raft Debris (IRD) content time series from Naafs et al. (2013) from Central Atlantic derived from Dolomite/Calcite ratio. Higher values show IRD layers and Heinrich events are clearly distinguishable (see Fig. S4). IRD layers consist of coarse-grained lithic clasts from iceberg discharge and low foraminifer contents (Heinrich, 1988). They were climatically interpreted as extreme cooling of the SSTs before Greenland interstadials (Bond et al., 1998).The timespan from 60 000 to 44 000 yr b2k incorporates GIs 17 to 12 within~~

20  (Genty et al., 2003; Wainer et al., 2009).

The timespan from 60 000 to 44 000 yr b2k incorporates GIs 17 to 12 within the records of this region. Speleothem growth in the Villars Cave is evident with a hiatus between 55 700 to 52 000 yr b2k. $\delta^{18}$O values indicate warm and moist conditions with more negative values than during later phases of the record. The GIs 17, 16, 13 and 12 are identifiable within the data. According to Wainer et al. (2009) the temperature optimum of the recorded time phase was during early MIS3 at around

25 52 000 yr b2k. In contrast, the highest amount of tree pollen for this period (with about 50 %) falls within the hiatus of the speleothem but still indicating warm and wet early MIS3 conditions for this region. Also, dust content from marine sediment core MD01-2443 are on relatively low values with minor variations. Their lowest values until the Holocene were during the tree pollen maximum.  Until the end of this time phase

30 towards 45 000 yr b2k, conditions tend to get worse as indicated by rising dust content, decreasing amount of tree pollen, lower $\delta^{18}$O values from Villars Cave and with H5 from the marine cores. The aridity is low during the early MIS3 phase and rises towards the end of the time phase.

The period from 44 000 to 30 000 yr b2k (GIs 11-5) shows general aridity (Reille and de Beaulieu, 1990). The speleothem $\delta^{18}$O values are very low, only GI8 stands out a bit. The speleothem growth was significantly slower and stopped at the end of this time phase. The tree pollen show high variations and high absolute values, but high amount of steppe vegetation and a small remaining tree population complete the in general increased aridity for Southern Europe. The dust values replicate the interstadial / stadial changes with stronger impression on Heinrich events.

From 30 000 to 15 000 yr b2k (GIs 4-2) no speleothem growth is known from Villars Cave. The tree pollen show low values with a minimum around 25 000 yr b2k, were no tree pollen occur within the record. During LGM, grasses (60 to 100 %) replace the last Pinus woodland, which was still apparent at GI3 and 4. Dust values,  show arid conditions and low temperatures, Heinrich events are well expressed, apart from a low variability. Dust values are highest at the end of this period from 17 000 to 14 000 yr b2k during the same time, were Ruth et al. (2007) detected most eolian dust in NGRIP ice cores (cf. Fig. 2 and Fig.  6). The aridity was strongest during LGM especially during the end of this phase. With the onset towards the Holocene around 15 000 yr b2k, speleothem growth restarted in the Villars cave, the amount of tree pollen drastically increases and dust values decrease with a delay of approximately 4 000 years after YD cold event. The Altithermal is displayed by precipitation and temperature increase visible in the records. The aridity decreases with the onset of the Holocene and stays relatively constant on lower values throughout.

[Figure]

[Figure]

**Figure S4:** : Southern European climate over the last 60 000 years: **(a)** Villars Cave (Genty et al., 2003, 2006; Labuhn et al., 2015; Wainer et al., 2009) show speleothem growth phases, which require mobile water from frequent precipitation; **(b)** δ¹⁸O data with few apparent GIs comparable to **(g)**, more negative δ¹⁸O values account for more humid conditions (Genty et al., 2003); **(c, d)** Lac Du Bouchet pollen data (Reille and de Beaulieu, 1990) are divided into tree- and herb & grass pollen. While trees require more precipitation, grasses are dominant for more arid conditions; **(e)** MD01-2443/4 marine cores CaCO₃ (Hodell et al., 2013) ~~indicates more arid conditions with lower values, higher values account for more humid conditions; **(f)** Aridity index for Southern Europe as result from **(a-e)**, for detailed information see method section; **(g)** SST from marine cores MD01-2443/4 (Martrat et al., 2007) resembling HE and most GIs; **(h)** Dol/Cal ration from IODP-306-U1313 (Naafs et al., 2013) show detailed HE structure; **(i)** δ¹⁸O data from NGRIP ice core (North Greenland Ice Core Project Members et al., 2004) in comparison.~~

**S5 Portuguese Margin:**

 indicates more arid conditions with lower values, higher values account for more humid conditions; **(f)** Aridity for Southern Europe as result from **(a-e)**, for detailed information see method section; **(g)** δ¹⁸O data from NGRIP ice core (North Greenland Ice Core Project Members et al., 2004) in comparison.

**S5 Portuguese Margin:**

The Portuguese margins sediment cores are known to be highly impacted by climate change on orbital and millennial timescales. Constant sedimentation rates are responsible for good stratigraphies and they are influenced by high- and low-latitude processes (Hodell et al., 2013). Today's mean annual temperatures are about 15- °C, winters are mild (10- - 13 °C)

5  and summers are moderate (18— - 22- °C) with up to 3 000 mm yearly precipitation. Portuguese margin and Southern Europe contain the same dust record, the MD01-2443 sediment core shows dust in the $CaCO_3$ content. For Portuguese margin, a marine sediment core with terrestrial pollen input can be used to reconstruct the vegetation of the coast. Tree populations of sediment core MD95-2039 show rapid shifts, following GI-scheme. During Heinrich events 1 - 6, SST's dropped in order of 5 – 10 °C.

10   Short GIs show less impact on the extend of woodland (Roucoux et al., 2005). Buraca Gloriosa Cave speleothems show the climate evolution of the last 220 000 years (Denniston et al., 2018). The cave is located 30 km from the Atlantic coast, near

15  the both marine cores used for this region.

Within the timespan from 60 000 to 50 000 yr b2k (GIs 17-13),  humid conditions can be reconstructed for Portuguese Iberian margin. Speleothem growth was continuous in Buraca Gloriosa Cave and shows  high growth rates. The tree pollen in general were on a maximum during this time phase, showing high amplitude fluctuations between cooler and drier stadials and warmer and

20  wetter interstadials (Roucoux et al., 2005), as well as a decline in dust content from 60 000 towards 50 000 yr b2k from relatively high values to lowest values until the Holocene. The high amount of tree pollen, speleothem growth and the decline in the dust content indicates a general warmer and wetter early MIS3 and consequently, low aridity with the begin of the phase and an intermediate aridity towards the end. During the period from 50 000 to 32 000 yr b2k (GIs 12-5) an intermediate aridity is estimated for the region. H5 and H4 are

25  visible within all records indicating a rapid climate change throughout the region. Speleothem growth rates were lower than before . Around H4 a hiatus of 41 000 to 36 000 years b2k in speleothem growth can be seen. Heinrich events 5 and 4 are strongly expressed within the MD95-2039 core indicating a dramatic decrease in climate conditions (Roucoux et al., 2005). A minor rise in

30  dust is visible within the $CaCO_3$ content of MD01-2443 from in general intermediate dust values throughout this period for H5 and H4. Regarding all this, an intermediate aridity is estimated throughout this time phase with strong variability between interstadials and stadials for Portuguese margin region.

Between 32 000 and 15 000 yr b2k (GIs 4-2), the speleothem growth at Buraca Gloriosa Cave.­ shows two hiatus during H3 and H1 (from 32 000 to 30 000 and 17 000 to 15 000). The amount of tree pollen was very low, open, herb-dominated steppic vegetation indicates cool and arid conditions. Less aridity towards the end of the phase is indicated from a slight increase in tree pollen and heath population, which required amelioration in climate (Roucoux et al., 2005).

5  Intermediate dust values with a decrease towards the end are visible, again with stronger impressed Heinrich events compared to the other stadials.  The LGM, which falls into this period, can be characterized by intermediate aridity but surprisingly low dust values towards the end.

The time phase from 15 000 yr b2k (GI1, YD) to present is marked by the onset of the Holocene with Bølling / Allerød. Speleothem growth restarted and remains constant until 10 000 yr b2k, a growth recovery from

10  3 000 to 1 000 yr b2k is also apparent. A rapid increase in tree pollen  around 15 000 yr b2k marks a strong increase in temperature and precipitation. Younger Dryas cold event reduced that for a short duration, but afterwards the climate improved to Holocene and present day conditions. The dust values were highest during the begin of the time phase around 15 000 yr b2k and declines through the early Holocene temperature optimum.  Aridity was very low with the begin of this phase but in the past 5 000 years, dust

15  values increase again indicating higher aridity than during Altithermal.

[Figure]

(a)   (b)   (c)   (d)   (e)   (f)   (g)   (h)   (i)

[Figure]

**Figure S5:**  **(c, d:** Portuguese margin climate over the last 60 000 years: **(a)** Buraca Gloriosa Cave (Denniston et al., 2018) show speleothem growth phases, which require mobile water from frequent precipitation; **(b, c)** MD95-2039 pollen data (Roucoux et al., 2005) are divided into tree- and herb & grass pollen. While trees require more precipitation, grasses are dominant for more arid conditions; **(e)** MD01-2443/4 marine cores CaCO₃ (Hodell et al., 2013) indicates more arid conditions with lower values, higher values account for more humid conditions;  **(i)** Aridity index for Portuguese margin as result from **(a-d)**, for detailed information see method section; **(f)** δ¹⁸O data from NGRIP ice core (North Greenland Ice Core Project Members et al., 2004) in comparison.

**S6 Mediterranean Sea:**

Southern Europe and the Mediterranean Sea region are known for hot and dry summers and mild and wet winters. The temperature average is about 18.9 °C and the total precipitation about 1000 mm/year with more precipitation in the western

regions than in the eastern (Deutscher Wetterdienst, 2018). The archives for this synthesis are spread around the eastern Mediterranean region. The westernmost archive is the Lago Grande di Monticchio, a maar lake in Basilicata, southern Italy. The cores comprise climate information for the last 140 000 years (Allen et al., 1999; Brauer et al., 1999; Watts, 1985 and others) and completely varve counted, supplemented

5    by a tephra chronology. The pollen from Monticchio show similar behaviour to the NGRIP indicating a closely coupled system between North Atlantic and the Mediterranean region (Allen et al., 1999; see Fig. S6). Speleothem growth occurs at several sites. Dim Cave in south western Turkey shows continuous speleothem growth from 12 000 yr b2k until 90 000 yr b2k with significant lower growth rates during glacial times (40 000 – 18 000 yr b2k, Ünal-İmer et al., 2015). A second speleothem from Soreq Cave in Israel shows continuous growth for the last 140 000 years. Two distinct isotopic events can be separated

10   within the record within the last 60 000 years (Bar-Matthews et al., 1997, 2000). The dust record in M40/4_SL71 from SE Ionian Sea shows increased K/C ratio during arid conditions due to deflation of Kaolinite bearing dust, which was sedimented into

15   basins during more humid conditions (Ehrmann et al., 2017). Kaolinite is a common mineral in North African dust and hence a useful dust tracer. During Heinrich events, the Mediterranean region was heavily arid and minor maxima of Kaolinite appear. The timespan from 60 000 to 55 000 yr b2k comprises GIs 17 to 15. Speleothem growth occurs in Dim and Soreq Cave and $\delta^{18}O$ values are on relative highs around 57 000 yr b2k for both caves. The growth rates in Dim cave are lower than in younger parts of the speleothem. The amount of tree pollen is at intermediate values of 50 % and so is the K/C ratio, which shows H6

20   recorded in the sediment core at about 60 000 yr b2k. Overall, the aridity was at intermediate values.
GIs 14 to 9 are within the time of 55 000 to 40 000 yr b2k. The Dim Cave speleothem growth rate rises around 50 000 yr b2k, the $\delta^{18}O$ values sink, suggesting wetter climate. In addition, $\delta^{18}O$ was on peak value, which confirms the rapid. In Soreq Cave , $\delta^{18}O$ values where between 55 000 and 52 000 yr b2k higher and show similar conditions as today (Grant et al., 2012). Furthermore, Monticchio tree pollen

25   are on a maximum during GI14 and GI12 (interpreted as pollen Assemblage Zones 13 and 11, see Allen et al., 1999). The core shows GI like appearance for this timespan. The dust record exhibits intermediate values with some minor peaks during stadial phases and H5 event. H4 is not visible due to a tephra layer within the event (Ehrmann et al., 2017). The precipitation was high during this time span considering fast speleothem growth, large amounts of tree pollen and intermediate dust values. The aridity index shows humid conditions, especially during interstadial times for this phase.

30   Glacial conditions are clearly visible between 40 000 and 17 200 yr b2k, with GIs 8 to 1. Speleothem growth is continuous through all time, but slowest during glacial at Dim Cave (Ünal-İmer et al., 2015). Soreq cave speleothem growth also continued showing variability in $\delta^{18}O$ on small scale with some minor peaks during GI3 and 4 . The Monticchio pollen are low to medium on tree content during this time span with some variability. Higher tree pollen amount

is present during interstadials and herbaceous taxa & steppe pollen increase in stadial times (Brauer et al., 2007). The K/C ratio shows H3 to H1 and is apart from that at intermediate values. The glacial time has frequent precipitation minima during Heinrich events and increased precipitation in times of interstadials. The aridity index remains on intermediate values during this time but showing climate ameliorations during interstadials.

5   Within the time span from 17 200 to 10 000 yr b2k are GI1 and the YD as well as the transition towards the Holocene. Speleothem growth at Dim and Soreq cave was fast and the δ¹⁸O values increase in both speleothems and the sediment core LC21. Tree pollen drastically rapidly increase from glacial values of about 30 % to nearly 90 % after Bølling / Allerød. The YD is clearly visible in the SL71 dust record with increased K/C ratio. The aridity index increases strongly towards humid conditions during the early Holocene.

10  During the Holocene (10 000 yr b2k to present) most of the records stay remain constant. Dim Cave speleothem did not grow anymore but δ¹⁸O values from Soreq cave and LC21 show consistent values with a variation between 8 500 and 7 000 yr b2k, where low δ¹⁸O values indicate doubled precipitation and present day temperatures (Bar-Matthews et al., 1997). The amount of tree pollen stays constantly high with a small decrease to 75 % at about 2 000 yr b2k. The largest variation can be seen in the dust content, where the dust deflation reaches maximum values after the early Holocene optimum (EHTO). The large dust

15  deflation, mainly originating from Sahara (Ehrmann et al., 2017), is consistent with the northward extension of the Saharan desert in modern times compared to early Holocene conditions (Jolly et al., 1998). The precipitation and temperature for the Mediterranean region within the Holocene was nearly at present day conditions while the African continent underwent a huge change.

[Figure]

[Figure]

| Dim Cave | | Soreq Cave | | Lago Grande di Monticchio | | M40/4_SL71 | | NGRIP |
|---|---|---|---|---|---|---|---|---|
| Speleothem growth | δ¹⁸O [‰] | Speleothem growth | δ¹⁸O [‰] | Tree pollen [%] | Herb & Grass pollen [%] | K/C [Ratio] | Aridity index | δ¹⁸O [‰] |
| (a) | (b) | (c) | (d) | (e) | (f) | (g) | (h) | (i) |

**Figure S6:** Mediterranean Sea climate over the last 60 000 years: Dim Cave (Ünal-İmer et al., 2015) **(a, b)** and Soreq Cave (Bar-Matthews et al., 2000; Grant et al., 2012) **(c, d)** show speleothem growth phases, which require mobile water from frequent precipitation; **(b, d)** δ¹⁸O data with few apparent GIs comparable to **(j)**, more negative δ¹⁸O values account for more humid conditions; **(e, f)** Lago Grande di Monticchio pollen data (Brauer et al., 2007) are divided into tree- and herb & grass pollen. While trees require more precipitation, grasses are dominant for more arid conditions; **(g)** Dust reconstruction from M40/4_SL71 marine core K/C ratio (Ehrmann et al., 2017) indicates more arid conditions with higher values, lower values account for more humid conditions; **(h)** Aridity index for Mediterranean Sea region as reconstructed from **(a, c, e, g)**. For detailed information see method section; **(i)** δ¹⁸O data from NGRIP ice core (North Greenland Ice Core Project Members et al., 2004) in comparison.

**S7**

~~The Cariaco basin is located off the north central coast of Venezuela, directly south of the "Tortuga" island forming the Gulf of Cariaco. The basin is separated from the Caribbean Sea by a series of shallow sills. Below ~275 m water depth, anoxic conditions occur and lead to excellent preservation conditions (Gibson and Peterson, 2014). The sediment cores from the Cariaco basin are well known for their laminated sediments with darker layers from warmer interstadials in contrast to light colored, bioturbated sediments from stadials indicating deep water oxygenation (Deplazes et al., 2013; Hughen et al., 2004; Peterson et al., 2000). The lightness (L*) values as well as the molybdenum (Mo) content show good accordance with the NGRIP δ¹⁸O, all GIs are visible in the comparison (see Fig. S7). The cores which are used for that comparison are MD03-2661, MD03-2662 and the ODP 165-1002 site which were drilled close to each other and correlated in previous studies~~

(Deplazes et al., 2013; Gibson and Peterson, 2014; Peterson et al., 2000). The pollen were investigated for the timespan from 30 000 to 60 000 years b2k in core MD03-2622 by González and Dupont (2009). Speleothem growth at the Bahamas occurred between 24 000 and 44 000 yr b2k in the Sagittarius Blue Hole as well as between 9 800 and 15 000 yr b2k (Hoffmann et al., 2010). The speleothems were collected at a depth of 15 m below present sea level, a growth continuing during Holocene is impossible due to the rising sea level. Dust from the Sahara region is blown out during all times over the Atlantic and can be found in the Cariaco basin as well. Al/Ti ratios in the bulk sediment are higher during interglacials due to fluvial input (values of 27 represent pure fluvial input) and lower during stadials because of Saharan dust input (values of 14 represent pure Saharan dust input, Yarincik et al., 2000).

The Cariaco basin region is known for its high-resolution time series throughout the last glacial cycle. For the last 60 000 years, there are several variations within the archives. In the Reflectance and Mo data, all GIs are visible, except GI2. This indicates a closely coupled system of the Cariaco basin to the North Atlantic rapid climate changes. The dust values from Al/Ti ratio are in general at intermediate values, lower ratio representing higher Saharan dust contents during Heinrich events. The pollen record show tree pollen contents between 10 % and 50 %. Decreases in tree pollen contents during Heinrich events are related to an increase in salt marsh pollen (see Fig. S7: H5, 48 000 yr b2k). Extremely dry atmospheric conditions for the coastland of Cariaco basin combined with warm temperatures during Heinrich events and stadial times lead to hypersalinity and a drastic change in vegetation (González and Dupont, 2009). The aridity index stays at intermediate values during this time phase with inclines towards higher aridity during (Heinrich-) stadials. The speleothem growth started at 44 000 yr b2k during a phase of enhanced tree pollen content and relatively lower dust values indicating a general increase in precipitation. This increase followed shortly after GI 12, which is visible in the MD03-2622 core in the Mo content as well as in the Reflectance data of Core MD03-2621. Al/Ti ratio shows higher dust amount between 30 000 and 16 000 yr b2k with some variations, but still indicating less precipitation. The Sagittarius Blue Hole speleothem did not grow from 24 000 to 14 500 yr b2k (LGM), the same time as the highest dust values appear within the record. This suggests a relatively arid phase during LGM for the Cariaco region which is also visible within the aridity index. Precipitation seems to increase again together with higher temperatures between 14 500 and 9 300 yr b2k where speleothem growth occurs in the Sagittarius Blue Hole. The growth ends with rising sea level during Holocene. The Al/Ti ratio increases during Holocene towards more fluvial composition indicating more river discharge because of a higher precipitation. The aridity index rises with the onset of Bølling / Allerød but insufficient amount of data prevents an appropriate analysis during the Holocene.

[Figure]

**Figure S7:**  **(a)**  **(b, c)**  **(d)**  **(e)**  **(a-d).**  **(f)**  **(g)**  **(h)**

**S8 Santa Barbara basin:**

The region of the Santa Barbara basin is well known for its brief interstadial events of the past 60 000 years. The basin is located at the inner continental border of Southern California with a depth of about 600 m and contains oxygen-depleted water below 475 m (Behl, 1995; Behl and Kennett, 1996; Heusser, 1998). Due to this depletion zone, an excellent preservation of sediment material, pollen and warves occurs within the ODP-893A. In the benthic foraminifer record of ODP-893A (Cannariato et al., 1999) all GIs are visible as in the $\delta^{18}O$ record of the NGRIP ice core. About 800 km east of the Santa Barbara basin the "Cave of the Bells" speleothem in Arizona is located (Wagner et al., 2010). Aridity in the southwestern USA and climate information stored in the NGRIP show a similar pattern of fast interstadial / stadial changes. Cooler temperatures in high latitudes are connected to increased moisture in this region. Hence, higher $\delta^{18}O$ values were interpreted as warmer

temperatures corresponding to drier winters (Wagner et al., 2010). Also, a speleothem of the "Fort Stanton Cave" in New Mexico can help to understand variabilities within the region and shows the GIs 1-12 (Asmerom et al., 2010). No larger dust deflation is known for that region, hence no paleodust record can be used within the synthesis. The climate today is as in Mediterranean regions with temperatures between 9.9 °C and 18,6 °C and a precipitation of about 600 mm/year (NOAA-

5    NCDC weather service, 2018).

Between 60 000 and 48 000 years b2k (early MIS3, GIs 17-13) a high amount of tree pollen of about 80 % indicates high temperatures and at least moderate precipitation (see Fig. S8S7). The benthic foraminifers show high abundances, which is interpreted as an interstadial signal (Cannariato et al., 1999). Bells Cave speleothems show growth starting at about 54 000 years b2k. The $\delta^{18}$O values of -9 ‰ indicate warm temperatures and moderate precipitation. Similarly, the Fort Stanton

10   speleothem also shows high $\delta^{18}$O values up to -5.5 ‰ during early MIS3 period. This time was characterized by warm temperatures and intermediate precipitation.

The timespan from 48 000 to 27 540 yr b2k comprises the GIs 12 to 3. All of them are visible in the foraminifer content as well as in the Fort Stanton speleothem. The Cave of the Bells speleothem comprises a hiatus between 24 000 and 29 000 yr b2k so that GI2 to 4 are missing within that record. The precipitation and temperature vary from interstadials to stadials, but the

15   amount of tree pollen stays relatively constant at around 80 %. That indicates, the range of temperature and precipitation change did not pass a threshold on which an abrupt vegetational change would have occurred and temperatures and precipitation stayed close to early MIS3 conditions in respect to the stadial / interstadial changes.

From 27 540 to 14 700 yr b2k (onset of the Bølling / Allerød) the time is characterized by a drop of the Dysoxic benthic foraminifer content to low values between 0 and 15 % in contrast to the continuous high amount of tree pollen before. The

20   Bells Cave speleothem's hiatus continues until 24 000 yr b2k and growth started again with the onset of GI-2 (23 340 yr b2k), also visible in the foraminifer content. Speleothem growth at the Fort Stanton speleothem was continuous during this time span. The combination of all proxies shows cooler temperatures and increased moisture for the Santa Barbara basin during this period.

The Holocene part from the onset of Bølling / Allerød (B/A, 14 700 yr b2k) until present shows the YD in all records, but the

25   records end around 10 000 yr b2k. Both speleothems show lower $\delta^{18}$O values as well as ODP-893A low dysoxic foraminifer contents. The vegetational record shows a decrease in tree pollen and a corresponding increase in grass pollen. Most recently, the speleothems and the ODP-893A foraminifer records show the start of the Holocene at 11 700 yr b2k by increased values indicating warmer climate. The precipitation increases drastically with sinking tree pollen amount (Heusser, 1998) towards the Holocene.

30   Due to a lack of a dust record, continuous speleothem growth and insignificant changes in the amount of tree pollen, the aridity index for this region is inconclusive and constantly at intermediate values. Therefore, the Aridity index calculation for this region ends around 10 000 yr b2k.

[Figure]

**Figure S8:S7:** Santa Barbara Basin climate over the last 60 000 years: Bells Cave (Wagner et al., 2010) **(a, b)** and Fort Stanton (Asmerom et al., 2010) **(c, d)** show speleothem growth phases, which require mobile water from frequent precipitation; **(b, d)** δ18O data with few apparent GIs comparable to **(h)** and **(j)**, more positive δ18O values account for increased precipitation; **(e, f)** ODP893A marine core pollen data (Heusser, 1998) are divided into tree- and herb & grass pollen. While trees require more precipitation, grasses are dominant for more arid conditions; **(g)** Aridity index for St. Barbara Basin region as result from **(a, c, e)**. For detailed information see method section; **(h)** Dysoxic Benthic Foraminifer data from marine core ODP893A (Cannariato et al., 1999) with distinguishable GIs in comparison to

speleothem and NGRIP $\delta^{18}O$ data **(b, d, ); (i); (i)** $\delta^{18}O$ data from NGRIP ice core (North Greenland Ice Core Project Members et al., 2004) in comparison.
* * *
~~Long, continuous and high resolution paleoclimate archives from southern hemisphere are scarce beside ice cores from Antarctica, so the processes behind climate observation and climate changes on SH are hard to distinguish. One of the important parts of the global climate system are the southern hemisphere westerlies. Together with the Antarctic circumpolar current, both processes regulate meridional heat flux and transport global climatic signals (Toggweiler et al., 2006). Present days mean annual precipitation is similar for the sites. About 2 500 mm/yr is typical for the speleothem sites in New Zealand as well as for the paleovegetation record in Australia, the mean annual temperature varies around 13 °C. This region is strongly imprinted by the southern westerlies, the Hollywood Cave in the north of the Southern New Zealand island is directly affected as the cave is below the flow current. These speleothems are the closest, long term and well dated ones to the Lynch Crater pollen profile. The speleothem HW3 shows continuous growth for the last 73 000 years (see Fig. S9). More negative isotopic values indicate wetter climate as well as stronger westerlies, accounting for North Atlantic cooling. This speleothem resembles Greenland Ice Core Isotopic profiles and Heinrich events although it is antipodean to Greenland and North Atlantic (Whittaker et al., 2011). Lynch Crater pollen profiles are known for their continuous records. The Crater is caused by a volcanic explosion and is filled with lake and swamp deposits. Inlets from surrounding springs feed the lake, which receives most rainfall between December and April nowadays. Only few dust archives are established for the Australia – Oceania region, high resolution records are missing entirely. The E26.1 marine sediment core from 'Lord Howard Rise' lies in the Tasmanian Sea between the pollen and speleothem record and comprises the dust history of the last 350 000 years, sadly on low resolution of about 5 000 years but with significantly higher dust rates during glacial times (Fitzsimmons et al., 2013; Hesse, 1994; Hesse and McTainsh, 2003).~~

~~The timespan from 60 000 to 52 000 yr b2k (GIs 17 to 14) is marked by very high amounts of tree pollen. According to Kershaw (1976), subtropical rainforest surrounded the Lynch Crater. This would support the lower $\delta^{18}O$ values of Hollywood Cave speleothem accounting for wetter climate conditions (Whittaker et al., 2011). Speleothem growth is apparent throughout the last 60 000 years with continuous growth but different growth rates, which were relatively high during this period. Dust record from E26.1 is relatively low but indicating an increase towards the end of that period. The aridity was low during this time, as indicated by high amounts of tree pollen, high speleothem growth rates and lower dust values.~~

~~From 52 000 to 39 000 yr b2k (GIs 13-9) the climate was fluctuating. While the speleothem growth rates were high at the beginning and the end of that period, the time between 44 000 and 48 000 yr b2k shows the lowest growth rates of the record. The amount of tree pollen decreases to about 40 % in average, also with major variations. The isotope values of the speleothem remain low, accounting for wetter conditions with a clearly visible H5 and 4 events around 48 000 and 30 000 yr b2k with lowest isotope values through that time (Whittaker et al., 2011). A dust peak comes along with a low tree pollen zone around~~

50 000 yr b2k and a decrease in dust afterwards according to H5 times with more precipitation related to less dust transport. The aridity index for this time shows variable, humid conditions and resemble the climate amelioration of southern hemisphere during H5 as well as the following intermediate conditions.

Between 39 000 and 29 000 yr b2k (GIs 8 – 5), climate getting unstable also visible in intermediate aridity. Although the speleothem continuously grows, the isotopes indicate drier conditions than before, beside an increase during H3 (~ 30 000 yr b2k) accounting for wetter conditions towards the end. The amount of tree pollen decreases to values around 20 % according for minimum of that time. The vegetation changed to sclerophylls, which needs less precipitation compared to rainforest vegetation before (Kershaw, 1976). Dust stays on intermediate values through that time. An increase in tree pollen going along with a decrease in isotopes $\delta^{18}O$ and $\delta^{13}C$ in Hollywood Cave mark the begin of the next time phase (LGM from 29 000 to 16 000 yr b2k, GIs 4 – 2 within) with climate amelioration for parts of this region. A drastic increase in tree pollen to about 55 %, increased speleothem growth and lower isotopes indicate higher precipitation and warmer climate, contrarily the dust content also increases during LGM, indicating arid conditions for Central Australia dust deflation regions with increased dune activity (Fitzsimmons et al., 2013). The dust peak in Tasmanian Sea is synchronous to the EDML dust maximum from Antarctica (Wegner et al., 2015). The aridity index displays variable precipitations at low to intermediate conditions at average. After LGM ends (16 000 yr b2k to present, GI1) the onset towards the Holocene took place. Younger Dryas (NH cold event) is clearly visible as a tree pollen spike within the otherwise intermediate values (around 30 %). Speleothem growth rates were high until 10 000 yr b2k, no further isotope data are publically available for this speleothem. Dust values decrease throughout the whole period, also showing the YD as warm / wet event, supporting the idea of a YD warm event for southern hemisphere (Carlson, 2013; Shakun and Carlson, 2010). Varying tree pollen amounts are apparent for the rest of the Holocene, dust decreases further and speleothems grow, indicating humid conditions until 5 000 yr b2k and a drastic decrease in climate conditions afterwards.

[Figure]

**Figure S9:** ~~Australia - Oceania climate over the last 60 000 years: **(a)** Hollywood Cave New Zealand (Hellstrom et al., 1998; Whittaker et al., 2011; Williams, 1996; Williams et al., 2005) shows speleothem growth phases, which require mobile water from frequent precipitation; **(b)** δ¹⁸O data with few apparent GIs comparable to **(g)**, more positive δ¹⁸O values account for increased precipitation (Whittaker et al., 2011); **(c, d)** Lynch Crater pollen data from NE Australia (Kershaw, 1994) are divided into tree- and herb & grass pollen. While trees require more precipitation, grasses are dominant for more arid conditions; **(e)** Aeolian content from E26.1 marine core (Hesse, 1994) indicates more arid conditions with higher values, lower values account for more humid conditions; **(f)** Aridity index for Australia - Oceania region as result from **(a, c-e).** For detailed information see method section; **(g)** δ¹⁸O data from NGRIP ice core (North Greenland Ice Core Project Members et al., 2004) in comparison. **(h)** Dust concentration from EDML (Wegner et al., 2015), higher particle concentrations indicate more arid conditions and vice versa; **(i)** δ¹⁸O data from WAIS Divide ice core (WAIS Divide Project Members et al., 2015) in comparison to speleothem and NGRIP δ¹⁸O data **(b, g)**.~~

[Figure]

**Figure S10:S 8 Global tree pollen pattern**

[Figure]

**Figure S87:** Global tree pollen records for the last 60 000 years. Higher amounts of tree pollen indicate increased humidity. **(a)** Central European ELSA-Vegetation-Stack (Sirocko et al., 2016); **(b)** Chinese Sihailongwan Maar Lake (Mingram et al., 2018); **(c)** North-West Africa marine core GIK15627-3 (Hooghiemstra et al., 1992); **(d)** Southern Europe Lac Du Bouchet (Reille and de Beaulieu, 1990); **(e)** Portuguese margin marine core MD95-2039 (Roucoux et al., 2005); **(f)**  Lago Grande di Monticchio for Mediterranean Sea region (Brauer et al., 2007); **(hg)** St. Barbara Basin marine core ODP893A (Heusser, 1998)

Figure S8 shows all tree pollen of this synthesis. Only three regions worldwide encompass sediment cores with pollen data for the whole last 60 000 years. The ELSA-Vegetation-Stack for Central Europe (Sirocko et al., 2016) consists of the sediment cores from Holzmaar and Dehner Maar, which cover the last 60 000 years in total. The Mediterranean record from 'Lago Grande di Monticchio' also covers this time as

well as pollen from the Sihailongwan Maar Lake (China). However, Sihailongwan Maar Lake pollen profiles of Mingram (Mingram et al., 2018) and Stebich (Stebich et al., 2015) diverge for Holocene and need to be adjusted. Pollen records for the Arabian Sea region are not available. The pollen assemblage of St. Barbara Basin is on very high values throughout the whole record due to sedimentation pattern in the marine basin as regional effects (Heusser, 1995, 1998).

5   Early MIS3 phase shows tree pollen maxima for the following regions: Central Europe, China, Portuguese margin, ~~Mediterranean Sea, and Oceania. Although, the timings of the pollen maxima are not synchronous, enhanced precipitation and humidity for early MIS3 can be stated for these regions. The tree pollen show impairing climate conditions during late MIS3 towards LGM conditions. A well expressed LGM is visible in the pollen archives of Central Europe, China, Southern Europe, Portuguese margin and the Mediterranean Sea while the Oceania records shows increased tree pollen. This northern / southern~~

10   ~~hemisphere atmospheric teleconnection currently described by e.g. Shakun and Carlson (2010) and initially described by Sirocko et al. (1996b). Apart from Oceania, the LGM was mainly arid for all regions. The following transition towards the Holocene with the onset of Bølling / Allerød (14 700 yr b2k) is visible in all pollen records by massive changes in the tree pollen content. St. Barbara Basin and Oceania show decreasing pollen values (see S8 & S9).~~ and Mediterranean Sea. Although, the timings of the pollen maxima are not synchronous, enhanced precipitation and humidity for early MIS3 can be stated for

15   these regions. The tree pollen show impairing climate conditions during late MIS3 towards LGM conditions. A well expressed LGM is visible in the pollen archives of Central Europe, China, Southern Europe, Portuguese margin and the Mediterranean. The LGM was mainly arid for all regions. The following transition towards the Holocene with the onset of Bølling / Allerød (14 700 yr b2k) is visible in all pollen records by massive changes in the tree pollen content. St. Barbara Basin shows decreasing pollen values (see S7). All other regions show increasing tree pollen amounts indicating more humid conditions

20   and climate amelioration for the early Holocene.

[Figure]

**Figure S11:S9:** Global age / depths relations of various sediment cores. Solid lines show cores used for this synthesis. Thick red line shows Lago Grande di Monticchio, which is warve counted and shows GI like appearance. Points show apparent GIs. Dotted lines give other well-known high resolution records not included in this synthesis. For references of included cores see chapter 2 and S1-S9S7; Potrok Aike, Southern Patagonia, Argentina (Kliem et al., 2013); Lake Tulane, Florida, USA (Grimm et al., 2006); Petén-Itzá, Guatemala (Correa-Metrio et al., 2012); Bear Lake, Utah-Idaho, USA (Jiménez-Moreno et al., 2007); Lake Suigetsu, Japan (Bronk Ramsey et al., 2012).

Stratigraphical uncertainties are the major sources of error for this synthesis. All stratigraphies of original publications remained unchanged, apart from referring all records to years b2k (before the year 2000 CE). Different dating methods in

general have various uncertainties. Archives of this synthesis were dated by $^{14}$C ages, optically stimulated luminescence (OSL), U/Th dating of speleothems, varve counting , tuning to NGRIP or marine isotope stack, paleomagnetismus correlation, tephra chronology or cross correlating (see Tab. S1). Age uncertainties of speleothem U/Th dating range between 1 and 4 %, resulting in the most reliable ages for speleothems. $^{14}$C calibrations were greatly improved and e.g. Hughen et al.

5  (2006) has calibrated the $^{14}$C curves with the absolutely dated Hulu speleothems. Nevertheless, $^{14}$C ages of this synthesis bear maximum errors up to 12 % - or up to 7 000 years. Error values of OSL dating range up to 10 % or about 5 000 years for the last 60 000 years. Warve counting errors depend strongly on the sedimentation rate of the archive, but error values can stack up to 5 000 years as well. For NGRIP tuning, age uncertainties of the ice core are minimized by several dating approaches and stack up to 2 500 years for the last 60 000 years. To sum it up, no definitive ages for the previously described turning point

10  can be determined from this work. Nevertheless, synchronous patterns occurred during climate history and the estimated ages give a good evaluation of the turning points of the precipitation.

The sampling resolution of the records is also extremely important. More samples and analyses result in a much more detailed explanations and interpretations for the records. High frequency sampling, especially in sediment cores is mainly depending on the sedimentation rate. Figure S9 shows all sediment archives and records of this work with the age-depths relation.

15  The thick, red line accounts for the warve-counted Lago Grande di Monticchio core. Points indicate GI appearance in the marine records of Santa Barbara Basin (ODP893),  Chinese Loess (Jingyuan Loess sequence), Arabian Sea (SO-136KL) and Portuguese margin (MD01-2444) and Lake sediments of Central Europe (ELSA-Dust-Stack). The highest sedimentation rates during the last 60 000 years can be found in Santa Barbara Basin and ELSA-AU2 core with about 1-2 mm/year.

20  Several other high resolution records with extend into MIS3 (Fig. S9, dotted lines) are published but not used for this synthesis. They are additionally shown for comparison and completeness. These cores were too far away from the chosen ten key regions to fit into a proper synthesis or do not extend until the begin of MIS3 (60 000 yr b2k). However, these cores are good climate archives with striking sedimentation rates of 0.26 mm/yr (Lake Tulane), 0.5 mm/yr (Bear Lake), 0.75 mm/yr (Lake Suigetsu) up to Petén-Itzá (~1 mm/yr) and Potrok Aike (~1.6 mm/yr) and should be mentioned within this synthesis.

| Region | Archive | Proxy type | Location | Time covered [ka b2k] | Dating method | Reference |
|---|---|---|---|---|---|---|
| Central Europe | Speleothem | Speleothem | Bunker Cave | 0 - 52 | 1, 2 | Fohlmeister et al., 2012, 2013; Weber et al., 2018 |
| | Speleothem | Speleothem | Spannagel Cave | 0 - 275 | 1 | Holzkämper et al. 2005; Spötl & Mangini, 2002, 2012 |
| | Lake Sediment core | Pollen | Dehner Maar & Holzmaar | 0 - 63 | 2, 3, 4, 5 | Sirocko et al., 2016 |
| | Lake Sediment core | Dust | Dehner Maar | 0 - 130 | 2, 6, 5 | Seelos et al., 2009 |
| Arabian Sea | Speleothem | Speleothem | Socotra Cave | 2 - 5; 40 - 54 | 1 | Fleitmann et al., 2007 and refs. within; Burns et al., 2003 |
| | Speleothem | Speleothem | Oman Caves | 0 - 11 | 1 | Fleitmann et al., 2007 and refs. within |
| | Marine Sediment core | Dust | Western Arabian Sea | 0 - 135 | 2, 4, 7 | Leuschner & Sirocko, 2003 |
| | Marine Sediment core | TOC | Western Arabian Sea | 0 - 65 | 2, 4, 8 | Schulz et al., 1998 |
| | Marine Sediment core | Reflectance | Western Arabian Sea | 0 - 80 | 2, 5 | Deplazes et al., 2014 |
| NW-Africa | Speleothem | Speleothem | Susah Cave | 30 - 66 | 1 | Hoffmann et al., 2016 |
| | Speleothem | Speleothem | Grotte Prison de Chien | 3 - 97 | 1 | Wassenburg et al., 2012 |
| | Marine Sediment core | Pollen | West of Morocco | 6 - 250 | 2, 8 | Hooghiemstra et al., 1992 |
| | Marine Sediment core | Dust | West of Senegal | 0 - 57 | 2, 8 | Collins et al., 2013 and refs. within |
| | Marine Sediment core | SST | West off Portugal | 0 - 110 | 2, 8 | Bard, 2002 |
| China | Speleothem | Speleothem | Hulu Cave | 11 - 75 | 1, 8 | Wang et al., 2001; Liu et al., 2010 |
| | Lake Sediment core | Pollen | Sihailongwan Maar Lake | 0 - 65 | 2, 3, 4 | Mingram et al., 2018 |
| | Lake Sediment core | Pollen | Sihailongwan Maar Lake | 0 - 12 | 2 | Stebich et al., 2015 |
| | Loess profile | Dust | Jingyuan Loess | 0 - 80 | 5, 6, 9 | Sun et al., 2010 |
| Southern Europe | Speleothem | Speleothem | Villars Cave | 32 - 83 | 1 | Genty et al, 2003 |
| | Speleothem | Speleothem | Villars Cave | 5 - 15 | 1 | Genty et al., 2006 |
| | Speleothem | Speleothem | Villars Cave | 28 - 51 | 1 | Wainer et al., 2009 |
| | Lake Sediment core | Pollen | Lac Du Bouchet | 15 - 68 | 2, 10 | Reille & Beaulieu, 1989; |
| | Marine Sediment core | Dust | MD01-2443/4 | 0 - 420 | 2, 5, 7, 8, 9 | Hodell et al., 2013, Barker et al., 2011 |
| | Marine Sediment core | IRD | IODP-306-U1313 | 0 - 70 | 2, 5, 9 | Naafs et al., 2013 |
| Portuguese margin | Speleothem | Speleothem | Villars Cave | 32 - 83 | 1 | Genty et al., 2003 |
| | Speleothem | Speleothem | Villars Cave | 5 - 15 | 1 | Genty et al., 2006 |
| | Speleothem | Speleothem | Villars Cave | 28 - 51 | 1 | Wainer et al., 2009 |
| | Marine Sediment core | Pollen | MD95-2039 | 10 - 65 | 2, 5, 9 | Roucoux et al., 2005 |
| | Marine Sediment core | Dust | MD01-2443/4 | 0 - 420 | 2, 5, 7, 8, 9 | Hodell et al., 2013, Barker et al., 2011 |
| | Marine Sediment core | IRD | IODP-306-U1313 | 0 - 70 | 2, 5, 9 | Naafs et al., 2013 |
| Mediterranean Sea | Speleothem | Speleothem | Dim Cave | 5 - 85 | 1 | Ünal-İmer et al., 2015 |
| | Speleothem | Speleothem | Soreq Cave | 0 - 150 | 1 | Grant et al., 2012 |
| | Speleothem | Speleothem | Soreq Cave | 0 - 140 | 1 | Bar-Matthews et al., 2000 |
| | Lake Sediment core | Pollen | Lago Grande di Monticchio | 0 - 130 | 2, 3, 9 | Brauer et al., 2007 |
| | Marine Sediment core | Dust | MD40/4 _SL71 | 0 - 180 | 2, 4, 8 | Ehrmann et al., 2017 |
| | Marine Sediment core | Foraminifers δ18O | LC21 | 0 - 156 | 2, 8, 9 | Grant et al., 2012 |
| Cariaco Basin | Speleothem | Speleothem | Sagittarius Blue Hole, Bahamas | 9 - 45 | 1, 2 | Hoffmann et al., 2010 |
| | Marine Sediment core | Pollen | MD03-2622 | 28 - 75 | 2, 8, 9 | González & Dupont, 2009 |
| | Marine Sediment core | Dust | ODP 1002C | 0 - 578 | 2, 7, 8, 9 | Yarincik et al., 2000; Haug et al., 2000 |
| | Marine Sediment core | Reflectance | MD03-2621 | 0 - 80 | 2, 9 | Deplazes et al., 2013 |
| St. Barbara Basin | Speleothem | Speleothem | Bells Cave | 8 - 53 | 1 | Wagner et al., 2010 |
| | Speleothem | Speleothem | Fort Stanton | 11 - 57 | 1 | Asmerom et al., 2010 |
| | Marine Sediment core | Pollen | ODP 893A | 9 - 135 | 2, 8, 9 | Heusser, 1998 |
| | Marine Sediment core | Foraminifers δ18O | ODP 893A | 0 - 60 | 2, 8, 9 | Cannariato et al. 1999 |
| Australia - Oceania | Speleothem | Speleothem | Hollywood Cave, NZ | 0 - 140 | 1 | Hellstrom et al., 1998; Whittaker et al., 2011; Williams, 1996; Williams et al., 2005 |
| | Lake Sediment core | Pollen | Lynch Crater | 0 - 192 | 2 | Kershaw, 1994 |
| | Marine Sediment core | Dust | E26.1 | 0 - 340 | 2, 7 | Hesse, 1994 |
| Greenland | Ice Core | δ18O | NGRIP | 0 - 105 | 3 | NGRIP Members et al., 2004 |
| Antarctica | Ice Core | δ18O | WAIS Divide | 0 - 68 | 3 | WAIS Divide Project Members, 2015 |

**S10 Matlab code for error calculation:**

```
%Aridity Index simulation
%input here your estimated error as [%]:
fehlerSp=2;
fehlerPo=3;
fehlerDu=5;

%read the CSV data without headlines. "," separated
m = csvread('filename_template.csv');

%Separate on columns
%m3=m(:,3)
%m4=m(:,4)
%m5=m(:,5)

mat=zeros(3,1201,100);
%mat(:,1)=m3

for k=1:3

    for i=1:100
        mat(k,:,i)=m(:,k+2);
    end
end

% fill matrix with random numbers
for a=1:1201
    nSp=mod(mat(1,a,1)+1,2);
    for i=1:fehlerSp
        mat(1,a,i)= nSp;
    end

    nPo=mat(2,a,1);
    if nPo==1
        for i=1:fehlerPo/2
            mat(2,a,i)=0;
        end

        for j=i:fehlerPo
            mat(2,a,j)=2;
        end
    else
        for i=1:fehlerPo
            mat(2,a,i)=1;
        end
    end

    nDu=mat(3,a,1);
    if nDu==1
```

```matlab
            for i=1:fehlerDu/2
                mat(3,a,i)=0;
            end

            for j=i:fehlerDu
                mat(3,a,j)=2;
            end
        else
            for i=1:fehlerDu
                mat(3,a,i)=1;
            end
        end

    end

%simulation starts here-->
%number of simulations
noRuns = 100000;

%Std Matrix erstellen
sims=zeros(noRuns,1201);

for sIndex=1:noRuns
    %1201 Random Werte zwischen 0 und 2
    r1 = ceil(rand(1201,1)*100);

    %recalculate index from mat
    vec_speleothem=zeros(1,1201);
    vec_pollen=zeros(1,1201);
    vec_dust=zeros(1,1201);

    for i=1:1201
        vec_speleothem(i)=mat(1,i,r1(i));
        vec_pollen(i)=mat(2,i,r1(i));
        vec_dust(i)=mat(3,i,r1(i));
    end

    tmp =(vec_speleothem+vec_pollen+vec_dust);
    sims(sIndex,:) =tmp;
end

result=zeros(1201:1);

%stadtad deviation for each value individually
for k=1:1201
    result(k) = std(sims(:,k));
end
resultTransp=result.';
csvwrite('error_template_name.txt',resultTransp);
```

**Time covered**

| Region | Archive | Proxy type | Location | [ka b2k] | Dating method | Reference |
|---|---|---|---|---|---|---|
| Central Europe | Speleothem | Speleothem | Bunker Cave | 0 - 52 | 1, 2 | Fohlmeister et al., 2012, 2013; Weber et al., 2018 |
| | Speleothem | Speleothem | Spannagel Cave | 0 - 275 | 1 | Holzkämper et al. 2005; Spötl & Mangini, 2002, 2012 |
| | Lake Sediment core | Pollen | Dehner Maar & Holzmaar | 0 - 63 | 2, 3, 4, 5 | Sirocko et al., 2016 |
| | Lake Sediment core | Dust | Dehner Maar | 0 - 130 | 2, 6, 5 | Seelos et al., 2009 |
| Arabian Sea | Speleothem | Speleothem | Socotra Cave | 2 - 5; 40 - 54 | 1 | Fleitmann et al., 2007 and refs. within; Burns et al., 2003 |
| | Speleothem | Speleothem | Oman Caves | 0 - 11 | 1 | Fleitmann et al., 2007 and refs. within |
| | Marine Sediment core | Dust | Western Arabian Sea | 0 - 135 | 2, 4, 7 | Leuschner & Sirocko, 2003 |
| | Marine Sediment core | TOC | Western Arabian Sea | 0 - 65 | 2, 4, 8 | Schulz et al., 1998 |
| | Marine Sediment core | Reflectance | Western Arabian Sea | 0 - 80 | 2, 5 | Deplazes et al., 2014 |
| NW-Africa | Speleothem | Speleothem | Grotte de Piste | 5 - 12 | 1 | Wassenburg et al., 2012, 2016 |
| | Speleothem | Speleothem | Grotte Prison de Chien | 3 - 97 | 1 | Wassenburg et al., 2012 |
| | Marine Sediment core | Pollen | West off Morocco | 6 - 250 | 2, 8 | Hooghiemstra et al., 1992 |
| | Marine Sediment core | Dust | West off Senegal | 0 - 57 | 2, 8 | Collins et al., 2013 and refs. within |
| China | Speleothem | Speleothem | Hulu Cave | 11 - 75 | 1, 8 | Wang et al., 2001; Liu et al., 2010 |
| | Lake Sediment core | Pollen | Sihailongwan Maar Lake | 0 - 65 | 2, 3, 4 | Mingram et al., 2018 |
| | Lake Sediment core | Pollen | Sihailongwan Maar Lake | 0 - 12 | 2 | Stebich et al., 2015 |
| | Loess profile | Dust | Jingyuan Loess | 0 - 80 | 5, 6, 9 | Sun et al., 2010 |
| Southern Europe | Speleothem | Speleothem | Villars Cave | 5 - 15; 32 - 83 | 1 | Genty et al., 2006; Genty et al., 2003 |
| | Speleothem | Speleothem | Villars Cave | 0 - 3 | 1 | Labuhn et al., 2015 |
| | Speleothem | Speleothem | Villars Cave | 28 - 51; 11 - 178 | 1 | Wainer et al., 2009; 2011 |
| | Lake Sediment core | Pollen | Lac Du Bouchet | 15 - 68 | 2, 10 | Reille & Beaulieu, 1989; |
| | Marine Sediment core | Dust | MD01-2443/4 | 0 - 420 | 2, 5, 7, 8, 9 | Hodell et al., 2013, Barker et al., 2011 |
| Portuguese margin | Speleothem | Speleothem | Buraca Gloriosa Cave | 0 - 220 | 1 | Denniston et al., 2018 |
| | Marine Sediment core | Pollen | MD95-2039 | 0 - 65 | 2, 5, 9 | Roucoux et al., 2005, Vogelsang, E., 2001 |
| | Marine Sediment core | Dust | MD01-2443/4 | 0 - 420 | 2, 5, 7, 8, 9 | Hodell et al., 2013, Barker et al., 2011 |
| Mediterranean Sea | Speleothem | Speleothem | Dim Cave | 5 - 85 | 1 | Ünal-Imer et al., 2015 |
| | Speleothem | Speleothem | Soreq Cave | 0 - 150 | 1 | Grant et al., 2012 |
| | Speleothem | Speleothem | Soreq Cave | 0 - 140 | 1 | Bar-Matthews et al., 2000 |
| | Lake Sediment core | Pollen | Lago Grande di Monticchio | 0 - 130 | 2, 3, 9 | Brauer et al., 2007 |
| | Marine Sediment core | Dust | MD40/4_SL71 | 0 - 180 | 2, 4, 8 | Ehrmann et al., 2017 |
| St. Barbara Basin | Speleothem | Speleothem | Bells Cave | 8 - 53 | 1 | Wagner et al., 2010 |
| | Speleothem | Speleothem | Fort Stanton | 11 - 57 | 1 | Asmerom et al., 2010 |
| | Marine Sediment core | Pollen | ODP 893A | 9 - 135 | 2, 8, 9 | Heusser, 1998 |
| | Marine Sediment core | Foraminifers ẟ18O | ODP 893A | 0 - 60 | 2, 8, 9 | Cannariato et al, 1999 |
| Greenland | Ice Core | ẟ18O | NGRIP | 0 - 105 | 3 | NGRIP Members et al., 2004 |
| Antarctica | Ice Core | ẟ18O | WAIS Divide | 0 - 68 | 3 | WAIS Divide Project Members, 2015 |

**Table S1:** Paleoclimatic records used in this synthesis. Records with grey background are not included into the generation of the aridity index. Dating methods: 1 Th/U-dating; 2 $^{14}$C-Radiocarbon dating; 3 Varve counting; 4 Tephrachronology; 5 Ice core tuning; 6 OSL Luminescence dating; 7 Orbital tuning; 8 Oxygen isotope tuning; 9 Cross correlation; 10 Paleomagnetic correlation

List of relevant changes:
- Linguistic revision of the entire manuscript
- Clarifications on methods and rephrasing the whole method section, including error estimations
- Deleted regions Oceania & Cariaco Basin as referees suggested
- Portuguese margin & NW-Africa: speleothem record changed, according to referees suggestions and mistakenly wrong cave selection by us
- Holocene section for villars cave speleothem added
- Unnecessary information throughout the manuscript were deleted
- Fig.2 (truncated before) changed
- Scripts and templates now available as supplementary material
- Discussion section on PalMod requirements impacting the site selection and results added

---

## Author Response (AR4)

Dear Authors,
I kindly ask you to share the comments on the uncertainty assessment with the readers, as suggested by the reviewer.

Dear Authors,
many thanks for clarifying the points. Most is settled in my opinion.
You clarified the issue with uncertainty assessment to me. Please make this also clear to the reader, and clarify for the reader that the uncertainty is difficult to assess quantitatively, and that these estimates are estimates for maximum uncertainty but may not be accurate. Please add this to the discussion, thereafter the manuscript can be published in my opinion.

Dear Reviewer,
Thanks again for your comment. As you suggest, we have included the comments about uncertainty assessment into the section 'Error estimation' (additions in blue):

[revised manuscript text omitted]